# Spatial coding dysfunction and network instability in the aging medial entorhinal cortex

Charlotte S. Herber [1] ✉, Karishma J. B. Pratt[2,5], Jeremy M. Shea[2,5], Saul A. Villeda [2,3] & Lisa M. Giocomo [1,4] ✉

Across species, spatial memory declines with age, possibly reflecting altered hippocampal and medial entorhinal cortex (MEC) function. However, the integrity of cellular and network-level spatial coding in aged MEC is unknown. Here, we leveraged in vivo electrophysiology to assess MEC function in young, middle-aged, and aged mice navigating virtual environments. In aged grid cells, we observed impaired stabilization of context-specific spatial firing, correlated with spatial memory deficits. Additionally, aged grid networks shifted firing patterns often, but with poor alignment to context changes. Aged spatial firing was also unstable in an unchanging environment. In these same mice, we identified 458 genes differentially expressed with age in MEC, 61 of which had expression correlated with spatial coding quality. These genes were interneuron-enriched and related to synaptic plasticity, notably including a perineuronal net component. Together, these findings identify coordinated transcriptomic, cellular, and network changes in MEC implicated in impaired spatial memory in aging.

Numerous cognitive domains decline over the human lifespan[1], posing a significant challenge to our aging societies[2]. In particular, declining spatial cognition limits the functional independence of aged individuals, as learning new routes and returning efficiently to remembered locations becomes more difficult[3–5]. Non-human primates[6] and rodents[7,8] also experience age-mediated spatial memory decline. To address this behavioral dysfunction, it is critical to better understand aging-dependent molecular, cellular, and circuit-level changes in the neural systems that support spatial cognition.

Across mammalian species, neural systems in the medial temporal lobe, including the medial entorhinal cortex (MEC) and hippocampus (HPC), are required for spatial memory[9–11]. The MEC contains grid cells that fire periodically during environmental traversals and have firing fields that hexagonally tile physical space in rodents[12], non-human primates[13], and humans[14]. This firing is proposed to provide a map of space that can support path integration[15,16]. Head direction-[17],

border-[18,19], speed-[20,21], and object vector-tuned[22] cells have also been identified in MEC, providing information regarding an animal's movement through the environment and sensory features likely relevant to navigation. Additionally, MEC neurons can change their firing rates or shift where their firing fields are active, phenomena collectively referred to as 'remapping'[19,23–30]. MEC remapping events often occur in response to changes in task demands and environmental features[19,24–30], potentially facilitating the differentiation of distinct contexts. Such remapping in MEC grid cells is likely complemented by place cells[31] and goal-vector cells[32,33] in the reciprocally connected HPC, which can also exhibit context-dependent remapping[34–36]. Collectively, this network of functional cell types across MEC and HPC may provide the necessary neural substrates for an animal to navigate to goals in novel and familiar environments.

Several lines of evidence suggest that MEC-HPC circuit dysfunction contributes to aged spatial memory deficits. Aged humans

¹Department of Neurobiology, Stanford University School of Medicine, Stanford, CA, USA. ²Department of Anatomy, University of California San Francisco, San Francisco, CA, USA. ³Bakar Aging Research Institute, San Francisco, CA, USA. ⁴Howard Hughes Medical Institute, Stanford University School of Medicine, Stanford, CA, USA. ⁵These authors contributed equally: Karishma J. B. Pratt, Jeremy M. Shea. ✉e-mail: csh47@stanford.edu; giocomo@stanford.edu

navigating novel virtual environments take less accurate and more inefficient paths to learned destinations in a manner correlated with reduced activation of medial temporal lobe structures[37,38]. Both aged humans[4, 38, 39] and rodents[8] avoid allocentric navigation strategies, suggesting a failure to map novel environments or to retain such a map. This idea is supported by recordings of aged rodent place cells, which are both less stable over days in the same environment[40,41] and less flexible in novel environments, failing to remap with changing context cues[42,43]. More indirectly, fMRI experiments uncovered compromised grid-like representations in the MEC that correlated with the extent of path integration errors in aged humans[44]. Longitudinal structural imaging has also revealed that small EC volume changes are sufficient to predict human spatial memory decline over time[45]. Notably, neural circuits in the MEC are affected early in forms of aging-mediated cognitive impairment, including preclinical Alzheimer's disease (AD)[46,47]. Additionally, grid-like representations are impaired in young adults at risk of developing AD due to expression of the APOE-ε4 allele[48], and MEC grid coding is degraded in multiple transgenic rodent models of AD[49–51]. However, changes in MEC spatial coding have yet to be directly investigated in normal aging.

In particular, it is unclear how aging impacts the quality or stability of tuning to navigational variables across MEC functional cell types. The integrity and flexibility of population-level spatial maps in the aged MEC also remain unknown. Since the HPC and MEC are reciprocally connected, one possibility is that spatial coding dysfunction in these regions might interdependently contribute to spatial memory decline in aging. Eventually, rejuvenating aged spatial cognition dependent on MEC-HPC networks will also require a more precise understanding of the molecular mechanisms that drive cellular and circuit dysfunction. Towards this goal, mouse brain aging across regions has been comprehensively studied using bulk and single-cell transcriptomic approaches[52–54]. However, characterizing aging-dependent changes in MEC across phenotypic levels in the same animals would provide unique insight into which genes might drive cellular and circuit dysfunction relevant to behavioral impairment.

Here, we combined in vivo silicon probe recordings[55] with neuronal bulk sequencing in MEC in the same mice, complemented by single-nucleus RNA sequencing (snRNA-seq), to identify neural and molecular substrates of aged spatial memory function. Advanced electrophysiologic tools permitted the simultaneous recording of hundreds of neurons per day from each mouse. As a result, we could robustly analyze age effects on MEC spatial coding at the animal level. Moreover, we interrogated how aging altered single neuron firing patterns and population-level spatial coding phenomena. Using a virtual-reality (VR) task with two dynamically interleaved contexts and another with invariant cues[23], we demonstrated how aging impacts the flexibility and stability of MEC spatial coding at both these levels. Finally, by correlating key spatial coding metrics with the expression of neuronal genes differentially expressed across age groups, we identified potential molecular drivers of aging-mediated spatial cognitive decline in MEC.

## Results
### Assessing spatial memory and entorhinal neural activity in virtual reality (VR)
To identify neural correlates of spatial memory impairment in the aging MEC, we first recorded MEC neural activity from head-fixed young (mean age ± standard error of the mean [SEM] = 2.58 ± 0.07 months, $n = 4$ male, 5 female mice), middle-aged (MA, 12.63 ± 0.09 months, $n = 5$ male, 5 female mice), and aged (22.05 ± 0.021 months, $n = 5$ male, 5 female mice) C57Bl/6 mice navigating VR environments (Fig. 1). To record neural activity during behavior, we acutely inserted Neuropixels silicon probes[56] into the MEC for up to six neural recordings per mouse (up to three in each hemisphere) (see "Methods" section) (Fig. 1a). Using this approach, we

recorded in vivo activity from thousands of cells in each age group ($n = 15,152$ young, 15,011 MA, and 13,225 aged cells).

To interrogate spatial memory during navigation, mice performed a VR task that contained two hidden reward locations (the Split Maze [SM] task). On each traversal of the 400 cm linear VR track, called a trial, mice could request water rewards by licking within one of two reward zones, spanning 50 cm (Fig. 1b). Each reward location was associated with distinct visual cues (e.g., floor patterns, landmark tower shape) in the second half of the track, termed context A or B (Fig. 1b and Supplementary Fig. 1a). To control for the differences in proximity to landmarks, reward-context associations were counterbalanced within each age group. In each session, contexts A and B were presented in successive groups of 60 trials each (blocks) and then pseudo-randomly alternated for 80 trials (alternation) (Fig. 1c). At the beginning of each block, rewards were dispensed automatically for ten trials to indicate the reward location. After this, accurate context discrimination and licking were required to receive rewards.

By the sixth session, young and MA mice exhibited licking at the reward zone on block and alternation trials, while aged mice failed to lick consistently in either context during alternation (Fig. 1f and Supplementary Fig. 1). To quantify behavioral performance improvements over days while accounting for variance among animals, we fit linear mixed effects models (LMMs) to the fraction of rewards requested during blocks and alternation over sessions, with animal identity as a random effect (see "Methods" section). Block performance improved equivalently across age groups over sessions (Session, $\beta = 0.045$, $p = 0.004$; session × aged, $\beta = -0.008$, $p = 0.709$; aged vs young, $\beta = -0.182$, $p = 0.079$). By contrast, alternation performance improved significantly less for aged vs young mice (Session, $\beta = 0.106$, $p < 0.0001$; session × aged, $\beta = -0.085$, $p < 0.0001$; aged vs young, $\beta = -0.292$, $p = 0.059$) (Fig. 1g). This indicates an aging-mediated deficit in context discrimination during rapid context alternation.

To control for possible age-mediated differences in the ability to run or motivation to lick, we next implemented a simpler task in a separate group of mice, in which they could lick for rewards at randomly appearing, visually marked zones (Random Foraging [RF]) (Fig. 1d)[23]. Other track visual cues were invariant on all trials (see "Methods" section) (Fig. 1e and Supplementary Fig. 1b). Using the same electrophysiological approach as in the SM task, we recorded 10,590 cells from young mice (mean age ± SEM = 4.23 ± 0.56 months, $n = 5$ male, 3 female mice) and 10,228 cells from aged mice (mean age ± SEM = 22.99 ± 0.51 months, $n = 2$ male, 5 female mice). Across age groups, mice exhibited equivalent behavioral performance in this task (Session, $\beta = 0.064$, $p = 0.0001$; session × aged, $\beta = 0.008$, $p = 0.737$; aged vs young, $\beta = -0.018$, $p = 0.886$) (Fig. 1h) and had similar learning curves (Fig. 1i). This indicated that the motivation and ability to consume rewards in VR remained grossly intact in aged mice. Notably, we observed similar running speed differences across age groups in the SM and RF tasks (see Supplementary Figs. 1c, d). Reward-triggered licking was also equivalent across age groups in the SM block phase and RF task (see Supplementary Fig. 1i). Therefore, it is unlikely that aged alternation deficits in the SM are attributable to impairments in running or licking behavior. Additionally, to exclude the possibility that vision differences impacted context discrimination, all SM mice completed a quantitative contrast sensitivity assessment prior to recording (see "Methods" section) (Supplementary Fig. 1e). The estimated contrast sensitivity thresholds of SM mice did not differ across age groups (Supplementary Fig. 1f).

Lastly, to determine the degree of variability in the aged spatial memory impairment on the SM task, we examined individual behavioral trajectories over SM task experience. While mice showed stereotyped improvement over sessions in the SM blocks within and across age groups, aged mice had more heterogeneous learning curves

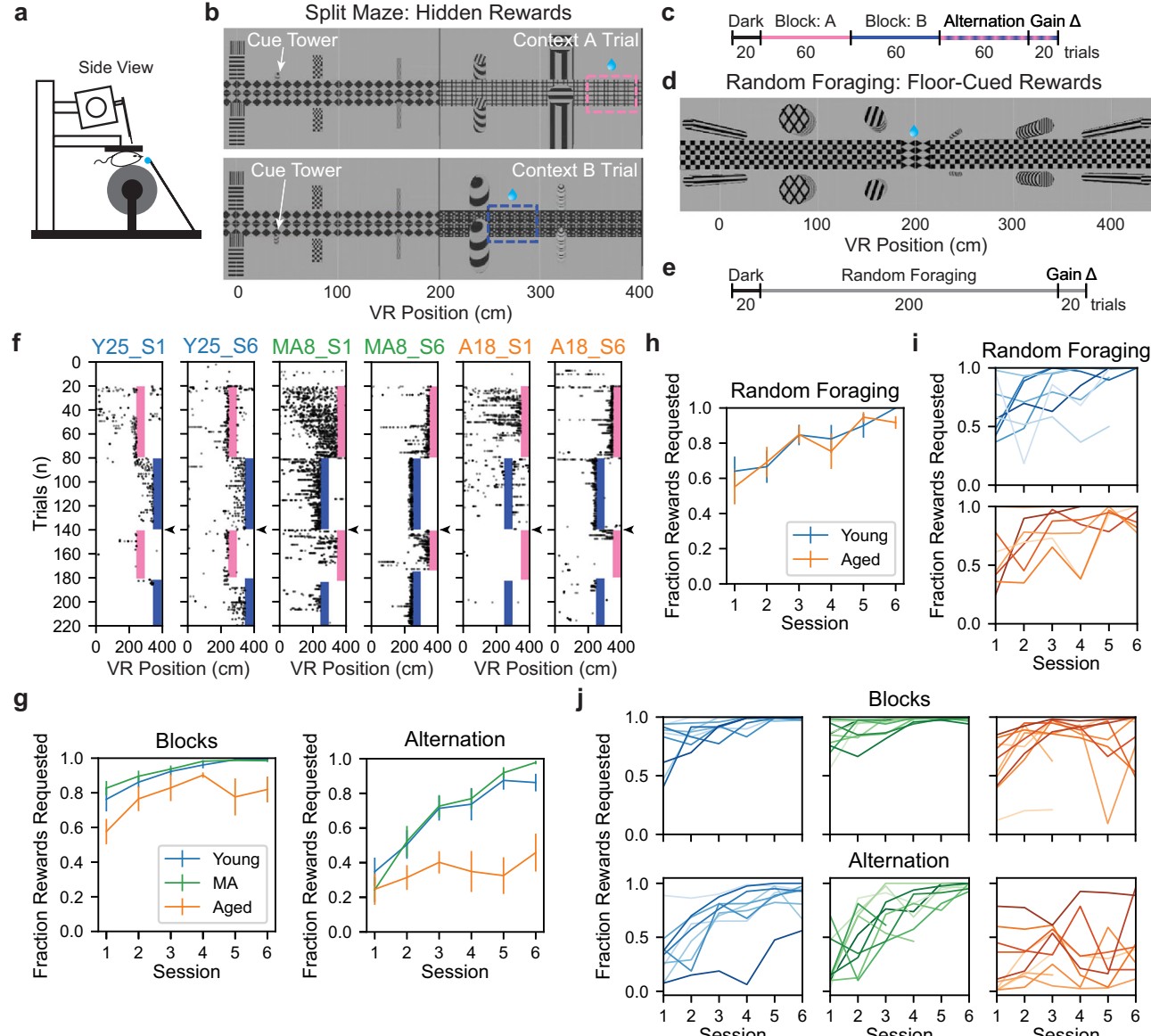

**Fig. 1 | Aged mice exhibit impaired dynamic spatial memory in a VR context discrimination task. a** Schematized acute recording setup. **b** Aerial view of Split Maze (SM) VR track on context A (top) vs B (bottom) trials (see Supplementary Fig. 1a). Dashed lines indicate hidden reward zones. Doors at 200 cm and 400 cm (obscured here) separate track halves. **c** Schematized SM session trial structure. Context was pseudo-randomized by trial in the SM alternation phase (see "Methods" section). **d** Aerial view of Random Foraging (RF) VR track (see Supplementary Fig. 1b). Diamond checkerboard indicates a randomly appearing cued reward zone. **e** Schematized RF session trial structure. **f** Non-consummatory SM lick raster plots from a representative young (Y), middle-aged (MA), and aged (A) mouse (left to right). Shading indicates reward zones (pink: context A, dark blue: context B; colors maintained throughout). Subpanel titles indicate mouse and session number (S).

Arrowheads denote when pseudorandom context alternation began. Alternation trials are context-sorted, so the trial number does not reflect trial order. **g** Average SM task performance (fraction rewards requested) over sessions during the block (top) vs alternation (bottom) phases by age group ($n = 9$ young [blue], $n = 10$ MA [green], and $n = 10$ aged [orange] mice) (age group colors maintained throughout). Vertical bars indicate SEM of the age group. **h** Plotted as in (**g**), average RF performance over session by age group ($n = 8$ young, 7 aged). **i** Individual RF task learning curves for young (top) and aged (bottom) mice. Lines correspond to mice. **j** Individual SM learning curves during blocks (top) vs alternation (bottom) for young, MA, and aged mice (left to right), excluding the two last sessions if one was terminated early (see "Methods" section). See Supplementary Fig. 1.

and failed to improve consistently over days during SM alternation (Fig. 1j). We considered to what extent variables other than age might explain these differences. Neither the location of the context A reward (reward order, 270 vs 370 cm context A reward location, $\beta = -0.046$, $p = 0.673$), nor male sex (male vs female, $\beta = -0.101$, $p = 0.3400$) predicted differences in alternation performance (Supplementary Fig. 1h). However, male sex predicted greater aged alternation performance (aged × male vs aged × female, $\beta = 0.371$, $p = 0.0150$). Together, these results indicated spatial memory deficits in aged mice on the SM task, consistent with prior work in aged rodents[7,8,40–43].

## Single-cell spatial coding correlates with heterogeneous spatial memory in aging

We next considered the question of what neural circuit differences might underlie the variability in aged spatial cognition, within and across sexes, among these genetically identical mice. To address this, we examined how spatially tuned MEC cell types in the SM task are impacted by aging[12,15,17–22]. We differentiated putative grid from non-grid spatial (NGS) cells using established methods to assess distance tuning during dark running (see "Methods" section) (see Supplementary Fig. 2a–f)[55,57,58]. Putative grid cells had peaks in spatial firing

autocorrelation on dark trials, which putative NGS cells lacked despite exhibiting spatial tuning on VR trials (see Fig. 2a–d vs Supplementary Fig. 3a–d; see also Supplementary Fig. 2b–d). Notably, we observed that the density of recorded grid and NGS cells in VR corresponded to that observed in freely moving experiments[57,59–61] and was unaffected by aging (Fig. 2d and Supplementary Figs. 2a and 3d).

Next, we examined grid and NGS cell spatial firing across SM task phases by sorting alternation trial activity by context. We observed that grid cells exhibited stronger context-dependent changes in firing field locations compared to NGS cells (Fig. 2a vs Supplementary Fig. 3a; see also Supplementary Fig. 2g–j). To further reveal patterns in the similarity of each cell's spatial firing activity across trials, we generated cross-trial correlation matrices (see "Methods" section). By the final session, many young and MA grid cell cross-trial correlation matrices displayed a checkerboard structure with (1) similar spatial firing sustained for each trial group (block A [A], block B [B], alternation A [A′], and alternation B [B′] trials); (2) similar firing on context-matched trials (e.g., A × A′); and (3) dissimilar firing on context-mismatched trials (e.g., A × B′) (Fig. 2e). Qualitatively, these features were less common among aged grid cells and among NGS cells from all age groups (Supplementary Fig. 3e). This led us to postulate that stable, context-dependent grid cell firing might constitute a neural correlate of SM context discrimination that declines with age.

To assess how age affects spatial firing stability, we computed the moving average pairwise correlation of spatial firing on neighboring trials within the SM block and alternation phases for each grid and NGS cell (see "Methods" section). Using nested LMMs to capture variance across cells and animals, we modeled the effect of the interaction of age group and session on grid and NGS cell stability in each task phase (Fig. 2f and Supplementary Fig. 3f). This revealed that spatial firing stability decreased during alternation for each cell type and age group (Grid Intercept, Block vs Alt.: $0.332 \pm 0.038$ vs $0.133 \pm 0.026$; NGS Intercept, Block vs Alt.: $0.250 \pm 0.020$ vs $0.122 \pm 0.016$) (all $p < 0.0001$). Moreover, we observed that grid firing stability in each task phase increased over sessions for young and MA mice (Session × Young, Session × MA: Block: $\beta = 0.011$, $\beta = 0.036$; Alt.: $\beta = 0.009$, $\beta = 0.027$) (all $p < 0.0001$). By contrast, among aged mice, grid spatial firing stability did not change or decreased over sessions (session × aged: Block: $\beta = 0.000$, $p = 0.881$; Alt.: $\beta = -0.010$, $p < 0.0001$) (Fig. 2f). Similar age impairments in the improvement of NGS cell spatial firing stability were observed in each task phase (Supplementary Fig. 3f).

To interrogate how age modulates context-specificity of grid and NGS firing, we calculated the ratio of mean similarity between context-matched and -mismatched trials in each task phase. A similarity ratio > 1 implies orthogonal spatial firing across contexts. We found that spatial firing context specificity, like stability, decreased during alternation (Grid Intercept, Block vs Alt. LMM: $1.193 \pm 0.052$ vs $1.003 \pm 0.012$; NGS Intercept, Block vs Alt.: $1.102 \pm 0.024$ vs $1.016 \pm 0.008$) (all $p < 0.0001$) (Fig. 2g and Supplementary Fig. 3g). Additionally, young and MA mice exhibited an increasing grid cell similarity ratio over sessions (Session × Young, Session × MA: Block: $\beta = 0.022$, $\beta = 0.064$; Alt.: $\beta = 0.012$, $\beta = 0.011$) (all $p < 0.0001$), and aging reduced this improvement over time in both task phases (Session × Aged: Block: $\beta = -0.005$, $p = 0.277$; Alt.: $\beta = 0.001$, $p = 0.398$). Consistent results were observed among NGS cells (Supplementary Fig. 3g). This suggests that aged mice also fail to orthogonalize spatial firing across contexts over time compared to their younger counterparts.

Finally, we assessed the relationship between task performance and the stability and context-specificity of MEC spatial firing across age groups, splitting sessions by task phase (block vs alternation; Fig. 2h, i and Supplementary Fig. 3h, i). The mean spatial firing stability of co-recorded grid cells related to task performance across phases in each age group (Young, $r = 0.38$; MA, $r = 0.66$; Aged, $r = 0.71$) (all $p < 0.0001$) (Fig. 2h). Among MA and aged sessions, this was also true during

alternation alone (Young, $p = 0.46$; MA, $r = 0.66$, $p < 0.0001$; Aged, $r = 0.40$, $p = 0.003$). Similarly, the mean grid cell similarity ratio correlated with performance across task phases (Young, $r = 0.47$; MA, $r = 0.47$; Aged, $r = 0.68$) (all $p < 0.0001$) and during alternation alone in each age group (Young, $r = 0.48$, $p = 0.0003$; MA, $r = 0.42$, $p = 0.0011$; Aged, $r = 0.37$, $p = 0.0068$) (Fig. 2i). Similar relationships between task performance and mean NGS cell spatial firing stability and similarity ratio were found (Supplementary Fig. 3h, i). These findings suggest that decreased performance during rapid context alternation may relate to decreased spatial stability and context-specificity relative to the block phase. Moreover, impaired stabilization of context-specific grid and NGS cell spatial firing over alternation task experience may underlie the specific deficit of aged mice in rapid context discrimination and reward recall.

### Network-wide context-aligned remapping dysfunction in aging

We next examined neural correlates of aging spatial memory performance at the population level in MEC. Given the more pronounced context-dependence of grid vs NGS cell spatial firing activity in the SM task, we focused on the grid cell population. To identify remapping events and their alignment to VR context switches, we implemented a factorized k-means algorithm to cluster trials with similar network-wide spatial firing activity, termed spatial maps (see "Methods" section)[23]. To address the considerable heterogeneity in the structure of trial-by-trial network-wide similarity matrices (Fig. 3a), we optimized the k hyperparameter for each session (see "Methods" section) (Supplementary Fig. 4a–d). We validated the k optimization procedure on RF spatial cell networks, which have been previously studied in young mice (Supplementary Fig. 4e, f)[23]. After optimization, k-means labeled maps captured the structure in SM and RF network similarity matrices (Supplementary Fig. 4f, g).

To assess the flexibility of grid networks in aging, we first computed the frequency of remaps (transitions between spatial maps) in each task phase. In both task phases, MA and aged grid networks remapped more than young counterparts (young vs MA vs aged, block: $0.0409 \pm 0.0075$ vs $0.063 \pm 0.0099$ vs $0.0792 \pm 0.0096$, Kruskal–Wallis test, $H = 16.5$, $p = 0.00025$, post-hoc Conover test, young vs MA, $p = 0.033$; young vs aged, $p = 0.0001$; MA vs aged, $p = 0.056$; alternation: $0.0758 \pm 0.0116$ vs $0.144 \pm 0.0174$ vs $0.1529 \pm 0.0218$; $H = 10.2$, $p = 0.0060$, young vs MA, $p = 0.010$; young vs aged, $p = 0.016$; MA vs aged, $p = 0.84$) (Fig. 3b). As expected, remaps were more frequent in the SM vs the RF task, in which no context changes occurred and networks in both age groups remapped similarly rarely (Supplementary Fig. 4h).

To examine the alignment of spatial maps and VR contexts transitions, we imposed context identities on labeled maps based on their occupancy of and similarity to network activity across task phases (see "Methods" section). Qualitatively, by the sixth session, young and MA grid networks exhibited transitions in map identity near VR context switches more often than aged grid networks (see right axes of Fig. 3a). We next implemented a logistic regression model to quantify age and session effects on the probability that a given trial had aligned spatial map and VR context identity (see "Methods" section) in the block (Fig. 3c) and alternation phases (Fig. 3d). While block trial map-context alignment probability exceeded chance (50% per trial) across sessions for all age groups, alternation trial alignment probability exceeded chance only for young and MA mice after session 4. Specifically, for young and MA grid networks, each session predicted increased odds of an aligned block (Young, Odds Ratio (OR) = 1.13; MA, OR = 1.24) and alternation trial (Young, OR = 1.28; MA, OR = 1.15) (all $p < 0.0001$). By contrast, map-context alignment probability on aged block (Aged, OR = 0.77, $p < 0.0001$) and alternation trials (Aged, OR = 0.95, $p = 0.043$) did not increase over sessions. In fact, the single predictor of above-chance map-context alignment for aged alternation trials was the number of consecutive trials since a pseudorandom context switch. With each additional alternation trial in the same context, the

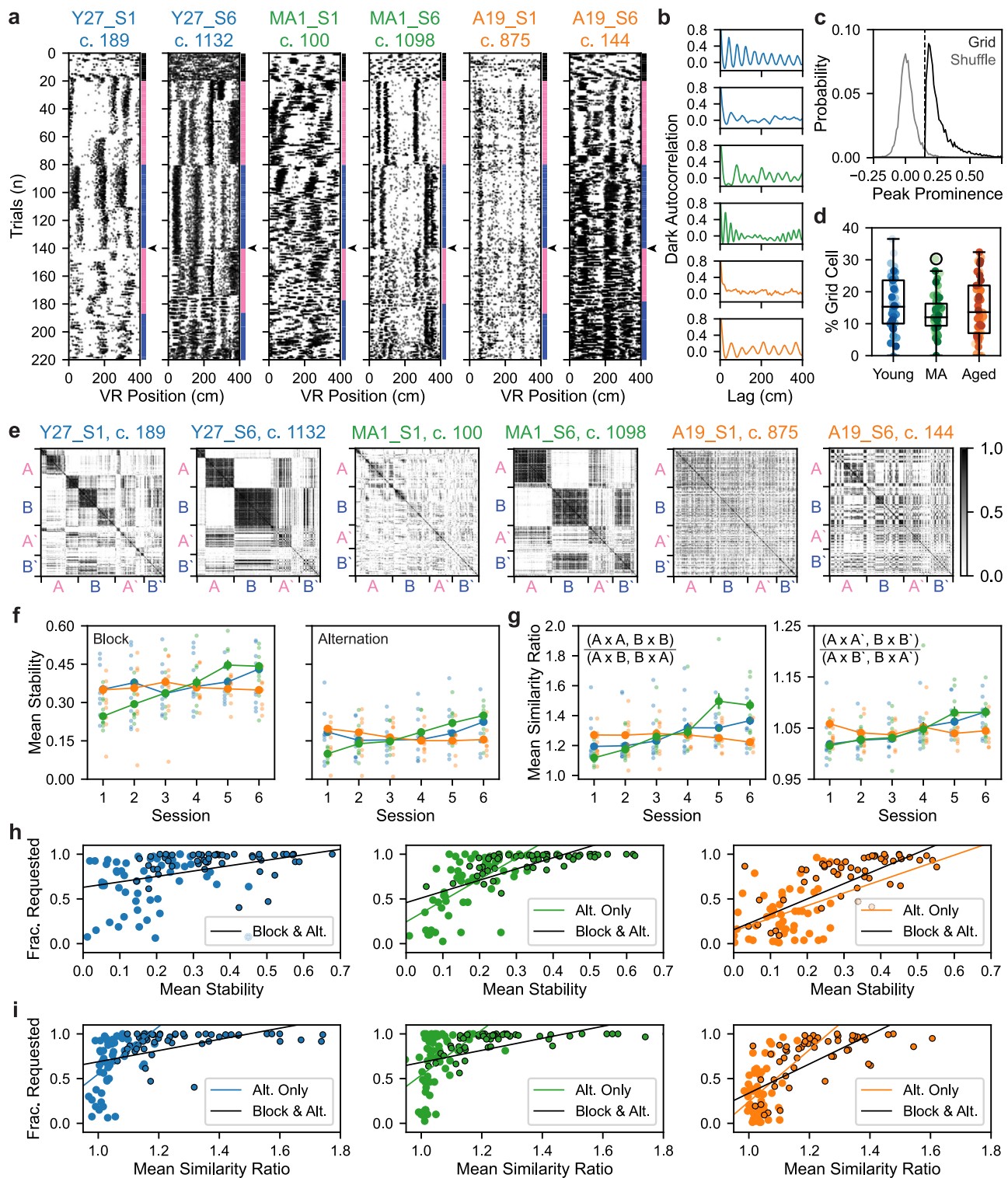

probability of aged, but not young or MA, map-context alignment increased by 9% (Aged, OR = 1.09, $p = 0.0043$; Young, OR = 0.98, $p = 0.47$; MA, OR = 0.94, $p = 0.0388$). During the block phase, trials since a switch did not predict aged trial alignment probability (Aged, OR = 1.00, $p = 0.99$). Ultimately, these findings reveal that aged grid networks exhibit more frequent but, ultimately, dysfunctional remapping that fails to improve the alignment of spatial map and context identity during the SM alternation phase over sessions. While rapid context alternation worsens alignment for all age groups, alignment improves for young and MA mice with alternation task

experience. Instead, during alternation, aged spatial maps uniquely reflect recent rather than current visual cues, consistent with diminished flexibility of visually driven remapping.

Importantly, we determined that the discreteness and coordination of remapping events were comparable across age groups (see "Methods" section) (see Supplementary Fig. 5a–d). We also confirmed that positional information remains distinct across spatial maps in aging (see "Methods" section) (see Supplementary Fig. 5e–h). Additionally, we accounted for age differences in the timing of context recognition on each trial by performing population analyses using only

**Fig. 2 | Formation of stable and orthogonal spatial activity patterns across VR contexts relates to dynamic spatial memory. a** Representative SM grid cell raster plots from sessions 1 vs 6 (left vs right subpanels) in young (Y), MA, and aged (A) mice. Dots are individual spikes. The right axis indicates trial type (dark [black], context A [pink], context B [dark blue]). Subpanel titles indicate mouse, session (S), and cell number (c.). Arrowheads indicate when context-sorted alternation trials started. **b** Dark firing rate (FR) autocorrelations of cells in (**a**) (top to bottom), revealing distance-tuning that identifies putative grid cells. **c** Probability densities of maximum peak prominence of cells' real vs the spike-shuffled dark FR auto-correlation differed significantly ($n = 6508$ model pairs, grid vs shuffle, $0.25 \pm 0.0010$ vs $0.01 \pm 0.0008$, Wilcoxon signed-rank test, $p < 0.0001$). Vertical line indicates peak threshold. Bin size was 0.01. **d** Box and whisker plot of grid cell density by age group ($n = 54$ young, 58 MA, 55 aged sessions). Dots are colored by mouse (see Supplementary Fig. 1c). Box edges, center, whiskers, and black circles indicate age group interquartile range (IQR), median, $1.5 \times$ IQR, and outliers. Age

did not alter session grid cell density (young vs MA vs aged, $16.17 \pm 1.14\%$ vs $13.38 \pm 0.79\%$ vs $15.05 \pm 1.24\%$, Kruskal–Wallis test, $H = 2.58$, $p = 0.27$). **e** Cross-trial correlation matrices of cells in (**a**) (left to right), omitting dark and gain change trials and context-sorting alternation trials. The color bar indicates the correlation value. **f** Effects of age and session on mean grid cell spatial firing stability across SM task phases fitted by linear mixed effects models (LMMs) ($n = 2441$ young, 2035 MA, 2032 aged grid cells). Large dots and vertical bars indicate the age group mean and SEM. Small dots represent LMM-fitted session averages, jittered by age group. **g** As in (**f**) for the mean grid cell spatial firing similarity ratio. **h** Task performance (fraction [frac.] of rewards requested) vs mean grid cell stability across age groups and task phases ($n = 106$ young [left], 114 MA [middle], 106 aged [right] phases). Black vs colored dot outline indicates block vs alternation phase. Lines represent linear regression fits for significant correlations. **i** As in (**h**) for mean grid cell similarity ratio. See Supplementary Figs. 2 and 3.

grid network activity in the back of the track (see "Methods" section) (see Supplementary Fig. 5i–k). Collectively, these control measures ensured that age differences in map-context alignment were not attributable to differences in the nature of remapping.

Lastly, we assessed how map-context alignment and remapping rate related to task performance across SM task phases. In each age group, map-context alignment related strongly to performance across task phases (Young: $r = 0.60$, MA: $r = 0.54$, Aged: $r = 0.58$) (all $p < 0.0001$) (Fig. 3e). Examining alternation alone, this map-context alignment by performance relationship collapsed for aged sessions (Young: $r = 0.43$, $p = 0.0025$; MA: $r = 0.41$, $p = 0.0049$; Aged: $r = 0.14$, $p = 0.39$). By contrast, remapping frequency did not relate to alternation performance in any age group (Supplementary Fig. 4i). We then used logistic regression to assess whether the occurrence of map-context alignment or a remap predicted performance, measured by the probability of a reward request, on individual block and alternation trials (Fig. 3f-g). Echoing session results, alignment predicted more than doubled request probability on block trials (Young, OR = 2.06; MA, OR = 2.39; Aged, OR = 3.88) (all $p < 0.0001$). This positive predictive relationship between alignment and performance held for young and MA, but not aged, alternation trials (Young, OR = 1.52; MA, OR = 1.95; Aged, OR = 0.69) (all $p < 0.0001$). While remaps predicted decreased request probability on MA and aged block trials (Young, $p = 0.42$; MA, OR = 0.49, $p < 0.0001$; Aged, OR = 0.45, $p < 0.0001$), they did not predict alternation request probability (Young, $p = 0.31$; MA, $p = 0.25$; Aged, $p = 0.39$). These results reveal that aging disrupts the relationship between grid network map-context alignment and SM alternation performance at the session and trial levels. Furthermore, we observed that mice with increased grid network map-context alignment over sessions also showed improved alternation performance ($r = 0.44$, $p = 0.029$) (Fig. 3h). Ultimately, these findings indicate that dysfunctional MEC remapping in aging contributes to impaired spatial memory.

### Reduced LFP power during running in middle-aged and aged mice

We next sought to uncover the impact of aging on MEC network-level oscillatory activity as measured by the local field potential (LFP). Incorporating new information into MEC spatial maps at fast time scales may require temporal coordination of neuronal activity, facilitated by theta frequency (6–12 Hz) oscillations in the LFP[62,63]. Additionally, gamma frequency (slow: 20–50 Hz; fast: 50–110 Hz) oscillations in MEC LFP are critical for communication with the HPC and support spatial learning[64].

We observed qualitative decreases in LFP power in theta and gamma frequency bands in MA and aged sessions compared to young ones (Fig. 4a). This was also apparent in the mean power spectral density, pooled across sessions in each age group (Fig. 4b). To quantitatively compare the power of theta and gamma rhythms across age

groups, we first controlled for observed differences in running speed distributions across age groups (see Supplementary Fig. 1c, d), as theta dynamics in MEC are influenced by running speed and acceleration[65,66]. In particular, we sampled MEC LFP power in each session from both tasks by in a speed range (20–40 cm/s) that collapsed running speed differences across age groups ($n = 98$ young, 58 MA, 97 aged sessions, mean running speed, young vs MA vs aged sessions, $29.93 \pm 0.12$ vs $29.6 \pm 0.13$ vs $29.39 \pm 0.22$, Kruskal–Wallis test, $H = 2.85$, $p = 0.24$; peak running speed, $40.0 \pm 0.0$ vs $40.0 \pm 0.0$, $39.86 \pm 0.07$, $H = 3.05$, $p = 0.22$). Indeed, starting in middle age, mean theta (young vs MA vs aged power [$\mu V^2$/Hz], $43.66 \pm 3.56$ vs $30.41 \pm 3.94$ vs $27.29 \pm 2.40$, Kruskal–Wallis test, $H = 14.29$, p = 0.0008, post-hoc Conover test, young vs MA, $p = 0.0086$, young vs aged, $p = 0.0012$), slow gamma ($1.46 \pm 0.09$; $0.94 \pm 0.08$; $0.92 \pm 0.06$, $H = 21.63$, $p < 0.0001$, young vs MA, $p = 0.0004$, young vs aged, $p < 0.0001$), and fast gamma power ($0.57 \pm 0.04$ vs $0.39 \pm 0.05$ vs $0.35 \pm 0.02$, $H = 23.44$, $p < 0.0001$; young vs MA, $p = 0.0002$, young vs aged, $p < 0.0001$) decreased (Fig. 4c). These results raise the possibility that the rapid incorporation of information into spatial maps supported by theta rhythm, as well as the coordination of spatial coding across the MEC-HPC circuit by gamma rhythms, may be impaired over healthy aging.

### Aged spatial coding instability in an invariant VR environment
Given the dysfunction observed in aged single-cell and network-level spatial coding in the dynamic SM task, we next examined the stability of aged spatial coding in the invariant RF task. As some RF sessions lacked dark trials, we considered all spatial cells (grid and NGS) together (see "Methods" section) (Fig. 5a–d). Consistent with prior works, spatial cells in young mice exhibited stable spatial firing patterns across neighboring RF trials (Fig. 5a), producing smooth, periodic trial-averaged spatial tuning curves[23,67]. By contrast, in aged mice, we frequently observed spatial cells with drift in firing field locations across neighboring trials, resulting in less smooth tuning curves. Consistent with the SM task, spatial coding by the aged MEC was quantifiably degraded during RF sessions. Averaging across co-recorded spatial cells, we observed decreased spatial firing coherence (young vs aged sessions, $0.7425 \pm 0.0108$ vs $0.6480 \pm 0.0136$, Wilcoxon rank sum test, $p < 0.0001$) (Fig. 5e); spatial information score ($0.0928 \pm 0.0081$ vs $0.0636 \pm 0.0056$, $p = 0.0032$) (Fig. 5f); and spatial coding stability ($0.1314 \pm 0.0087$ vs $0.0895 \pm 0.0075$, $p = 0.0003$) (Fig. 5g) with age. As expected given the relationship between tuning curve smoothness and consistent firing field locations, spatial firing coherence and stability were collinear (Fig. 5h). These findings demonstrate that aged spatial firing quality is impaired in the unchanging RF environment, reflected in spatial firing instability.

Since MEC neurons project to multiple brain regions that process spatial information[67], we next considered to what extent network-level positional information output by MEC would be degraded by spatial cell instability in aging. We fit circular-linear regression models, termed

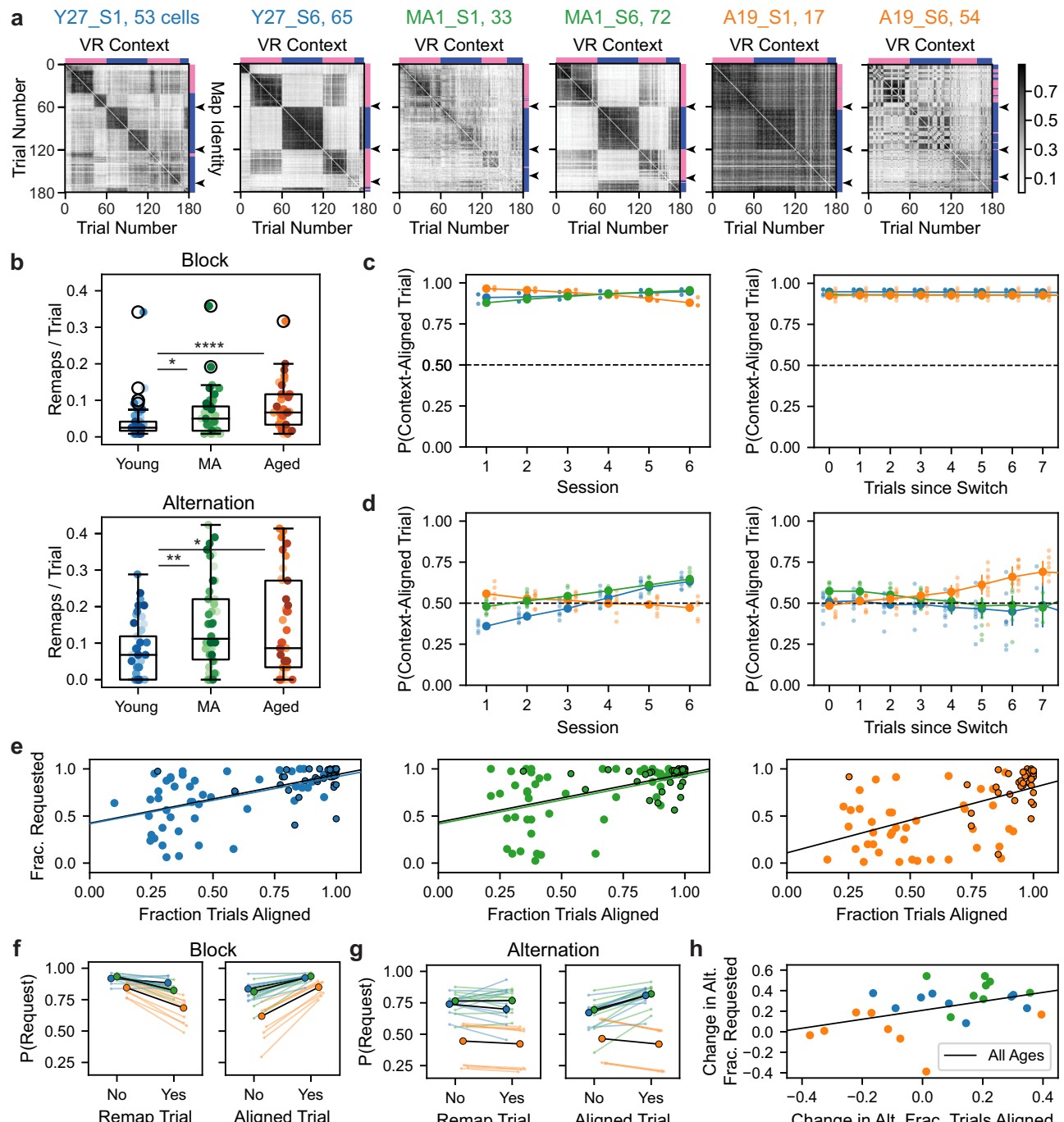

**Fig. 3 | Grid network remapping is more frequent but less aligned to context in aged mice. a** SM grid network trial-by-trial similarity matrices (same sessions as Fig. 2a) with dark and gain trials omitted and alternation trials sorted by context (top axis: context A [pink] vs B [dark blue]). These compare with k-means map identities by trial (right axis) (see "Methods" section). Arrowheads denote expected map identity transitions. Subpanel titles indicate mouse, session (S), and network grid cell count. Color bar indicates correlation value. **b** Plotted as in Fig. 2d, remapping frequency across age group ($n = 47$ young, 46 MA, and 41 aged sessions) in the block vs alternation phases (left vs right) (Kruskal–Wallis tests followed by post-hoc Conover tests). **c** Effect of age and session (left) or trials since a context switch (right) on the probability (P) of aligned map identity and VR context on block trials (context-aligned trial), modeled by logistic regression ($n = 15902$ trials, pseudo $R^2 = 0.02398$, LLR $p < 0.0001$). Large dots and vertical bars indicate age group fitted mean and SEM probability. Small dots represent fitted session averages, jittered by age group. Dotted horizontal lines indicate chance level alignment.

X-values with data from at least three mice per group were plotted. **d** Plotted as in (**c**) for a logistic regression model of probability of context alignment on alternation trials ($n = 8079$ trials, pseudo $R^2 = 0.0853$, LLR $p < 0.0001$). **e** Plotted as in Fig. 2h, task performance (fraction [frac.] rewards requested) vs fraction of context-aligned trials across age groups ($n = 94$ young, 92 MA, 82 aged sessions) and task phases. **f** Effect of age and remapping (left) or spatial map-context alignment (right) on probability of block reward requests modeled by logistic regression ($n = 15902$ trials, pseudo $R^2 = 0.1420$, LLR $p < 0.0001$). Bold dots and lines indicate age group fitted mean probability. Pale dots and lines indicate animal mean fitted probability, colored and jittered by age group. **g** As in (**f**) for alternation request probability ($n = 8079$ trials, pseudo $R^2 = 0.2098$, LLR $p < 0.0001$). **h** Change (mean last−first three sessions) in alternation (alt.) performance vs fraction of context-aligned trials ($n = 8$ young, 9 MA, and 9 aged mice). Line indicates linear regression fit. See Supplementary Figs. 4 and 5.

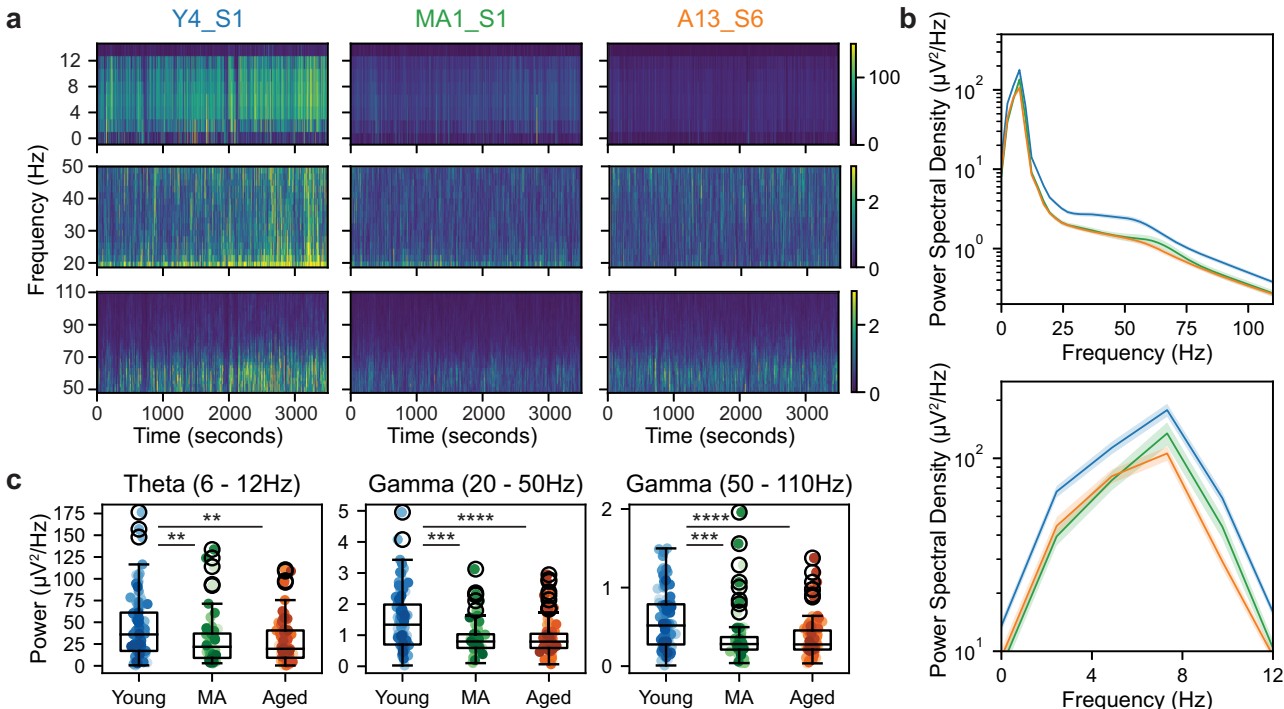

**Fig. 4 | Theta, slow and fast gamma power during running is diminished in aged mice. a** Example spectrograms in the theta (top row; 6–12 Hz), slow gamma (middle; 20–50 Hz), and fast gamma (bottom; 50–110 Hz) frequency bands for representative young, MA, and aged mice (left to right columns) for 3500 s of recording. Color coded for minimum (blue) and maximum (yellow) values. **b** Mean power spectral density across age groups (top: all frequency bands in (**a**); bottom: theta frequency range) with SEM shaded. **c** Box and whisker plot, plotted as in Fig. 2d, of mean LFP power in the theta, slow gamma, and fast gamma (left to right) frequency bands during running across age groups in both tasks (*n* = 98 young, 58 MA, 97 aged sessions) (Kruskal–Wallis tests followed by post-hoc Conover tests).

decoders, to estimate animal position from spatial cell network activity (see "Methods" section) (Fig. 5i–k). We found that decoder performance decreased when trained and tested on spatial cell activity from aged compared to young sessions (young vs aged, 0.6271 ± 0.0406 vs 0.4683 ± 0.0416, Wilcoxon rank sum test, *p* = 0.0194) (Fig. 5i; see also Supplementary Fig. 5e). Sample decodes on single trials revealed the greater number and severity of position estimation errors for aged compared to young sessions (Fig. 5j). Moreover, session decoder score was highly correlated with mean spatial cell spatial coherence (*r* = 0.66, *p* < 0.0001) (Fig. 5k), suggesting that degraded spatial coding by aged spatial cells is associated with degraded positional information output from the MEC.

### Network-wide increased speed gain and speed-tuning instability in aging

Given the degradation of position coding by the aged MEC, we next considered how the coding of other navigational variables, in particular speed, might also change. To address this, we identified positively and negatively modulated speed-tuned (+ vs − speed cells) (see "Methods" section) (Fig. 6a–c)[20,57]. The density of speed-tuned cells was unchanged by aging (see Supplementary Fig. 6). Next, we compared the gain, or sensitivity of, speed cells to changes in animal speed (see "Methods" section). In both + and − speed cells, we observed more gain to speed in aged vs young sessions (+ speed cells: young vs MA vs aged, 0.0445 ± 0.0015 vs 0.0508 ± 0.0027 vs 0.0556 ± 0.0026, Kruskal–Wallis test, *H* = 15.15, *p* = 0.0005; post-hoc Conover test, young vs MA, *p* = 0.1111, young vs aged, *p* = 0.0003; − speed cells: −0.0366 ± 0.0024 vs −0.0432 ± 0.004 vs −0.05 ± 0.0026, *H* = 24.33, *p* < 0.0001; young vs MA, *p* = 0.0816, young vs aged, *p* < 0.0001) (Fig. 6d). Trending differences between young and MA sessions imply that this change, like decreased LFP power, begins earlier than 21 months of age. In addition to altered speed tuning gain, +speed cells

from aged sessions demonstrated decreased speed tuning stability over trials (0.1905 ± 0.0063 vs 0.2143 ± 0.0061 vs 0.1693 ± 0.0047, Kruskal–Wallis test, *H* = 29.28, *p* < 0.0001; post-hoc Conover test, young vs MA, *p* = 0.0008, young vs aged, *p* = 0.0130) (Fig. 6e) (see "Methods" section). Moreover, speed tuning stability improved over SM sessions for young but not aged speed cells, in a manner related to SM task performance (Supplementary Fig. 6h–j) and echoing the unstable spatial coding of aged SM grid and NGS cells (Fig. 2f, h and Supplementary Fig. 3f, h). Taken together, these results uncovered increased speed tuning sensitivity and instability in aging that may be implicated in spatial map instability.

MEC speed coding is also supported by local interneurons (INs) and conjunctive speed-tuned grid cells[17,68,69]. To interrogate aged speed coding in these populations, we separated putative fast-spiking INs from excitatory cells (see "Methods" section) (Supplementary Fig. 6a–c), noting an increase in the density of recorded INs and speed-tuned INs in MA and aged vs young mice (Supplementary Fig. 6d, e). Consistent with our observations in + speed cells, we also observed increased speed gain and speed-tuning instability in positively modulated speed-tuned INs and conjunctive speed-tuned grid cells (Supplementary Fig. 6f, g). Thus, aging produced aligned changes in speed coding across cell types.

### Grid scale compression in aging

Speed gain and grid scale change proportionally when environmental dimensions are manipulated[30], consistent with a continuous attractor network model framework of entorhinal activity[70,71]. This suggested that grid scale compression could co-occur with increased speed gain in aging. To estimate grid scale (i.e., the distance between grid firing nodes) in 1D VR, we identified the location of the first peak in each grid cell's spatial firing autocorrelation on dark trials (see "Methods" section)[55]. In young and middle-aged sessions, we observed an increase

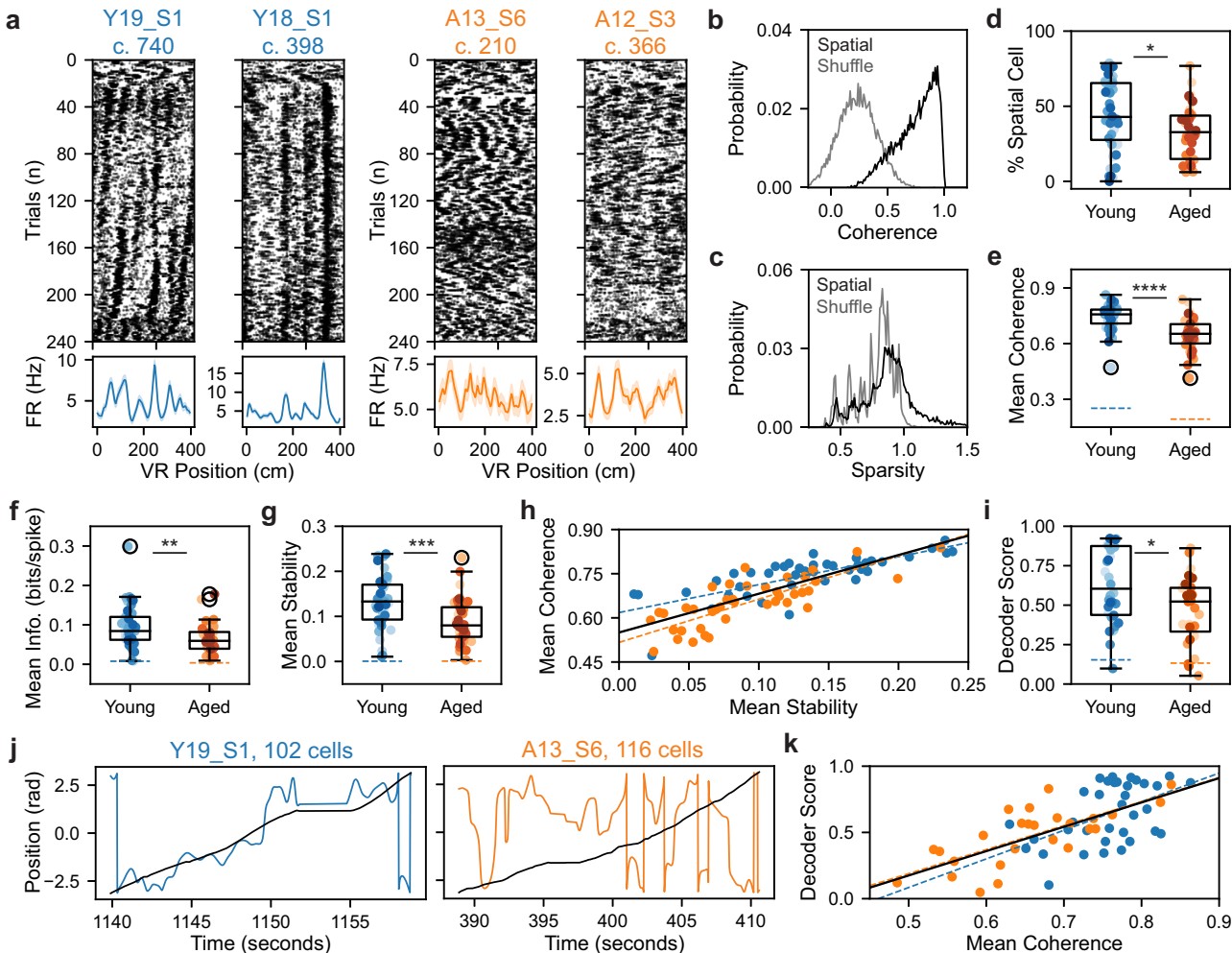

**Fig. 5 | Aged mice in stable VR environments exhibit spatial cell instability that impairs position decoding. a** Representative RF spatial cell raster plots (top) (plotted as in Fig. 2a) and trial-averaged spatial tuning curves (bottom) from young vs aged mice (left vs right). Shading indicates firing rate (FR) SEM. The first and last 20 trials were in the dark and gain-manipulated, respectively. **b** Probability density of spatial cell vs shuffle spatial firing coherence. Spatial cells' spatial firing coherence exceeded that of shuffled activity ($n = 8476$ model pairs, spatial cell vs shuffle, $0.7391 \pm 0.0019$ vs $0.2306 \pm 0.0018$, Wilcoxon signed-rank test, $p < 0.0001$). Bin size was 0.01. **c** As in (**b**) for spatial cell spatial firing sparsity, which differed from that of shuffled activity ($0.9154 \pm 0.0037$ vs $0.7716 \pm 0.0016$, $p < 0.0001$). Bin size was 0.01. **d** Plotted as in Fig. 2d, RF spatial cell density ($n = 44$ young, 42 aged sessions). Spatial cell density decreased with age (% spatial cells, young vs aged, $44.29 \pm 3.52$ vs $32.37 \pm 2.79$, Wilcoxon rank sum test, $p = 0.0136$). Dot colors indicate mouse (see Supplementary Fig. 1d). **e** Plotted as in (**d**), mean spatial cell coherence

decreased among aged sessions ($n = 43$ young, 42 aged sessions with spatial cells) (Wilcoxon rank sum test). **f** Plotted as in (**d**), for spatial cell spatial information (info) ($n = 43$ young, 42 aged sessions) (Wilcoxon rank sum test). **g** Plotted as in (**d**), for spatial cell within-map stability ($n = 43$ young, 42 aged sessions) (Wilcoxon rank sum test). **h** Session mean spatial firing coherence related to stability across (black line: $r = 0.80$) and within age groups (dashed lines: young, $r = 0.76$; aged, $r = 0.80$) (all $p < 0.0001$). **i** Plotted as in (**d**), position decoder performance for models trained and tested on aged vs young spatial cells (see "Methods" section) ($n = 34$ young, 27 aged sessions with optimal $k = 2$) (Wilcoxon rank sum test). **j** Example decoder performance on single trials from young vs aged sessions (left vs right), corresponding to sessions in (**a**). Lines indicate VR vs decoded position (black vs colored). **k** Plotted as in (**h**), decoder score related to mean spatial cell coherence across ($r = 0.66$, $p < 0.0001$) and within age groups (young, $r = 0.51$, $p = 0.0018$; aged, $r = 0.73$, $p < 0.0001$).

in grid scale from dorsal to ventral MEC, consistent with prior work (Fig. 6f)[55]. However, aged grid cell autocorrelation peaks were commonly closer together, indicating smaller grid spacing, with weaker gradients from dorsal to ventral recording sites.

Consistent with observations of grid scale in the autocorrelations (Fig. 6f), comparing grid scale distributions across age groups revealed a peak corresponding to smaller aged grid scales (Fig. 6g) and a leftward shift of grid scale cumulative density for aged and MA vs young cells (two-sided Kolmogorov–Smirnov test, aged vs young, $D = 0.12$, $p < 0.0001$; MA vs young, $D = 0.085$, $p < 0.0001$; aged vs MA, $D = 0.076$, $p < 0.0001$) (Fig. 6h). To control for the relationship between grid scale and dorsal-ventral location in MEC[12,17,72], we also compared the cumulative density of estimated recording depths along the probe axis across age groups (see "Methods" section). There were significant

differences in depth sampling across age groups (aged vs young, $D = 0.057$, $p = 0.0014$; MA vs young, $D = 0.14$, $p < 0.0001$; aged vs MA, $D = 0.18$, $p < 0.0001$) (Fig. 6h), so we fit the linear relationship between grid cell depth and scale within each age group via linear regression (Fig. 6i). Indicative of grid scale compression, the slope of grid cell depth and scale increased 78% ($m = 0.016$ vs $m = 0.009$), and the y-intercept decreased 35% among aged vs young mice ($b = 47.3$ vs $b = 72.9$). Moreover, we used LMMs to model grid scale as a function of age group and unit depth, accounting for cell and animal variance (see "Methods" section). Accounting for the effect of unit depth ($\beta = 0.009$, $p < 0.0001$), being aged predicted decreased grid scale (aged vs young, $\beta = -49.532$, $p < 0.0001$) and compressed grid scale gradients (depth × aged, $\beta = 0.006$, $p = 0.045$) (Fig. 6j). These results raise the possibility that grid scale changes in the aged MEC may influence

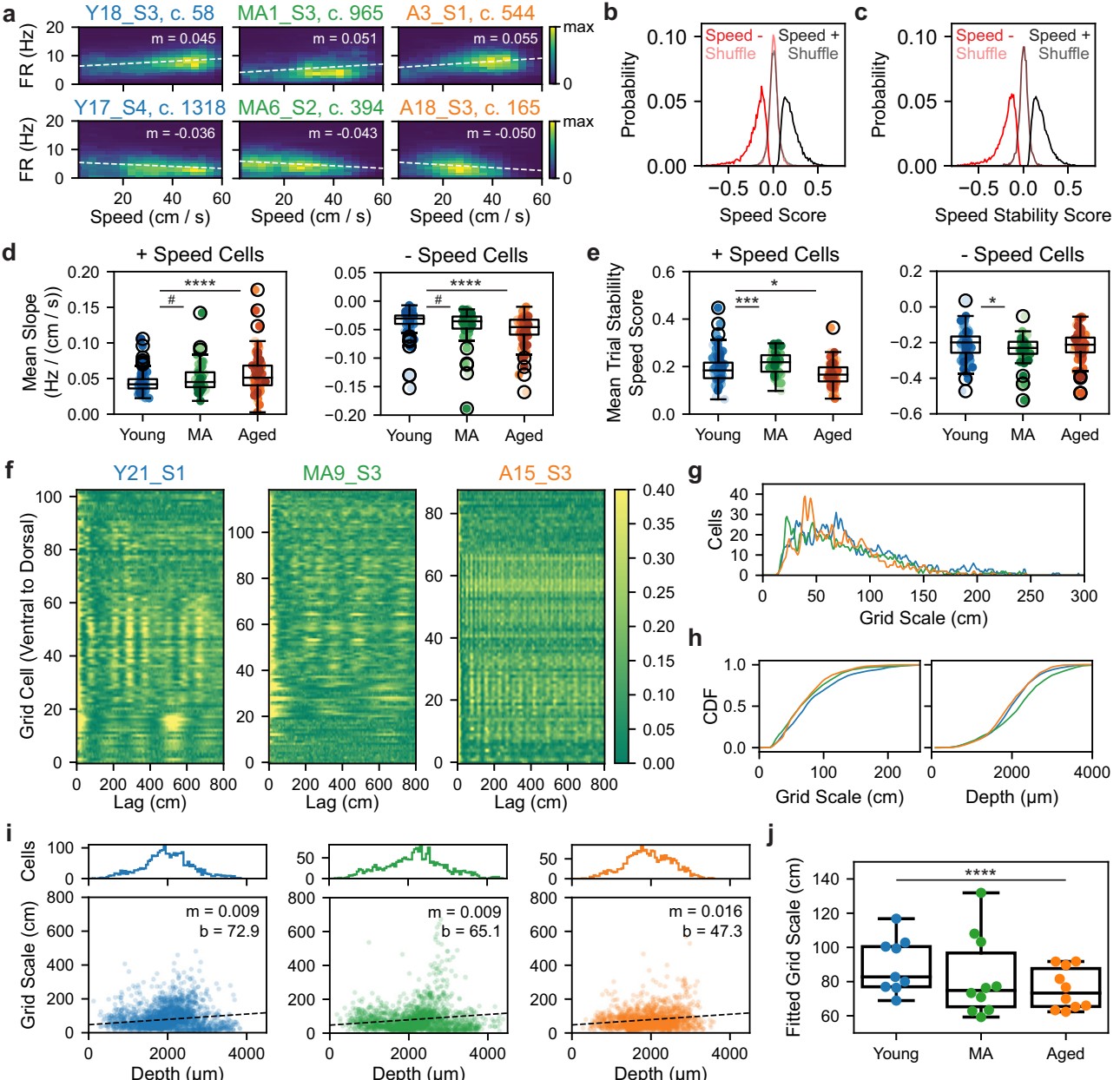

**Fig. 6 | Increased speed gain and speed-tuning instability accompany grid scale compression in aging. a** Double histograms of speed and firing rate (FR) for + (top) and − (bottom) speed cells across age groups. Dashed lines indicate linear regressions ($m$ = FR − speed slope). The color bar indicates occupancy in 0.02 s bins. **b** Probability density of + (black) and − (red) speed cells' speed scores vs shuffle, which differed ($n$ = 11528 model pairs, +speed cell vs shuffle, 0.1928 ± 0.0009 vs 0.0003 ± 0.0006, Wilcoxon signed-rank test, $p$ < 0.0001; $n$ = 6548 model pairs, − speed cell vs shuffle, −0.2010 ± 0.0015 vs 0.0002 ± 0.0006, $p$ < 0.0001. Bin size was 0.01. **c** Plotted as in (**b**) for speed stability scores vs shuffle, which differed (+ speed cell vs shuffle, 0.1936 ± 0.0009 vs −0.0023 ± 0.0006, Wilcoxon signed-rank test, $p$ < 0.0001; - speed cell vs shuffle, −0.2024 ± 0.0015 vs -0.0019 ± 0.0007, $p$ < 0.0001). **d** Plotted as in Fig. 2d, mean FR-speed slope of + (left) and − (right) speed cells across age groups ($n$ = 96 young, 58 MA, 96 aged sessions with + speed cells; $n$ = 96 young, 58 MA, 94 aged sessions with −speed cells) (Kruskal−Wallis tests followed by post-hoc Conover tests). **e** As

in (**d**) for the mean trial stability speed score. −Speed cell trial stability was not age-modulated (−0.2125 ± 0.0081 vs −0.2393 ± 0.0098 vs −0.2165 ± 0.0082, Kruskal−Wallis test, $H$ = 6.41, $p$ = 0.040; post-hoc Conover test, young vs aged, $p$ = 0.59). **f** Heatmaps of grid network dark FR autocorrelation across age groups with cells sorted by depth from the brain surface (ventral [bottom] to dorsal [top]). The color bar indicates autocorrelation value. **g** Histogram of grid scales by age group ($n$ = 2441 young, 2035 MA, and 2032 aged cells). Bin size was 1 cm. **h** Cumulative density of grid scales (left) and depths (right) by age group (two-sided Kolmogorov−Smirnov tests). **i** Histograms of grid depths (top) and scatter of grid scale and depth (bottom) by age group. Line indicates linear regression fit ($m$ = slope, $b$ = y-intercept). **j** Plotted as in (**d**), LMM-fitted grid scale across age groups ($n$ = 2441 young, 2035 MA, 2032 aged grid cells), indicating decreased depth-adjusted grid scale with age. Dots indicate LMM-fitted animal averages. See Supplementary Fig. 6.

the scale or stability of place fields in the aged HPC[40–43], given their known interdependence in young animals[61]. Ultimately, altered long-term place map stability might further contribute to impaired spatial learning and memory in aging via these changes in entorhinal inputs.

## Transcriptomic correlates of spatial coding dysfunction in aged MEC neurons

To investigate the molecular and cellular correlates of the MEC spatial coding dysfunction uncovered in healthy aging, we performed bulk RNA sequencing of MEC neuronal nuclei isolated from the

hemispheres of mice that completed the RF (Fig. 7a and Supplementary Fig. 7a) and SM tasks (Supplementary Fig. 8) (see "Methods" section). By comparing neuronal gene expression across RF age groups, we identified 458 genes that were differentially expressed (DEGs) with age in MEC (see Supplementary Data 2). The expression of 172 genes increased while the expression of 286 genes decreased with age in MEC (absolute fold change [FC] ≥ 0.5, mean expression value ≥ 25, and adjusted p-value [p-adjusted] ≤ 0.05) (Fig. 7b). A heatmap of the top 100 DEGs with age revealed consistent differential gene expression across mice within each age group (Supplementary Fig. 7b). To classify the role of aging DEGs in biological processes, we performed gene ontology (GO) and gene set enrichment analysis (GSEA) (see "Methods" section). We found that pathways regulating synaptic transmission, cell communication, and cell signaling were significantly enriched among aging DEGs (Fig. 7c). Furthermore, the two top annotated gene sets among aging DEGs were pathways related to reactive oxygen species and oxidative phosphorylation (Fig. 7d).

To examine connections between transcriptomic and functional changes in the aged MEC, we assessed the correlation of gene expression across this dataset with the LMM-fitted animal mean of each age-modulated neural feature in RF sessions (see "Methods" section) (Fig. 7e). Of these parameters, mean spatial firing coherence related most strongly to aging transcriptional changes ($r = 0.64$, $p = 0.01394$). In particular, the expression of 290 genes related negatively to spatial firing coherence across age groups, while 26 genes had positively correlated expression (Fig. 7f, g) (p-adjusted ≤ 0.10). A subset of the negative (57/290) and positive (4/26) coherence-correlated genes were aging DEGs. Next, we applied GO enrichment analysis to the modules of positively and negatively coherence-correlated genes (positive vs negative correlation modules) (Fig. 7h). While we found no pathways significantly enriched among positive correlation module genes, pathways related to pre- and post-synaptic translation were highly enriched in the negative correlation gene module. Supporting this, pre-synapse, post-synapse, and ribosomal cellular components were also enriched among this module (Supplementary Fig. 7c). Taken together, these results raise the possibility that genes required for synaptic translation are up-regulated in response to aged MEC spatial coding dysfunction.

To determine if spatial coding-associated transcriptomic changes affect particular MEC layers or neuronal subtypes, we employed an snRNA-seq approach in MECs from a pair of unilaterally recorded young and aged mice. We performed UMAP dimensionality reduction on snRNA-seq expression data and annotated the resulting clusters by superimposing the expression of known MEC layer and cell type marker genes (see "Methods" section) (Fig. 7i and Supplementary Fig. 7d, e). We observed a striking enrichment of positive correlation module gene expression in excitatory neuron clusters (Fig. 7j, k), particularly in Layer V/VI neurons (Supplementary Fig. 7j). Conversely, expression of negative correlation module genes was specific to INs (Fig. 7l, m) and enriched among parvalbumin-expressing (PV+) INs (Supplementary Fig. 7j). Subtler but similar enrichment patterns of genes up- and down-regulated with age were observed across clusters (Supplementary Figs. 7f–i). These findings converge to suggest that loss of transcriptomic and spatial coding function co-occur in aged MEC excitatory cells, including grid and speed cells, in a manner coordinated with the up-regulation of spatial coding-related genes in MEC INs. Further supporting this, MEC excitatory, but not IN, marker gene expression decreased with age (Supplementary Fig. 7k). Consistent with decreased MEC spatial firing coherence in aged mice (see Fig. 5d), expression of positive and negative coherence-correlated gene modules decreased and increased with age, respectively (Supplementary Fig. 7l). As aged MEC INs exhibit increased expression of synaptic translation genes in relation to spatial coding, these findings are most consistent with a protective response to counteract excitatory cell dysfunction and spatial memory deficits.

Finally, to validate our correlative approach to identifying molecular and cellular mediators of the functional decline of the aged MEC, we focused on the transcriptomic hit *Hapln4*, which was differentially expressed with age and coherence correlated (see Fig. 7b, g). As *Hapln4* encodes a component of perineuronal nets (PNNs), we assessed the colocalization of PNNs and PV+ INs that often produce PNNs in the RF mouse MECs contralateral to those used for RNAseq (Fig. 8a) (see "Methods" section)[73]. While the density of MEC PV+ INs was unaffected by age, PNNs were twice as abundant in the aged vs young MEC (Fig. 8b). Notably, the increased aged MEC PNN density was cell-type specific, driven by PNNs not colocalized with PV+ INs (Fig. 8c), which are rarer in young mice[73]. Moreover, the density of this differentially located PNN population related to *Hapln4* expression and mean spatial firing coherence across mice (Fig. 8d). In addition to corroborating the relationship observed between *Hapln4* expression and spatial firing coherence, these results nominate disrupted PNN homeostasis as a putative mediator of spatial coding decline in the aged MEC.

## Discussion

Here, we combined VR behavior tasks with in vivo electrophysiology to demonstrate that aging results in spatial cognitive inflexibility at the single-cell and network levels in the MEC, contributing to a specific impairment in rapid context discrimination. Among individual spatially tuned neurons, aging impaired the formation of stable, context-specific firing patterns over experience with the novel, dual-context SM task and the maintenance of stable firing patterns in the familiar, invariant RF environment. In each age group, SM task performance correlated with the stability and context-specificity of grid cell firing patterns. At the population level, aged grid networks specifically failed to rapidly transition between spatial maps during SM context alternation, with remapping occurring more frequently but with less coupling to context changes compared to during the SM block phase and the younger grid network in either task phase. Together, these results implicate disrupted adaptation of stable, context-driven spatial firing in MEC over long and short timescales in aged spatial cognitive inflexibility. Starting in middle age, we also found changes in the power of theta and gamma; increased gain and instability of velocity coding across speed-tuned cell types; and compressed grid scale gradients along the dorsal-ventral axis in MEC. These differences constitute potential circuit-level contributors to aged spatial cell and network instability in familiar contexts. Finally, by sequencing the RNA of MEC neurons from these mice, we identified 61 putative transcriptomic mediators of MEC spatial coding dysfunction, which are differentially expressed with age, correlated with spatial firing coherence, and enriched among INs.

To explore MEC substrates of aged spatial cognitive decline, we designed the SM to elicit aging spatial navigation deficits. Rodent spatial memory is canonically measured with 2D mazes, such as the Morris Water Maze[74], Barnes Maze[7], and spontaneous or forced alternation T-Maze paradigms[75]. In the SM task, we adapted the latter maze to a linear VR environment, permitting precise control of visual environmental features during the recording of MEC neural activity. In particular, we associated reward locations with VR contexts based on evidence that MEC participates in the recognition and relay of context identity information to the HPC[76–78]. By having animals experience both SM contexts in blocks of trials before alternation each day, we could dissociate impairments in short-term reward location learning and recall. As alternation performance was never tested without preceding blocks of trials (i.e., during extinction), it is not a direct proxy of long-term spatial memory. Instead, given that block behavior improved equivalently across age groups, aged alternation deficits reflect impaired context discrimination, short-term spatial memory, and/or long-term spatial learning over recording sessions. Importantly, the SM task elicited heterogeneous short-term spatial memory

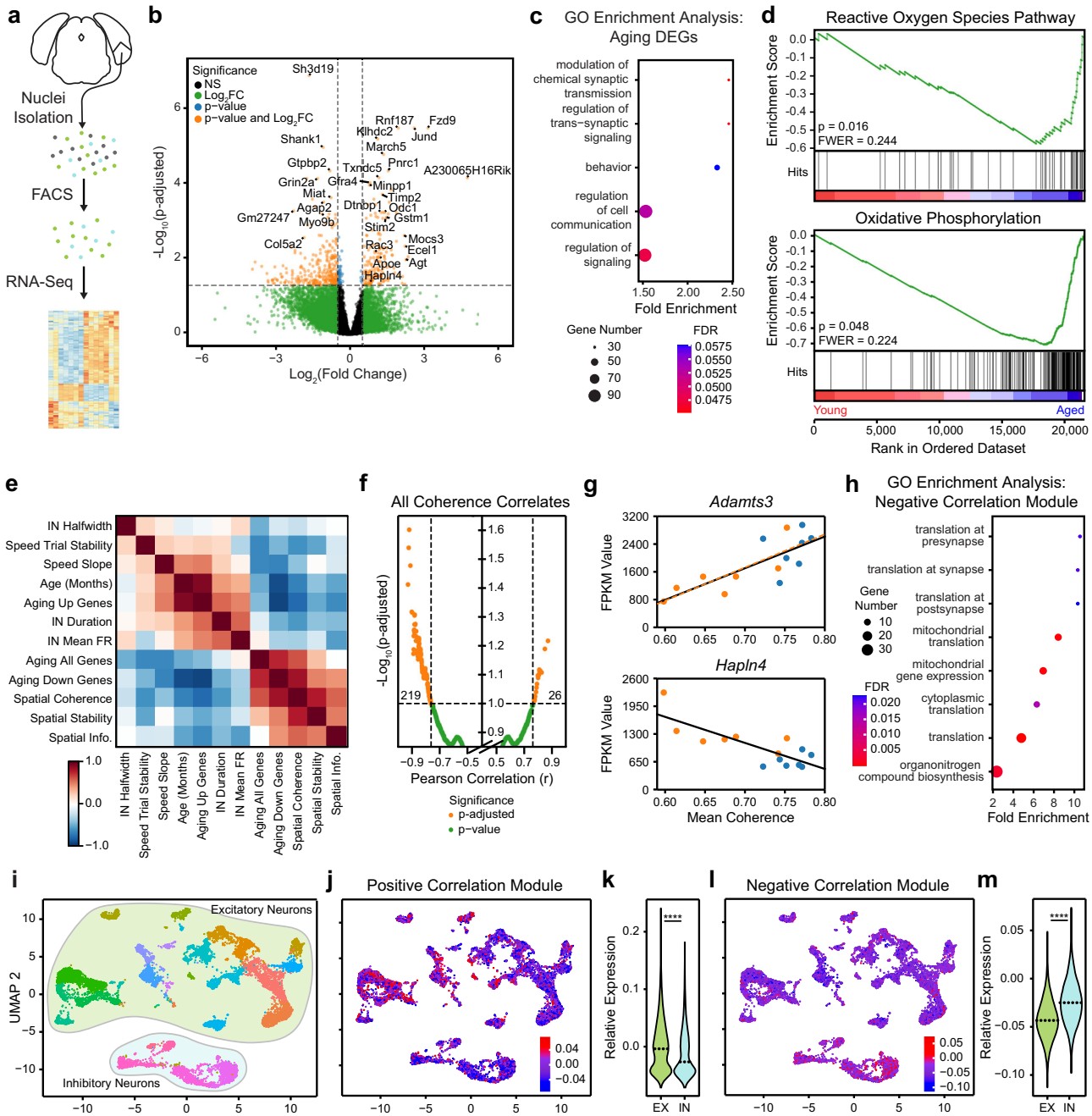

**Fig. 7 | Transcriptomic correlates of spatial coding dysfunction in the aged MEC. a** Schematized MEC neuronal nuclei RNA-seq library generation. **b** Volcano plot of differentially expressed genes (DEGs) (orange) in young vs aged MEC neuronal nuclei ($n = 7$ young, 7 aged mice) (Wald tests followed by Benjamini−Hochberg correction). **c** GO enrichment analysis of biological processes among DEGs (Fisher's Exact test followed by false discovery rate [FDR] correction). **d** Top age-correlated annotated gene sets (two-sided permutation test $p < 0.05$ followed by family-wise error rate correction [FWER] < 0.25). Enrichment score running sum is in green, and black vertical bars indicate gene set members placement (hits) among the dataset expression. Dataset genes are ranked by expression change (young to aged [left to right]). **e** Correlation among LMM-fitted animal-means of neural parameters computed for RF session cell types (INs, speed cells, and spatial cells); animal age; and DEG expression (total, age up-regulated [Up], and age down-regulated [Down]). The color bar indicates the correlation value. **f** Volcano plot of coherence correlation across the dataset genes. Totals of significantly negatively (left) and positively (right) coherence-correlated genes are noted (orange dots, linear regression $p \le 0.05$ and Benjamini−Hochberg FDR ≤

0.10). **g** Example genes that were positively (top, *Adamts3*, across age groups [black line]: $r = 0.77$, $p = 0.0012$; among aged mice [orange line]: $r = 0.81$, $p = 0.0272$) and negatively (bottom, *Hapln4*, across age groups: $r = -0.80$, $p = 0.0006$, among aged mice: $r = -0.71$, $p = 0.075$) coherence-correlated. **h** GO enrichment analysis of negatively coherence-correlated genes (Fisher's Exact test followed by FDR correction). **i** UMAP plot of MEC neuron snRNA-seq. Dot color reflects neuronal subtypes (see Supplementary Fig. 7e) with excitatory (EX) cells vs INs grouped into clouds (lime green vs light blue). **j** UMAP plot of positively coherence-correlated gene core module expression ($n = 26$ genes). The color bar indicates expression relative to the UMAP background. **k** Violin plot of positively coherence-correlated gene module relative expression in EX cells vs INs ($0.0023 \pm 0.0005$ vs $-0.0123 \pm 0.0008$, Wilcoxon rank sum test, $p < 0.0001$). Dotted lines indicate medians. **l** As in (**j**), for the negatively coherence-correlated gene module ($n = 290$ genes). **m** As in (**l**), for the negatively coherence-correlated gene module (EX vs IN, $-0.0426 \pm 0.0002$ vs $-0.0241 \pm 0.0004$, Wilcoxon rank sum test, $p < 0.0001$). See Supplementary Figs. 7 and 8.

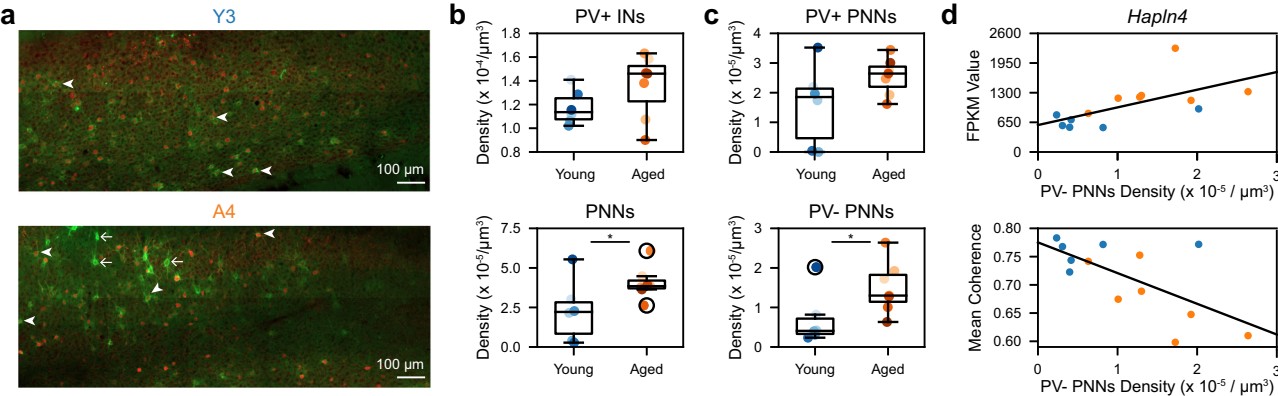

**Fig. 8 | Aging alters MEC perineuronal net density and distribution.**
**a** Representative images of the colocalization of *Wisteria floribunda* agglutinin-labeled perineuronal nets (PNNs [green]) and parvalbumin-expressing interneurons (PV+ IN [red]) in young (top) and aged (bottom) MEC after summing over Z steps (see "Methods" section). Arrowheads vs arrows denote PNNs co-localized with PV+ INs vs not. Brightness and contrast were linearly adjusted across the entire image with Fiji to ensure visibility. **b** Plotted as in Fig. 5d, PV+ IN (top) and PNN density (bottom) within the total analyzed MEC tissue volume across age groups. MEC PV IN+ density was unchanged by age ($n$ = 6 young, 7 aged mice with fixed MEC tissue; young vs aged PV IN+ density [×10$^{-4}$/μm$^3$], 1.1174 ± 0.0598 vs 1.3567 ± 0.1022, Wilcoxon rank sum test, $p$ = 0.15). MEC PNN density increased with age (young vs aged PNN density [×10$^{-5}$/μm$^3$], 2.2755 ± 0.7903 vs 4.0518 ± 0.3974,

Wilcoxon rank sum test, $p$ = 0.0455). Dots indicate animal data. **c** As in (**b**), the density of PNNs co-localized with PV+ INs (PV+ PNNs, top) vs not (PV- PNNs, bottom) in MEC across age groups ($n$ = 6 young, 7 aged mice). MEC PV+ PNN density was unchanged while PV- PNN density increased with age (young vs aged, PV+ PNN density [×10$^{-5}$/μm$^3$], 1.5765 ± 0.5534 vs 2.5510 ± 0.2349, Wilcoxon rank sum test, $p$ = 0.1985; PV- PNN density: 0.6986 ± 0.2765 vs 1.5009 ± 0.2493, $p$ = 0.0455). **d** Plotted as in Fig. 7g, the relationships between MEC PV- PNN density and *Hapln4* expression (top) and mean spatial firing coherence (bottom). Across age groups, MEC PV- PNN density related to *Hapln4* expression in the contralateral MEC ($r$ = 0.63, $p$ = 0.0213) and the animal's LMM-fitted mean spatial firing coherence ($r$ = −0.66, $p$ = 0.0142). PV+ PNN density did not relate to *Hapln4* expression ($r$ = 0.43, $p$ = 0.14) or spatial firing coherence ($r$ = −0.28, $p$ = 0.35) (not shown).

performance among aged mice (see Fig. 1j), especially across sexes (see Supplementary Fig. 1h), consistent with previous work. In humans, healthy cognitive aging occurs at highly variable rates[79–81], likely driven by a complex interaction among genetic[82], epigenetic[83], environmental, and other health and lifestyle factors[84]. Other groups have also reported increased vulnerability of female rodents[85], non-human primates[86], and humans[87] to spatial memory decline. Though this makes the statistical discernment of age effects on cognition more challenging, it creates natural experiments to identify factors that influence the rate of cognitive aging. For example, previous studies have stratified rodents into behaviorally age-impaired and -unimpaired groups that exhibited corresponding levels of hippocampal place coding dysfunction[88–91]. Similarly, studying aged individuals with remarkably preserved cognitive ability, so-called super agers, may offer insight into neuroprotective factors[92]. We observed at least one such aged mouse (A24; see darkest orange line in Fig. 1j and Supplementary Fig. 4g) in this study. This behavioral heterogeneity, across and within sexes, among genetically identical aged mice offered us a powerful opportunity to identify functional correlates of inflexible spatial cognition in MEC, including impaired context-driven changes in grid network firing patterns (see Supplementary Fig. 8a–c, p, q). Importantly, this suggests that the concordant, but more severe, MEC grid coding impairments and spatial memory deficits previously reported in mouse models of AD[49–51] arise from both transgene-mediated pathology and sex-modulated age effects that vary across mice. Dissociating such effects on in vivo circuit dysfunction in mouse models of neurodegenerative diseases, therefore, requires high-density recording approaches to sufficiently power comparisons with sex-balanced, age-matched control groups.

Since the HPC and MEC are reciprocally connected regions that cooperate to support spatial cognition, it is highly likely that aged spatial coding dysfunction in these two brain regions interact to impair spatial memory. Supporting this, we observed parallels between spatial coding impairments in aged grid cells in MEC and those previously reported in aged HPC place cells. In MEC, aged grid cells failed to stabilize context-specific firing patterns over sessions, which related to

aged SM alternation impairment (see Fig. 2f–i). Aged grid networks also exhibited increased remapping frequency (see Fig. 3b). Similarly, in HPC, aged place cells exhibit impaired experience-dependent spatial coding, such as persistent remapping over successive exposures to the same environment over days[40–42]. Aged place field stability also correlates with spatial learning[41]. Furthermore, aged rats showed HPC map mis-alignment to external cues correlated with an impairment in goal-directed navigation[43], which corresponds to the poor context alignment of aged spatial maps we observed (see Fig. 3d). Finally, we observed decreased theta power in aged MEC, adding to previous findings that theta rhythm changes in aged rodent HPC[93] and in aged humans[94–97]. Theta rhythm across the MEC-HPC network is modulated by medial septal inputs[98–100] and critically supports navigation and spatial memory[101,102]. Collectively, these findings raises the following non-mutually exclusive possibilities explaining aged spatial cognitive deficits: (1) unreliable sensory information is conveyed to the MEC-HPC network; (2) the integration of sensory information in either MEC or HPC fails, propagating similar spatial coding errors throughout the network; or (3) similar impairments arise independently in MEC and HPC through convergent molecular and cellular mechanisms. Future work co-recording HPC and MEC in aged mice is needed to further distinguish these possible ways in which grid and place coding deficits interact to impair spatial navigation.

Supporting the first possibility that MEC-HPC sensory inputs degrade, we observed increased speed gain and speed-tuning instability in aged MEC. Since the speed signal is altered across MEC positively speed-modulated speed cells, interneurons, and grid cells (see Fig. 6d, e and Supplementary Fig. 6f, g), this raises the possibility that inputs to MEC carrying velocity information are impacted by aging. In addition to impacting the vestibular and visual systems of aged animals[103], aging may specifically alter the function of the brainstem circuit recently shown to drive MEC speed cell activity, originating in the pedunculopontine tegmental nucleus (PPN)[104]. In fact, PPN neuronal density decreases with age in rats[105] and humans[106]. Examining velocity signals in the aged PPN would help to identify the source of aged velocity coding dysfunction. This is an important, open

question because continuous attractor models of grid network activity predict that proportional MEC velocity gain and grid scale changes are required for stable spatial coding and accurate path integration[70,71]. Increased speed gain and grid scale compression co-occurred in aging (see Fig. 6), but it is unclear how precisely coordinated these changes are at behavioral timescales. In addition to speed tuning instability, uncoordinated changes in speed gain and grid scale may contribute to MEC spatial coding instability in aging. Therefore, identifying circuit-level sources of degraded speed information may elucidate how spatial coding instability arises across MEC and HPC.

Providing evidence for the second possibility that MEC–HPC spatial coding deficits interact, we observed a pronounced impairment in fast gamma power (see Fig. 4c). Optogenetic perturbation of MEC fast gamma oscillations impairs spatial learning, likely via disturbed synchrony of MEC and dentate gyrus (DG)[64]. Reduced synchrony of MEC and DG oscillations during aging may interact with reduced synaptic innervation of DG and CA3 subregions by the entorhinal cortex in aging[107–109] to impair the transmission of contextual and positional information. Moreover, our findings reveal that the positional information output by aged MEC spatial cell networks is significantly degraded (see Fig. 5i). These results suggest that MEC spatial information and its transmission to HPC both become less reliable in aging. Additionally, we observed a few aged sessions with high grid network spatial map alignment to context but paradoxically impaired alternation performance, compared to almost no young or MA sessions of this type (see Fig. 3e). Therefore, poor performance in at least some aged mice may be explained by impaired transmission or utilization of MEC context information by downstream brain regions, including HPC. Finally, changes in MEC grid scale alone, as we observed (see Fig. 6f–j), might plausibly alter HPC place scale and stability[61]. Further exploring how aging changes the functional connectivity of MEC and HPC may elucidate how subregion-specific HPC place coding impairments arise[90,110].

Towards the goal of finding molecular and cellular factors linked with aging in MEC, we identified 458 genes differentially expressed with aging (DEGs) in MEC neuronal nuclei using bulk RNA sequencing in RF mice. We observed the up-regulation of genes related to oxidative phosphorylation and reactive oxygen species regulation in the aged MEC, consistent with a compensatory response to oxidative stress in normal aging[111]. Furthermore, we observed increased expression of the AD-associated genes *Apoe* and *App* with age in MEC (Fig. 7b), as well as differential expression of *Apoe* across aged sexes (Supplementary Fig. 8p). This aligns with recent work showing significant overlap between transcriptomic reprogramming in aged and AD non-human primate entorhinal cortex[112] and between AD-associated human brain gene modules and those in aged wild-type mice[113].

By examining the relationships between neuronal gene expression and in vivo MEC spatial coding features, we identified positively and negatively spatial firing coherence-related MEC gene modules. Consistent with age-mediated loss of MEC excitatory cell function, positively coherence-correlated gene expression was enriched among excitatory neurons, especially in Layer V/VI, and decreased with age, along with excitatory cell markers. Aging MEC Layer V/VI excitatory cell dysfunction might impact long-term memory formation and retrieval by impairing HPC output to neocortical networks, feedback projections from HPC to MEC Layer II/III, and subfield-specific HPC spatial coding[114,115]. Conversely, negatively coherence-correlated gene module expression was specific to MEC INs, enriched among PV+ INs, and increased with age. GO analysis revealed that this module is strongly related to synaptic translation, which is required for synaptic plasticity. In young mice, MEC PV+ INs are required for selective spatial coding[68] and receive inhibitory medial septal inputs that modulate MEC theta rhythm[98–100] and, in turn, support spatial learning and memory[101,102]. Taken together, these findings suggest two possible

transcriptomic mechanisms by which aging might impair MEC spatial coding: (1) expression of spatial coding supportive genes decreases in MEC excitatory cells while expression of genes deleterious to spatial coding increases in MEC INs; or (2) altered synaptic and neurotransmission-related gene expression in aged MEC INs occurs as a protective response to excitatory cell transcriptomic and coding dysfunction. As such, future investigations should aim to elucidate the roles of spatial coding-related genes up-regulated with age in MEC circuit function. Along with understanding the signaling pathways that modulate these genes' expression, such studies might uncover tractable molecular targets to improve aged spatial coding and memory.

Among 57 DEGs negatively correlated with spatial firing coherence, *Hapln4* is a notable gene of interest that emerged from this work. *Hapln4* encodes a secreted protein that links hyaluronans and chondroitin sulfate proteoglycans (CSPGs) to organize and stabilize perineuronal nets (PNNs) that surround INs[116–118]. Our findings concur with previous work showing that the percentage of HAPLN4+ PNNs increases with aging[119] and that aging unmasks PNN assembly impairments in *Hapln4* knock-out mice[120]. PNNs have been implicated in the control of synaptic connectivity and plasticity throughout the brain, primarily via their role in PV+IN-dependent gamma oscillations[121–123]. Disrupted PNN homeostasis has also been implicated in neuropsychiatric diseases, including AD and epilepsy, involving disrupted spatial memory[124,125]. In MEC, PNNs most often surround the cell bodies of PV+ INs and contribute critically to grid cell network stability[73,126]. Strikingly, we found that aging reproduced numerous spatial coding impairments and network activity changes caused by selectively degrading MEC PNNs in adult rats[126], including decreased spatial information (Fig. 5f); reduced ability to create stable representations of novel environments (Fig. 2) or to maintain ones of familiar environments (Fig. 5g); and diminished gamma power (Fig. 4). The strong concordance between neural phenotypes in PNN-enriched aged and PNN-depleted young MEC raises the possibility that impaired PNN homeostasis, possibly involving increased *Hapln4* expression or altered PNN distribution across MEC cell types (Fig. 8a–c), relates to spatial coding dysfunction in aged MEC. Supporting this, we found that *Cspg5*, which encodes another PNN component, was also negatively coherence-correlated and up-regulated in the aged MEC, while expression of a member of a metalloprotease family that degrades PNNs, *Adamts3*, was positively coherence-correlated. Moreover, we confirmed that aged *Hapln4* up-regulation, increased density of MEC PNNs not co-localized with PV+ INs, and spatial coding impairment are each related to one another (Figs. 7g and 8d). Ultimately, by identifying coordinated transcriptomic and spatial coding changes in aged MEC, this work motivates elucidating the role of *Hapln4* and other age-modulated, coherence-correlated genes in MEC IN synaptic plasticity, which may be a critical determinant of MEC-HPC spatial coding and spatial memory decline in aging.

## Methods

### Mice

All experimental approaches were approved by the Institutional Animal Care and Use Committee at Stanford University School of Medicine. A total of 18 (8 female, 10 male) young, 10 (5 female, 5 male) middle-aged (MA), and 17 (10 female, 7 male) aged C57Bl/6 mice (Charles River and The Jackson Laboratory) were used for bilateral neural recordings across two behavioral tasks. The young, MA, and aged groups were $3.35 \pm 0.31$, $12.63 \pm 0.09$, and $22.44 \pm 0.26$ months old (ranges: 2.30–6.64; 12.30–12.99; 21.12–24.41) at the time of the last recording. Young and MA mice were acquired from Charles River and the Jackson Laboratory, respectively, at 4–6 weeks of age. Aged mice were acquired from the Jackson Laboratory (Stock No. 000664) at 72–84 weeks old. All mice were housed in transparent cages (Innovive) with five same-sex littermates. Middle-aged and aged mice were aged in-house to 48–53 and 88–92 weeks old before surgical headbar

implantation. Mice were given an in-cage running wheel 4 weeks prior to this surgery. After headbar implantation, mice were co-housed with 1–3 same-sex implanted littermates unless separation was necessary in response to signs of aggression. After craniotomy surgery and during recording, all mice were single-housed. All mice were kept on a 12-h light-dark cycle at 21–23 °C and 30–40% humidity, with experiments conducted during the dark phase. An additional young animal underwent surgeries, behavioral training, and sham silicon probe insertion to generate neuronal gene expression data from MEC via snRNA-seq data from MEC, reflected in Fig. 7i–m and Supplementary Figs. 7d–j.

## VR setup and tasks

The VR recording setup was similar to previously published designs[23,57,127]. Head-fixed mice ran on a foam cylinder (15.2 cm diameter) fixed to rotate along one axis. Virtual environments were generated using commercial software (Unity 3D) and displayed on three 24-inch monitors around the mouse. A quadrature encoder (Yumo 1024 P/R) measured cylinder rotation, which a microcontroller (Arduino Uno) processed into motion signals to advance the VR environment. VR track gain was calibrated such that each track was 400 cm long. Upon reaching 400 cm, mice were teleported seamlessly to the track start, making the track seem circular. Each VR track featured black and white visual cues, including a patterned floor for optic flow and five pairs of landmark towers spaced 80 cm apart, each with unique dimensions and patterns (all with neutral luminance) (see Fig. 1b, f, and Supplementary Fig. 1a–b). Water rewards were delivered via a custom-built lickport, consisting of Tygon tubing attached to a metal spout in front of the mouse. In reward zones, mouse licks (requests) resulted in reward delivery. Licks broke an infrared beam positioned across the lickport, which triggered an audible solenoid valve (Cole Palmer) to pump water via a microcontroller (Arduino Uno). Failure to lick before the reward zone center resulted in an automatic reward delivery in some training and recording conditions. For each VR frame, the Unity engine recorded the mouse track position and frame time. Reward zone locations, lick times, and reward delivery times were also recorded.

With this setup, we collected behavioral and neural data in two VR tasks: the SM and the RF tasks. For the first 20 SM trials, mice ran in darkness with no visual cues. On each subsequent trial, mice encountered one of two sets of distinct visual cues (contexts A and B), each associated with one of two possible hidden reward zones (50 cm long, centered at 270 cm or 370 cm). Contexts A and B were introduced sequentially for ~60 trials each (blocks) and were then pseudorandomly alternated for another ~80 trials (alternation) via generation of a random number between (0, 2] by the Unity engine during the teleport between trials that dictated whether Context A or B cues would appear on the upcoming trial (see Fig. 1c). For the first ten trials in each block, rewards were automatically delivered if not requested. The association of reward locations with contexts was counterbalanced within age groups to control for the impact of the locations of hidden rewards. The first 200 cm (front) of the track was mostly shared across contexts, featuring a black and white diamond checkerboard floor and three pairs of rectangular prism towers, each with different sizes and patterns (see Supplementary Fig. 1a). In contexts A and B, respectively, a small cue tower was positioned immediately to the left or right of the floor at 40 cm from the start. Large black doors, which opened when the mouse was 10 cm away, obscured the second 200 cm (back) of the track, and a uniform gray landscape after 400 cm. In context A trials, the back of the track featured a floor with horizontal and vertical lines and two different cylindrical tower pairs. On context B trials, the back featured a dotted floor and two bullet-shaped tower pairs. During the last 20 alternation trials, the gain of the VR track translation to motion signals was decreased to 0.7, requiring the mouse to run 568 cm to advance the VR environment 400 cm[57]. We aimed for each SM mouse (n = 9 young [4 male, 5 female], 10 MA [5

male, 5 female], and 10 aged [5 male, 5 female]) to complete the task daily for six days. Mice A16 and MA7 showed signs of peri-craniotomy skin infection by the sixth session, so only five recordings were completed. Mouse MA4 exhibited signs of distress, including long periods of freezing, during its final recording. Behavioral data for the last two completed sessions of each of these mice were excluded. Finally, recording from A14 was terminated after the third session due to poor running and licking behavior post-craniotomy. These sessions are included in neural and behavioral datasets, since A14 completed all SM task trials. snRNA-seq MEC gene expression data from this animal are also reflected in Fig. 7i–m and Supplementary Fig. 7d–j.

In the RF task sessions, mice completed ~200 trials of running on a track with unchanging visual cues, including a checkerboard floor and five cylindrical tower pairs[23]. In addition, 50 cm reward zones appeared at a probability of $p = 0.0025$ or $0.005$ per cm, titrated to mouse performance, within the middle 300 cm of the track and at least 50 cm apart. Reward zones were marked by a diamond checkerboard pattern hovering slightly above the floor. After a successful reward request or complete reward zone traversal, the current zone disappeared, and the next became visible. In a subset of RF sessions (male mice only), 20 trials of dark running and 20 trials of gain manipulation were added to the beginning and end of the task, respectively. We also aimed to collect six sessions from each RF mouse (n = 9 young [6 male, 3 female], 7 aged [2 male, 5 female]). Hardware failures resulted in the collection of only five sessions from Y2, Y3, and Y11 and three sessions from Y16. Finally, recording from Y9 was terminated early due to poor running and licking behavior post-craniotomy. Behavioral data from these three partially completed sessions were excluded.

Before recording, a third task was administered to SM mice to estimate their contrast sensitivity. In this task, we repurposed the RF track but adjusted the reward zone opacity as mice ran 100 trials with a reward probability of $p = 0.005/cm$ (see Supplementary Fig. 1e). For the first five trials, reward zones appeared at full opacity (α = 1). For the next five trials, reward zones were invisible (α = 0). For the remaining trials, reward zone opacity was pseudorandomized (0 < α < 1). We also changed the floor to a uniform gray color and dropped the reward zone to the floor to prevent mice from recognizing low-opacity reward zones by subtle optic flow changes or the presence of a floating obstacle. To estimate contrast sensitivity, we calculated the fraction of rewards requested for binned reward zone opacities (bin size = 0.1) and fit that with the following sigmoid function using scipy (optimize.curve_fit), initialized at center = 0.1 and scale = 0.002:

$$y = \frac{1}{1 + e^{-\left(\frac{x - \text{center}}{\text{scale}}\right)}} \tag{1}$$

The center of the resulting psychometric curve was each animal's estimated contrast sensitivity threshold (see Supplementary Fig. 1f).

## In vivo survival surgeries

A mixture of oxygen and isoflurane was used to induce (4%) and maintain (0.5–2%) anesthesia. Perioperative pain control was achieved with a subcutaneous buprenorphine (0.05–0.1 mg/kg) injection post-induction. Following surgery and for three additional days after, mice were monitored and administered Baytril (10 mg/kg) and Rimadyl (5 mg/kg) subcutaneously. In the first surgery, we attached a custom-built titanium headbar, implanted a screw soldered to a gold ground pin, and made fiducial marks on the skull. These marks were made at ±3.3 mm relative to midline and 3.7 mm posterior to bregma to guide Neuropixels probe insertion. The ground pin was implanted ~2.4 mm anterior and ~1.2 mm lateral of the bregma. Metabond (Parkell S380) was used to cover any exposed skull surface and secure the ground pin and headbar. After behavioral training, we made bilateral craniotomies (~200 μm in diameter) at 3.7–3.95 mm posterior and 3.2–3.4 mm lateral to bregma in a second surgery. These were posterior and centered

to fiducial marks and exposed the transverse sinus. Bilateral duro-tomies were made central to the craniotomy sites using a bent 30-gauge needle. A plastic well was also implanted around each craniotomy and fixed with Metabond. Craniotomy sites were washed with sterile saline and sealed with a silicone elastomer (Kwik-Sil, World Precision Instruments) to conclude surgery and each recording day. After both surgical procedures, mice were administered 1 mL of saline and placed on warming blankets to recover.

## Training and handling

After headbar implantation, mice were handled for at least 5 minutes (min) twice each day. Three days after implantation, they were restricted to 0.8–1.5 mL of water per day with *ad libitum* access to food. Mice were monitored and weighed daily to maintain them at 80–85% of *ad libitum* postoperative weight. Water volume was increased within this range if signs of dehydration, such as skin tenting, were observed. Any water consumed during training or recording in VR was supplemented afterwards in the home cage to reach the total allocated amount.

After initiating water deprivation, mice were trained in phases (A–C) to run in VR and request rewards according to implicit rules. In Phase A, mice acclimated to head-fixation and moving in VR. Initially, the experimenter helped advance the cylinder forward and prevent backward motion in two daily sessions lasting 10 min each. When mice initiated forward movement independently, training time increased to 20 min, and the VR monitor was turned on, displaying the RF track. The next day, we introduced the lickport and dispensed water rewards (~1.5–2 μL per reward) automatically at each randomly appearing reward zone ($p = 0.10$/cm). With two daily 30-min training sessions, mice learned to associate solenoid clicks with reward delivery and, eventually, to request rewards by licking within cued reward zones. Gradually, we increased the goal trial number from 10 to 25 to 50 and 100, while decreasing the reward probability from $p = 0.10$ to 0.05 to 0.01 to 0.005. When mice requested >80% of rewards before automatic delivery, we switched to operant dispensation.

When mice operantly requested >80% of rewards and ran 100 trials in 30 min, training protocols diverged for the SM and RF tasks. SM task mice were next administered the contrast sensitivity assessment described above. Then, adapting the tower and floor cues from the RF track, SM mice were trained to lick in a single hidden reward zone per session (Phase B) and, then, to abruptly switch between rewarded locations during the session (Phase C). In Phase B, mice ran 110 trials and received automatic rewards in a hidden zone (switching between centers at 220 cm or 320 cm each session) for periods of 60 and then 40, 20, and 10 trials. When mice acquired the hidden reward location within 10 trials, Phase C began. The trial number doubled to 220, and the reward location switched at 110 trials. Only 10 automatic rewards were delivered at the session start and after the switch to indicate reward location. SM mice were considered ready for recording when they completed >220 trials/h and consumed >80% of rewards at both locations. After Phase A, RF task mice continued daily RF track training, lasting a maximum of one h. The goal trial number increased to 150, 200, and then 250 with reward probability ($p$) = 0.0025. RF mice were considered ready for recording when they completed >250 trials/h and operantly requested >80% of rewards. Therefore, the SM track cues were novel to the mice at the start of recording, whereas the RF track cues were highly familiar.

## In vivo electrophysiological data collection

All neural recordings occurred at least 16 h after the craniotomy surgery upon animal recovery. To record neural activity, mice were head-fixed on the VR rig, and a single craniotomy site was exposed and rinsed with saline. Immediately prior to recording, Phase 3B1 Neuropixels 1.0 silicon probes[57] with ground and reference shorted together were dipped in one of three fixable, lipophilic dyes (DiD, DiI, DiO,

ThermoFisher V22889) 10–20 times at 10 s intervals. Using a motorized micromanipulator (MP-225, Sutter), the probe was positioned over the craniotomy site at 10.6° from vertical and targeted to -50–300 μm anterior to the transverse sinus. The probe was located behind the mouse to minimize visual disturbances to the VR environment (Fig. 1a). To sample new cell populations on each day, three recordings were made per hemisphere, targeting at least 50 μm medial or lateral of previous recording sites on consecutive days. The 384 active recording channels on the probe were set to occupy the 4 mm closest to the tip. The reference electrode was connected to the ground pin, and then the probe was advanced slowly (1–5 μm/s) into the brain until mechanical resistance was met or activity near the probe tip quieted (~3000–4000 μm from the surface). The probe was retracted 50–100 μm and allowed to settle for at least 15 min. Finally, the craniotomy site was covered with sterile saline and silicone oil (Sigma-Aldrich 378429). Final probe depth and insertion position in medial-lateral and anterior-posterior dimensions were recorded daily.

Signals were sampled at 30 kHz with gain = 500 (2.34 μV/bit at 10-bit resolution) in the action potential band (high-pass 300 Hz to 10 kHz filter), digitized with a CMOS amplifier and multiplexer built into the probe, then written to disk with SpikeGLX software (https://billkarsh.github.io/SpikeGLX/). Local field potentials were similarly sampled at 2.5 kHz with gain = 250 (4.69 μV/bit at 10-bit resolution) and 0–300 Hz low-pass filter. Each Unity VR frame emitted a TTL pulse, which was then relayed by a microcontroller (Arduino Uno) to an auxiliary National Instruments data acquisition card (NI PXIe-6341 with NI BNC-2110). VR frame times were thus also recorded by SpikeGLX, permitting synchronization of VR data with electrophysiological traces.

## Offline spike sorting

The processing of electrophysiological recordings was performed by adapting Jennifer Colonell's fork of the Allen Institute for Brain Science ecephys library (open source from Github: https://github.com/jenniferColonell/ecephys_spike_sorting). Briefly, this pipeline employed CatGT (https://billkarsh.github.io/SpikeGLX/help/catgt_tshift/) to perform global common averaging referencing (-glbcar) and filter transient artifacts that appear on >25% of channels with peak amplitudes of at least 0.40 mV (-gfix = 0.4, 0.10, 0.02). Spike sorting and drift correction were then performed using the open source algorithm Kilosort 2.5[128]. Kilosort 2.5 improves on Kilosort 2, which identifies clusters in neural data but only tracks neural drift (https://github.com/MouseLand/Kilosort2)[129]. Next, ecephys modules extracted waveforms from the Kilosort output and calculated their properties (e.g., signal:noise ratio, duration, halfwidth, and peak:trough ratio) and the quality of corresponding spike clusters (e.g., autocorrelation contamination rate and interspike interval (ISI) violation rate, and number). Clusters were curated in Phy (https://github.com/cortex-lab/phy). All clusters with fewer than 100 spikes were excluded. Clusters with signal:noise ratio < 1.0, firing rate (FR) < 0.10 Hz, repolarization slope < 0, depth > 3200 μm from the probe tip, or halfwidth > 0.30 ms were excluded as noise using an open-source filtering module created by Emily A. Aery Jones (https://github.com/emilyasterjones/ecephys_spike_sorting/tree/master/ecephys_spike_sorting/modules/prePhy_filters). All remaining clusters were manually examined and labeled as good (i.e., likely corresponding to a well-isolated neural unit) or MUA (likely to represent multi-unit activity) based jointly on contamination rate and ISI violation rate and number. Only good units, or cells, with greater than 350 spikes were included for analysis in this paper. Sessions with fewer than 10 cells meeting these criteria were excluded from our neural datasets.

## Behavioral data preprocessing and synchronization

Clustered, curated neural spiking data were then synchronized to VR behavior data using the TTL pulse times recorded by SpikeGLX and saved by Unity with custom MATLAB code. Appropriate syncing was

confirmed by noting a high correlation between VR frame time differences and corresponding TTL time differences in SpikeGLX. As in previous work[23,55], we used interpolation to resample behavioral data at a constant 50 Hz framerate, since VR frame times are not constant. As the tracks were 400 cm long and effectively circular, given teleports, recorded positions less than 0 or greater than 400 cm were converted to the appropriate position on the previous or subsequent trial. Trial transitions were identified as time points where the difference in position across time bins was less than −100 cm (i.e. from -400 cm to -0 cm). As such, each VR frame was assigned a trial number. Running speed was computed as the difference in VR position between consecutive frames, divided by the framerate. After removing and interpolating over frames with speeds ≤ 5 cm/s or >150 cm/s, speed was smoothed with a Gaussian filter (standard deviation = 0.2 time bins). For all analyses, except lick and reward zone analyses (Supplementary Fig. 1i–l), stationary time bins (speed < 2 cm/s) were excluded. The stereotypy of reward-triggered licking and slowing for each session (Supplementary Fig. 1i) was examined by averaging position-binned lick counts and speed traces (bin size = 2.5 cm) across rewarded trials from 25 cm before to the end of the reward zone (75 cm total). We next found the location of absolute maxima in the mean lick and inverted mean speed traces and used the scipy.signal package's peak_prominences function to find the prominence of licking and slowing there, termed the slowing and licking magnitude.

### Tissue collection and storage

Immediately after probe explant after the final recording session, mice were sacrificed with an overdose of Euthasol (Virbac) and transcardially perfused with ice-cold phosphate-buffered saline (PBS). Brains were extracted and split into hemispheres. One hemisphere was selected at random to be flash frozen in a sterile RNase-free 1 mL Eppendorf tube on dry ice and stored at −80 °C for subsequent subdissection of the entorhinal cortex, nuclear RNA isolation, and RNA sequencing. The total time from recording end to hemisphere freezing was under 30 min for all mice. In preparation for immunohistochemistry, the remaining hemisphere was externally fixed in 4% paraformaldehyde in 0.1 M PB pH 7.4, for 48 h, followed by cryoprotection in 30% sucrose. Fixed hemispheres permitted validation of gene expression changes observed via RNA sequencing (see Fig. 8). To account for the loss of histologic confirmation of probe targeting, we required that recording sessions included in these analyses had greater than 50 probe channels with coordinated theta-rhythmic neural activity during running, which constitutes an electrophysiological signature of MEC[130].

### RNA isolation and bulk RNA-seq

Neuronal nuclei were isolated based on a previously described protocol with minor modifications[131]. Briefly, MECs, dissected from flash-frozen hemispheres at ±3.1 mm to 3.5 mm lateral and 4.75–5.25 mm posterior to bregma below 3.5 mm from dorsal brain surface, were dounce-homogenized (Wheaton, Cat. 357538) in 500 µL of EZ lysis buffer (Sigma Aldrich, NUC101) with 1 U/µL of RNase Inhibitor (Sigma Protector, 3335399001). Given this dissection approach, some nuclei from the lateral entorhinal cortex and parasubiculum may be included in this dataset. Samples were homogenized with 20 strokes each of the loose and tight pestles, and 500 µL of lysis buffer was added to the samples. Samples were incubated on ice for 7 min before filtering through a 40 mm filter and centrifuging at 500×g for 5 min at 4 °C. Conjugated mouse monoclonal anti-NeuN-AlexaFluor488 antibody (Millipore, Cat. MAB377X, RRID: AB_2149209) in a staining buffer (PBS with 1% BSA and 1 U/µL RNase Inhibitor) was added to the tube at a final dilution of 1:250. Samples were incubated on a tube rotator for 30 min at 4 °C and then spun for 5 min at 500×g at 4 °C. Samples were resuspended in 400 µL of staining buffer with Hoechst 33342 at a final concentration of 0.01 mg/mL. Samples were then filtered through a 35 mm Fluorescence-activated cell sorting (FACS) tube filter and sorted.

Nuclei were sorted by FACS on a BD FACSAria Fusion with a 100 mm nozzle and with a flow rate of 1–2.5. Nuclei were first gated by forward and side scatter, then gated for doublets with height and width. Nuclei that were both Hoechst+ and NeuN+ were sorted into the staining buffer with 2× concentration of RNase Inhibitor (see Supplementary Fig. 7a). Nuclei were spun down at 500×g for 5 min at 4 °C and resuspended in 250 µL Tri Reagent (Sigma Aldrich, T9424) for RNA expression analysis.

### Single-nucleus RNA-seq

Nuclei for snRNA-seq were isolated similarly to bulk samples. MECs from a young and aged mouse were sub-dissected and dounce homogenized as above. snRNA-seq samples were also stained with Hoescht 33342 and filtered as above, followed by FACS-sorting with the same parameters. Nuclei that were Hoescht+ were sorted into staining buffer with 2× concentration of RNase Inhibitor, spun down at 500 g for 5 min at 4 °C, and resuspended in the staining buffer. Nuclei were counted, and 25,000 from each sample were loaded onto a 10× Chromium Controller. Libraries were generated following 10× Genomics protocols and sequenced to a depth of 30,000 reads per nuclei on an Illumina 10B X. Fastqs were converted to feature barcode matrices using CellRanger 7.0, and downstream filtering and analysis were performed using Seurat v5[132].

### Statistical tools and reproducibility

Neural and behavioral data analysis was conducted in Python (3.8.16) using the NumPy (https://numpy.org/) and SciPy (https://scipy.org/) libraries to compute statistics and perform linear regressions. Linear mixed effects and logistic regression modeling were performed using the statsmodels package (https://www.statsmodels.org/). By default, $p$-values for reported β values and odds ratios from these models are the result of two-sided testing. Non-parametric tests were used to avoid assumptions about the normality of data distributions. Specifically, we applied two-sided Wilcoxon signed-rank tests to all paired data; two-sided Wilcoxon rank sum tests for all unpaired data with 2 groups; and Kruskal–Wallis $H$-tests for all comparisons across 3 groups. For multiple comparisons within 3 groups, we then performed pairwise two-sided Holm-corrected Conover's tests with the scikit_posthocs package (https://scikit-posthocs.readthedocs.io/). When non-parametric testing identified significant age group differences across neural activity metrics averaged across co-recorded cells in a session (Figs. 5e–g and 6d, e and Supplementary Fig. 6d, f, g), we verified these results with nested linear mixed effects models constructed with cells as the unit of analysis. To assess the significance of differential gene expression, we performed a two-sided Wald test followed by the Benjamini–Hochberg corrections of multiple comparisons. The significance of correlations between neural metrics and gene expression was assessed after Benjamini–Hochberg correction of $p$-values at a false discovery rate (FDR) = 0.10. GO biological process enrichment was confirmed by Fisher's Exact test and FDR calculation. Significant gene set enrichment was assessed by a two-sided permutation test followed by family-wise error rate correction (FWER). Unless otherwise noted, all statistical tests were two-sided; averaged data are presented as mean ± standard error of the mean (SEM); and correlation values are Pearson correlation coefficients. Immunohistochemical image analysis, but not behavioral and neural data collection and analysis, was performed blind to the conditions of the experiments. Sample sizes were consistent with previous similar studies and were not predetermined. Levels of significance across all figures are indicated as follows: $^*p < 0.05$, $^{**}p < 0.01$, $^{***}p < 0.001$, $^{****}p < 0.0001$. Results that approached significance ($0.05 < p < 0.10$) are indicated by a hashmark (#).

## Linear mixed effects models

To quantify behavioral performance and neural activity changes across sessions, we used linear mixed effects models (LMMs) that account for variance across animals. All models treated animal identity as a random effect, converged using restricted maximum likelihood estimation unless otherwise noted, and were implemented using the mixedlm method of the statsmodels.api package. We allowed only random intercepts for mice after confirming that including random slopes did not affect results, unless otherwise noted here. The significance of fixed and random effect coefficients for all LMMs was verified using two-sided Wald tests. Complete LMM results and Wald test results for each LMM have been included in Supplementary Data 1. Finally, for each LMM, we grossly verified key LMM assumptions. The linearity of relationships between model predictors and response variables and the homoscedasticity of residuals were assessed by plotting residuals against fitted values, which showed consistent linear relationships without constant variance in fitted values at particular residual values. Residual normality was also visually assessed with quantile-quantile plots. Finally, we confirmed that the features of the residual distributions did not vary systematically across animals generally or within age groups. Code to reproduce these plots is available.

For behavioral models (Fig. 1e, h and Supplementary Fig. 1g–k), fixed effects included age group as a categorical variable; session number as a continuous variable; the interaction of session and age group; sex as a categorical variable; and the interaction of sex and age group. In the models of SM block and alternation performance, we also included reward location order (context A reward centered at 270 cm or 370 cm) and the interaction of reward order and age group as categorical fixed effects. To model performance across SM trial groups (Block A [A], Block B [B], Alternation A Trials [A′], Alternation B [B′] Trials), we added the following other categorical fixed effects: number of rewards consumed in the previous trial group as a metric of satiety; VR context (A or B), the interaction of context and age group, task structure (block vs alternation), and the interaction of task structure and age group. In Fig. 1e, h and Supplementary Fig. 1g, h models, random slopes were also allowed for mice, since this improved model fit. When the behavioral model dependent variable was the fraction of reward requested, we confirmed that applying a logit transform of these fractions did not qualitatively change results or improve model fit. For interpretability, we chose to report untransformed fractions. Finally, an advantage of LMMs is robustness to missing data, such as the exclusion of incomplete or uncollected sessions (see VR setup). We have verified that age effects on SM behavior hold when excluded behavioral sessions are included.

For models of neural activity features (Figs. 2f, g and 6j and Supplementary Figs. 2h, j, 3f, g, and 6h, i), we also accounted for variance across cells using a nested random effects design. Fixed effects in these models included age group, sex, recording cohort, and the interaction of session and age group. As the interaction of sex and age group was not a significant fixed effect for any neural activity models and did not improve model fit via log-likelihood testing, it was not included in the final neural activity LMMs. Cohort (A–D) refers to groups of mice trained proximally in time and recorded sequentially as follows: all female RF mice (A), all male RF mice (B), all female young and aged SM mice and MA mice (C), and all male young and aged SM mice (D). The model of grid scale (Fig. 6j) also included unit recording depth and the interaction of depth and age group as continuous fixed effects. When the dependent variable was a similarity ratio (Fig. 2g and Supplementary Fig. 3g), we also confirmed that a logit transform of ratios did not qualitatively change results or improve model fit. For models of spatial firing with significant session and age group interaction effects (Fig. 2f, g and Supplementary Figs. 3f, g), we included alternative models with session added as a fixed effect in Supplementary Data Table 1. These models support the conclusion that aged grid and NGS spatial firing stability and context-

specificity improve significantly less than that of young grid and NGS cells over SM sessions.

To correlate gene expression with in vivo cellular features, like spatial firing coherence and within-map stability, we fit LMMs as described here to each age-modulated neural feature computed across RF sessions (see Fig. 7e) and SM sessions (see Supplementary Fig. 8a) and then calculated the animal mean fitted measure. If session number significantly modulated a given SM neural feature, we reported the difference between the animal-fitted mean across the last three sessions and first three sessions (last-first) as a measure of the change in that neural metric for a given mouse. This approach accounted for variability in these parameters across cells and animals and for differences in session number and spatial cell density across sessions (see Figs. 2d and 5d).

## Spatial firing rate and autocorrelation, shuffle procedure, and distance tuning

We generated spatial firing rate vectors empirically for recorded cells by binning track position into 2 cm bins and dividing spike counts by occupancy for each bin. We performed this for every trial to generate by-trial spatial tuning curves or across trials to generate an estimate of average firing rate as a function of position. When indicated in subsequent methods, we smoothed this vector with a Gaussian filter (standard deviation = 4 cm).

To assess distance tuning, reflected in periodic activity during dark running[55,57,58], we computed the autocorrelation of the smoothed by-trial spatial firing rate vector, linearized across dark trials, using lags up to 800 cm[55]. Next, we used scipy.signal package's find_peaks function to detect the prominence and lag of the maximum peak in the autocorrelation with a width > 10 cm, height > 0.1, and prominence > 0.15. 17,498 cells (40.2% SM cells) had a peak in their dark autocorrelation. Among these cells, the 99th percentile of shuffle peak prominences was $0.2017 \pm 0.00095$. Additionally, we computed a shuffled spatial autocorrelation after shifting spike times relative to elapsed distance by random offsets ($n = 100$ shuffles, offset drawn from a uniform random distribution with the interval [20 s, max_t] where max_t was the recording duration). For each shuffle, we computed the height of the autocorrelation at the same lag as the original maximal peak. A cell was determined to be distance-tuned if it had at least one qualifying peak in its dark autocorrelation, and the maximal peak prominence of the autocorrelation was greater than or equal to the 99th percentile of the shuffle distribution heights at that same lag.

The same spike time shuffle procedure was repurposed for the classification of speed-tuned and spatial cell types across both tasks and as a control for the comparison of cell and network-wide spatial firing quality across age groups in the RF task (Fig. 5e–h and Supplementary Fig. 5e).

## Calculation of spatial coherence, sparsity, and information

Spatial coherence was calculated as follows to assess the strength of spatial firing by measuring the local smoothness of the spatial firing rate vector[131]. First, a non-smoothed, trial-averaged spatial firing rate vector was calculated for each cell as above. For each 2 cm bin, the mean firing rate of the nearest eight bins was calculated. Then, coherence was computed as the correlation of firing rate with the mean nearest bin firing rate across bins, such that coherence ~1 implies perfectly spatial firing[133]. Spatial sparsity was calculated as follows to assess the coverage of the VR track by each cell's spatial firing[134], where $P_i$ is the probability of occupancy of bin $i$, $R_i$ is the mean firing rate in bin $i$, and $R$ is the overall mean firing rate:

$$\text{sparsity} = \sum (P_i * R_i^2)/R^2 \qquad (2)$$

Finally, spatial information was calculated in bits per second over position bins according to the previous work[135], as follows, with

variables defined as above:

$$\text{information} = \sum P_i R_i \log_2(R_i/R) \qquad (3)$$

## Speed tuning analyses

Speed tuning quality was determined by three scores: speed score, speed stability score, and trial stability speed score. Speed score was defined as the correlation between speed and instantaneous firing rate[20], calculated by smoothing the vector of spike counts at each time point with a Gaussian kernel (standard deviation = 40 cm). Speed, slope, and intercept were calculated using least-squares linear regression of instantaneous firing rate and speed to estimate speed tuning strength. To account for the correlation of running speed and position in VR, we calculated a speed stability score that takes the spike-weighted average of speed scores across track sections (5 × 80 cm)[57]:

$$\text{score} = \sum \frac{n_i p_i}{n} \qquad (4)$$

where $n$ is the total number of spikes, $i$ is the position bin index, and $n_i$ and $p_i$ are the number of spikes and speed score, respectively, calculated in bin $i$. Similarly, we calculated a trial speed stability score by taking the spike-weighted average of speed scores calculated on individual trials, such that $i$ was trial instead of position bin index. Trial stability or speed stability scores below the speed score for a given cell indicate instability in speed tuning over trials or position bins, respectively.

## Identification of functional cell types

Putative excitatory cells were separated from putative fast-spiking INs using thresholds on waveform duration and mean firing rate. Putative INs had a waveform duration < 0.35 ms or a mean firing rate (FR) > 40 Hz. For each session, we confirmed that the union of these thresholds captured the clustering of co-recorded cells in peak:trough ratio and duration space and separated cells into low waveform half-width, high FR vs high waveform halfwidth, low FR groups (Supplementary Fig. 6a)[136]. Among putative excitatory cells, we classified positively (+) and negatively (−) modulated speed cells[20]. A cell was speed-tuned if the absolute value of its speed score and speed stability score each exceeded the 99th percentile of the absolute value of speed and speed stability scores calculated on spike-shuffled data ($n = 100$ shuffles; see "Spatial firing rate and autocorrelation, shuffle procedure, and distance tuning" section). Speed cells were split into + or − groups based on the speed score sign. Among 64,316 cells across tasks, the 99th percentile of absolute values of shuffle speed scores was $0.1194 \pm 0.0002$, and the 99th percentile of absolute values of shuffle speed stability scores was $0.1239 \pm 0.0002$. Additionally, we classified spatial cells among putative excitatory cells using spatial cell coherence and sparsity scores. In the RF task, cells were defined as significantly spatial if their spatial coherence and sparsity scores both exceeded the 99th percentiles of shuffle coherence and sparsity scores for that cell ($n = 100$ shuffles). Among 20,928 RF cells, the 99th percentile of shuffle coherence and sparsity were $0.4440 \pm 0.0010$ and $0.8021 \pm 0.0022$, respectively. In the SM task, cells were defined as spatial if their Block A spatial coherence and sparsity scores both exceeded the 99th percentile of shuffle distributions of the corresponding scores (see Supplementary Fig. 2a–d). Only Block A trials were used to minimize the effect of context-dependent changes in spatial firing activity on these scores. Putative grid cells were identified as excitatory cells that exhibited significant distance tuning in the dark (see "Spatial firing rate and autocorrelation, shuffle procedure, and distance tuning" section) (Fig. 2a–d)[55,58]. In SM sessions, putative NGS cells were defined as the spatial cells that did not meet distance-tuning

criteria (Supplementary Fig. 3a–d). Putative grid cells with low dark (≤0.05 Hz) or overall (<0.3 Hz) mean FR were excluded. In some SM sessions, we observed a minority of classified putative grid and NGS cells with implausibly narrow (<4 cm) firing fields exclusively at reward delivery locations, consistent with unsuccessfully filtered lickport artifacts. To exclude these units, we applied a threshold on noise in trial-averaged spatial FR (mean Block A FR SEM/mean Block A FR > 0.45; mean Block B FR SEM/mean Block B FR > 0.45). Importantly, most putative grid cells also meet spatial cell criteria (Supplementary Fig. 2a [last]) and exhibit greater or equivalently coherent spatial firing compared to spatial cells within each age group (Supplementary Fig. 2b). In SM sessions, putative NGS cells were defined as the spatial cells that did not meet distance-tuning criteria (Supplementary Figs. 2a–c and 3a–d), such that putative grid and NGS cells constitute mutually exclusive subsets of spatially tuned cells recorded in MEC. For simplicity, we refer to putative grid and putative NGS cells as grid and NGS cells elsewhere. Finally, we also identified the subsets of non-spatial + or - speed cells, termed + or − speed only cells, and + speed-tuned INs and conjunctive speed-tuned grid cells, termed +IN speed cells and + grid speed cells, respectively (Supplementary Figs. 6e–g).

## Cross-trial correlation matrices and spatial stability, and similarity

To compute the spatial stability in both tasks and spatial similarity across VR contexts in the SM task, we first computed a smoothed (Gaussian filter, standard deviation = 5 cm), normalized by-trial spatial firing rate. As previously described in ref. 23, we re-scaled each neuron's firing rate separately to range between zero and one as such:

$$\mathbf{X}_{ijk} = \frac{\mathbf{X}_{ijk} - \min(\mathbf{X}_{::k})}{\max(\mathbf{X}_{::k}) - \min(\mathbf{X}_{::k})} \qquad (5)$$

Here, $\mathbf{X}_{ijk}$ denotes an $\mathbf{I} \times \mathbf{J} \times \mathbf{K}$ array, or tensor of normalized spatial firing rates, where $i$, $j$, and $k$, respectively represent trials, position bins, and cells. For SM sessions, we sorted alternation trials by VR context. As such, max() and min() operators were applied on a neuron-by-neuron basis. This rescaling step prevented neurons with high firing rates from washing out low-firing-rate neurons in population analyses.

To generate a cell's cross-trial correlation matrix, estimating the similarity of spatial firing across trial pairs (as in Fig. 2e and Supplementary Fig. 3e), we subtracted the mean FR from each trial in the matrix $\mathbf{X}_{ij}$ and computed the normalized cross-correlation of each pair of trials at a lag of 0 cm. Each cell's spatial stability score for a given set of trials was calculated as moving average correlation across all pairs of neighboring trials (5 nearest trials, see Figs. 2f and 5g and Supplementary Fig. 3f). In Figs. 5g and j, within map stability refers to stability calculated within each k-means labeled map and averaged across maps for each spatial cell (see "Population analysis and clustering algorithm with optimized k-values" section). This approach controls for reduced local stability caused by any spontaneous remapping events, which occur equally rarely across age groups in the RF task (see Supplementary Fig. 4i). To quantify context-specificity of each cell's spatial firing, we calculated a ratio between the similarity, or mean pairwise cross-correlation of spatial firing vectors, of context-matched (e.g., A × A, B × B, A × A′, B × B′) and -mismatched (e.g., A × B, A × B′, B × A′) trials. During the block phase, we computed the following similarity ratio for SM grid and NGS cells, which effectively quantifies the checkerboard structure in the top left corner of each cross-trial correlation matrix (see Fig. 2e):

$$\frac{\text{mean}(A \times A, B \times B)}{\text{mean}(A \times B)} \qquad (6)$$

During the alternation phase, we computed the following similarity ratio for SM grid and NGS cells to assess context-specificity with

respect to block trial spatial firing (i.e., similarity structure in the bottom left (or top right) matrix corner):

$$\frac{\text{mean}\left(\text{AxA}', \text{BxB}'\right)}{\text{mean}\left(\text{AxB}', \text{BxA}'\right)} \qquad (7)$$

Here, $A \times A$, $B \times B$, $A \times B$, $A \times A'$ $B \times Be'$, $A \times B'$, and $B \times A'$ denote the pairwise similarity of those pairs of trial groups. These ratios equaled 1 if context-matched trial spatial firing was equally as mutually similar as context-mismatched trial spatial firing. A ratio >1 implies context-specific spatial firing.

To quantify the degree of rate remapping in each grid and NGS cell across SM contexts, we found the percent change in the peak firing rate (i.e., largest firing rate in any spatial bin) across the trial-averaged normalized spatial firing vectors for block A and block B trials, as in previous work[23]. As a measure of global remapping, we calculated an alignment score between the normalized firing rate vectors in activity space (i.e., spatial dissimilarity). To do so, we computed the cosine similarity (vector dot product after normalization) between the block A and block B spatial firing vectors. We then subtracted this value from 1, such that a dissimilarity score of 0 would indicate an identical spatial firing pattern and a score of 1 would indicate orthogonal spatial representations. In Supplementary Fig. 2g–j, we compared the extent of grid and NGS cell rate and global remapping across SM contexts and age groups.

To generate network-wide trial-by-trial similarity matrices (Fig. 3a and Supplementary Figs. 4a–c, f, g, and 5i), we computed the correlation for vectors $\text{vec}(\mathbf{X}_{i::})$ and $\text{vec}(\mathbf{X}_{i'::})$ for all pairs of trials $(i, i')$ for all spatial cells in an RF session or all grid cells in an SM session. RF sessions with ≤10 spatial cells and SM sessions with <10 grid cells were excluded from population-level analyses, such as spatial map clustering and position decoding.

### Estimation of gain change responses and remapping coordination

Previous work has shown that dissociating visual flow and locomotion signals with reduced VR gain might identify putative grid cells, which are more sensitive than NGS cells to gain reductions[55,57]. We compared the relative magnitude of SM grid vs NGS cell responses to VR gain manipulation during alternation to validate the classification of these two functional cell types (Supplementary Fig. 2e, f). To do this, we selected a set of A′ trials before the gain change equal in number to A′ trials after the gain change. We then generated smoothed by-trial spatial firing rate vectors and cross-trial correlation matrices for each grid and NGS cell for those trial sets. Then, we computed the difference between the mean pairwise similarity within A′ trials before the gain change and the mean pairwise similarity of A′ trials before and after the gain change (within baseline−across gain change). The larger this difference, the greater the cell's gain change response. We replicated these steps to find the magnitude of B′ trial gain change responses and confirmed that the grid cell responses exceeded NGS cell responses for both contexts in each age group (Supplementary Fig. 2f).

To assess whether network-wide matrices would similarly reflect the composite of co-recorded cell activity in each age group, we estimated and compared remapping coordination across groups according to established methods[55]. In brief, remapping coordination was calculated as the correlation between the cross-trial correlation matrix of each unit and the network-wide similarity matrix, assembled using the remaining co-recorded units. No age group differences in remapping coordination were observed in either task (Supplementary Fig. 5c, d). Similarly, we computed the correlation between network-wide similarity matrices generated using spatial firing tensors from the front and back halves of the SM track (Supplementary Fig. 5i). We observed lower similarity between front and back matrices in MA and aged vs young sessions (Supplementary Fig. 5j), indicating a stronger

modulation of network activity by back track visual cues. While the probability of alignment of spatial maps to visual context on alternation trials is substantially lower for front vs back track grid network activity across sessions (Supplementary Fig. 5k [left] vs Fig. 3d left]), young front track grid network alignment probability on alternation trials uniquely increases over sessions. This suggested that young grid network spatial map alignment to context may be uniquely influenced by the cue tower. To control for possible subtle age effects on the timing of context recognition within trials, spatial firing tensors used for population-level analyses in the SM task (Fig. 3 and Supplementary Fig. 4) were assembled using back-track activity only.

### Population analysis and clustering algorithm with optimized $k$-values

To identify discrete spatial maps, consisting of clusters of trials with similar network-wide spatial firing patterns, in a data-driven manner, we adapted the factorized k-means clustering algorithm described in detail in Low et al. using Alex Williams' open-source lvl package (https://github.com/ahwillia/lvl/tree/master/lvl)[23]. Briefly, the low-rank factorization model of the network's normalized firing rate tensor ($\mathbf{X}_{ijk}$) is as follows:

$$\hat{\mathbf{X}}_{ijk} = \sum_{r=1}^{R} \mathbf{U}_i^{(r)} \mathbf{V}_{jk}^{(r)} \qquad (8)$$

In general, $R$ denotes the number of model components, or the model rank. As described in Low et al.[23], k-means clustering arises as a special case of this model, in which $R$ represents the number of clusters, traditionally termed k, or the number of spatial maps. In this case, $\mathbf{U}_i^{(r)}$ gives the elements of an $R \times I$ matrix, where each row gives the cluster assignment, or map label, for every trial $i$, coded in $R$-dimensional space (one-hot vectors). $\mathbf{V}_{jk}^{(r)}$ comprises the elements of an $\mathbf{R} \times \mathbf{J} \times \mathbf{K}$ array. For a given map label, $r$, the $\mathbf{J} \times \mathbf{K}$ slice of this matrix specifies a cluster centroid. This slice is interpretable as a spatial map in which the columns are $j$-dimensional vectors containing each neuron's spatial tuning curve. Identifying synchronous, network-wide transitions between spatial maps permitted the comparison of remapping frequency across age groups in each task (Fig. 3b and Supplementary Fig. 4h). Moreover, this enabled us to compare the alignment between spatial map and SM context transitions in each age group and over sessions. As reported by Low et al., this method accounts for both the global grid population and the rate remapping we observed across SM contexts (see Supplementary Fig. 2g, h)[23].

To capture the heterogeneity in trial-by-trial structure observed in network-wide similarity matrices, especially in response to the SM context switches, we first optimized the selection of $R$ for each session individually with the range $r = 1 - 4$. For interpretability, we substituted the signifier $k$ for $R$ as the k-means model rank in all figures and figure legends. To select $k$, we used a silhouette score maximization approach, deploying scikit-learn's silhouette_score function to measure average cluster quality[137]. For each $k > 1$, we fit a factorized $k$-means model with $k$ components and 100 restarts and calculated the average silhouette score across maps on each of 10 repetitions. Restarts accounted for model variability and helped to select the model best fit to the data within each repetition. The k that maximized the mean silhouette score across repetitions was selected (see Supplementary Fig. 4a).

To set the maximum possible k, we swept the range $k \le 8$ for SM sessions using this optimization procedure. Fewer than 10% of eligible SM sessions ($n = 13 / 134$) had an optimal $k > 4$. We compared the train and test cross-validation performance ($R^2$) of the k-means model at all possible k to assess if overfitting occurred at higher $k$. Indeed, we observed that an optimal $k > 4$ increased test $R^2$ minimally compared to the next best smaller $k$, while increasing train $R^2$ significantly (Supplementary Fig. 4b). In each session where $k > 4$, we also observed at least one low-quality cluster with no silhouette scores exceeding the

average across clusters. This indicated to us that $k > 4$ was rarely, if ever, appropriate to capture SM network-wide spatial firing patterns.

To exclude sessions that were not well-approximated by $k > 1$ models, termed one-map sessions, we compared the cross-validated performance (uncentered test $R^2$ averaged over ten replication sets) of $k$-means models fit on real vs shuffled datasets. Specifically, we employed a randomized cross-validation procedure in which 10% of the data were censored in a speckled-holdout pattern[138]. As in previous work[23], firing rate tensors ($\mathbf{X}_{ijk}$) were shuffled using a random rotation across trials. This preserved the overall data norm and correlations between neurons and position bins but disrupted the sparsity pattern on $\mathbf{U}_i^{(r)}$ imposed by $k$-means modeling. One-map sessions were those where the average model performance at the selected k on real data did not exceed that on shuffled data using a one-sided Wilcoxon signed-rank test ($\alpha < 0.05$) (Supplementary Fig. 4c, d). As previously reported in ref. 23, spontaneous remapping in most RF sessions was best captured by $k = 2$ models (Supplementary Fig. 4e). By contrast, we observed more variability in optimal k across SM sessions in all age groups due to heterogeneous remapping responses to context switches.

Finally, we confirmed that the discreteness of remapping was comparable across age groups in two ways. First, at the same rank as the optimal $k$, we compared the relative cross-validated test performance of the $k$-means model and truncated singular value decomposition (tSVD), which is a form of uncentered principal components analysis (PCA) in this case. As described in Low et al. [23], the interpretation that network activity belongs to discrete maps was supported by the comparable performance of the more restrictive $k$-means model to the tSVD model, which permits trials to constitute linear combinations of spatial maps instead of discrete spatial maps. Component-matched tSVD and $k$-means performance were strongly correlated in all sessions in each age group in both tasks (Supplementary Fig. 5a). Additionally, we confirmed that within-map mean pairwise trial similarity was significantly greater than across-map pairwise trial similarity for all age groups in both tasks (Supplementary Fig. 5b). These results indicated that discrete maps indeed capture network-wide similarity patterns in both tasks.

## Map label and identity assignment to assess context alignment

To address the fact that $k$-means cluster labels are arbitrarily assigned, we systematically re-labeled spatial maps. For SM sessions with optimal $k \geq 2$, spatial maps were re-labeled based on their order of appearance in the task, such that map 1 was the map with the most Block A trials. For $k = 3$–4 sessions, map 2 was the remaining map with the most Block B trials. For $k = 4$ sessions, map 3 was the remaining map with more trials before the midpoint of the context-sorted alternation phase. RF spatial maps were re-labeled in order of their mean running speeds, such that map 1 had the slowest running speed. Remaps were identified as changes in the map label between consecutive trials. To calculate remapping frequency, we divided the remap count by the trial count for each task phase.

To assess the alignment of SM context and spatial map transitions, it was necessary to assign context identities (A or B) to each labeled map (see Fig. 3a). After map labeling, map 1 dominated Block A trials and was thus assumed to represent context A. Similarly, map 2 was assigned to represent context B. For sessions with optimal $k > 2$, maps 3 and 4 were each assigned to represent context A or B, depending on whether they had greater relative mean pairwise similarity to Block A or B trials. As such, it was possible to assign up to three maps to represent one context for $k = 4$ sessions. We then computed the fraction of aligned trials, in which VR context and spatial map identity matched, overall and for each task phase.

## Logistic regression models of by-trial responses

To model binary outcomes for each SM alternation trial (i.e., map-context alignment or a reward request), we used the logit method of

the statsmodel.api package to perform logistic regression. In the model of map-context alignment probability by trial, we specified the following fixed effects as categorical variables: age group, the interaction of age group and sex, the interaction of age group and VR context (A/B), the interaction of age group and session, and the interaction of age group and the number of trials since a random VR context switch (0–11). To model reward request probability by trial, fixed effects included those above and the interaction of age group and remap occurrence as a categorical variable (No/Yes), and the interaction of age group and an aligned trial occurrence as a categorical variable (no/yes). Unlike the linear mixed effects models, these models do not account for animal variance with random effects.

## Local field potentials and power calculation

Local field potentials (LFP) were analyzed on one channel per session across both tasks (Fig. 4). We selected the channel with the highest theta power after subsetting on the 200 channels closest to the probe tip with at least one good recorded cell. Power spectral density was calculated using Welch's method from scipy.signal package. Power in each frequency band (6–12 Hz for theta, 20–50 Hz for slow gamma, and 50–110 Hz for fast gamma) was calculated using a multitaper spectrogram using the spectral_connectivity package with a window of $0.5\,s$[139]. To control for the effects of differences in running speed across age groups (see Supplementary Fig. 1c, d) on theta power[65,66], we up-sampled frequency band power traces (2 Hz) to the VR frame-rate (50 Hz) using a second order spline for interpolation and averaged power across timepoints where mice were running at intermediate speeds ($20 \leq speed \leq 40$ cm/s). We confirmed that this range collapsed mean and peak running speed differences across age groups (see "Results" section).

## Position decoding analysis

To predict the animal's position from the spiking activity of all co-recorded cells, we fit spherically projected multivariate linear models[140] as implemented by Low et al. [23]. This approach was selected because of the effective circularity of VR tracks. Predicting a circular variable, such as track position at a given time ($y_t \in [0, 2\pi)$), from linear covariates, such as the number of spikes fired by neurons $n$ at times $t$ ($s_{nt}$), requires a circular linear regression model. These models are termed decoders as common practice[141]. In this model, two regression coefficients, $\beta_n^{(1)}$ and $\beta_n^{(2)}$, are optimized for each neuron with an expectation maximization procedure previously described in ref. 140. The model estimate given set of test inputs after fitting to training data is as follows:

$$\hat{y}_t = \text{atan2}\left(\sum_{n=1}^{N} \beta_n^{(1)} s_{nt}, \sum_{n=1}^{N} \beta_n^{(2)} s_{nt}\right) \quad (9)$$

Here, atan2() corresponds to the two-argument arctangent function. Decoder score refers to the average of $\cos(y_t - \hat{y}_t)$ over time bins in each session's testing set, such that random position guesses over a circle would yield a score of 0 and perfect performance would have a score of 1.

For ten repetitions per session, we fit decoding models to training sets comprising a randomly selected 90% of the network activity from RF sessions with >10 spatial cells and an optimal $k = 2$, and tested them on the held-out 10% of network activity. Synthetically ablating non-spatial cells from the network did not qualitatively alter decoder score differences across age groups, so we presented results from models trained and tested on spatial cell activity only (Fig. 5h–j and Supplementary Figs. 5e, f). Each datapoint in these figures represents the average decoder score across repetitions. For each session, training data were down-sampled to match spike number, position bins, running speed, and observation number across training sets drawn from the same session (e.g., map 1, map 2, both maps). We compared

decoder scores to those for a spike time shuffle control for each session (see "Spatial firing rate and autocorrelation, shuffle procedure, and distance tuning" section). Shuffle training sets were selected and down-sampled in the same manner. Since this shuffle disrupts spatial firing patterns, this control approximated chance decoder performance.

To further illustrate that distinct patterns of grid network activity occur across VR contexts in the SM task, we applied the same position decoding approach to training sets drawn from SM grid network activity across trial groups for all sessions with ≥ 10 grid cells (Maps A, B, A′ and B′ in Supplementary Fig. 5g, h). Trivially, decoders trained and tested on different trials perform worse than those trained and tested on the same trials (Supplementary Fig. 5g). Moreover, decoders trained and tested across context-mismatched vs -matched trials performed worse (e.g., Train A/Test B′ vs Train A/Test A′) (Supplementary Fig. 5h), confirming the qualitative observation that grid network activity was more conserved across context-matched vs -mismatched trials.

### Estimation of grid scale and unit recording depth

To compare the dorsal-ventral gradient of entorhinal grid cell scales[12,17,72], across age groups, we estimated grid cell scale and recording depth from the brain surface along the probe using all SM sessions (Fig. 6f–j). Grid scale was calculated using scipy.signal package's find_peaks function to detect the location of the first peak in the autocorrelation on dark trials (width > 8 cm, height > 0.05, prominence > 0.05). We relaxed these peak detection criteria from those used to identify distance-tuning (see "Spatial firing rate and autocorrelation, shuffle procedure, and distance tuning" section) to consistently select the first, instead of the largest, significant autocorrelation peak. To estimate each grid cell's depth, we subtracted the median distance from the probe tip of the cell's recorded spikes from the maximum inserted depth of the probe for that recording session.

### Differential gene expression and correlational analysis

FASTQ files were mapped to the mouse transcriptome (mm39) that included nascent RNAs (introns included in the index) using Kallisto[142]. The mapped reads were aggregated into a gene-based count matrix. Differential expression analysis was performed on the count matrix using DESeq2[143]. Specifically, the likelihood ratio test was used for differential expression testing while controlling for sex-specific batch effects. A volcano plot of the differential expression results was generated with EnhancedVolcano (Fig. 7b). Heatmaps of the top differentially expressed genes with age (young vs aged) with row Z-score normalization across animals, as determined by adjusted $p$-value, were generated using pheatmap (Supplementary Figs. 7 and 8p).

To determine the relationship between aging gene expression and measures of neuronal activity, we performed correlational analysis between these measures. We determined the normalized expression of the gene sets that increase expression with aging (Aging Up, $n = 170$ genes expressed in every sample), decrease expression with aging (Aging Down, $n = 260$ genes expressed in every sample), and all aging genes (Aging All, $n = 430$ genes expressed in every sample) for each individual mouse that performed the RF task. The Pearson correlation of these values with the LMM-fitted values of the neuronal activity measures was computed. The relationships between variables are represented in a correlation matrix (Fig. 7e) generated using the corrplot package. Similarly, we assessed the correlative relationships among age differentially expressed genes (Aging Up, $n = 28$; Aging Down, $n = 11$; Aging All, $n = 39$ genes) and age-modulated neural features across SM mice (Supplementary Fig. 8a). To uncover individual genes whose expression was correlated with these variables, we used linear regression followed by a Benjamini–Hochberg correction (FDR = 0.10) to mitigate false positive correlations resulting from

multiple hypothesis testing. This procedure is illustrated in Fig. 7f, where we present the correlation coefficient and adjusted $p$-value of all coherence-correlated genes, significantly above vs not below the FDR threshold. Two examples of genes with significant correlations are presented in Fig. 7g. Very few transcriptomic correlates of any neural metrics met this stringent FDR threshold (Supplementary Fig. 8b). All genes shown in Supplementary Fig. 8c, q were instead at least related to the corresponding metric without the FDR correction (linear regression $p < 0.05$).

### GO and GSEA

GO enrichment analysis was performed on several gene sets using Panther[144]. Specifically, the biological process GO sets were probed with genes differentially expressed during aging and those that significantly correlated with spatial firing coherence (FDR = 0.10). The top non-redundant biological process sets by FDR that had an enrichment of 1.25 above expected are shown in the plots (Fig. 7c, h and Supplementary Fig. 7c). GSEA[145,146] was performed on the bulk neuronal RNA-seq. Briefly, the mouse-ortholog hallmark gene sets were probed with 21,388 expressed genes from the RNA-seq dataset for enrichment in the probed gene set of correlations between ages and expression. Gene set significance was determined using a permutation test and a threshold on family-wise error rate (FWER) (6 gene sets significant at $p < 0.05$; 2 gene sets significant with FWER < 0.25 shown in Fig. 7c).

### Single nuclei RNA-seq analysis

Cell Ranger was used to generate count matrices with include introns set to TRUE. Feature barcode count matrices were imported into Seurat v5[132]. High-quality nuclei with a feature number greater than 200 and less than 7000, as well as mitochondrial content less than 0.25%, were kept for downstream processing. Both samples were integrated using the IntegrateData function after identifying integration anchors. The libraries were processed using the standard Seurat workflow to generate clusters. Subsequently, neurons were subclustered and dimensionality reduction was performed again, and new clusters were generated. These clusters were annotated based on marker expression (Supplementary Fig. 7d) found from FindAllMarkers in Seurat, and separated into excitatory and inhibitory neurons (Fig. 7i), as well as layer-specific excitatory neurons and inhibitory subclusters (Supplementary Fig. 7e)[147–150].

To determine the relationship between cell type and age-related transcriptional changes, we generated module scores for genes that increase with age ($n = 160$ genes represented in snRNA-seq data) and decrease with age ($n = 242$ genes) using the AddModuleScore function in Seurat v5[132]. These scores were overlaid onto the UMAPs of neurons (Supplementary Fig. 7f vs h for age down vs age up), as well as used to determine the relative expression differences between excitatory and inhibitory neurons (Supplementary Figs. 7g, i). The Wilcoxon rank sum test with continuity correction was used to determine if there were significant differences in expression between the clusters. Similarly, we generated module scores for genes positively and negatively correlated with spatial firing coherence ($n = 26$ positively correlated genes and n = 290 negatively correlated genes after Benjamini–Hochberg correction that are represented in the snRNA-seq data) to investigate the relationship between coherence and cell type. The correlation core module scores were overlaid onto UMAPs of neurons (Fig. 7j, l) and used to determine expression differences between excitatory and inhibitory neurons (Fig. 7k, m). The correlation core module scores were also determined for each neuronal cluster in Supplementary Fig. 7j. Additionally, the bulk neuronal RNA-seq age-related changes of the correlation module genes are presented in Supplementary Fig. 7k.

To determine how markers of excitatory and inhibitory neurons change with age, we found marker genes for these two cell types in comparison to one another using the FindMarkers function in Seurat

v5[132]. The age-related changes of these gene sets in the bulk neuronal RNA-seq dataset are represented in Supplementary Fig. 7k.

## Immunohistochemistry, microscopy, and image analysis

Fixed, cryoprotected hemispheres dissected from RF mice were flash frozen and sectioned with a microtome in 40-μm-thick sections. The sections were collected from the microtome with a fine brush and placed in cryopreservation media (30% glycerol, 30% ethylene glycol in phosphate buffer). Double-label immunohistochemical staining for perineuronal nets (PNNs) and parvalbumin-expressing interneurons (PV + INs) was performed on free-floating sections under constant agitation. In brief, sections were washed (3 × 5 min in 1× TBST), pre-treated (30 min in TBST with 0.3% Triton X-100), washed (3 × 5 min in 1× TBST), processed for antigen retrieval via twice heating for 10 min at 95 °C in citrate buffer (pH 6.0; Sigma, C9999), washed again in 1× TBST at room temperature, and then blocked in a solution containing 5% normal donkey serum (NDS) in TBST for 1 h. Sections were then incubated in a primary antibody solution (1:1000 rabbit anti-PV antibody [Swant, RRID: AB_10000344] in blocking solution) for 8–12 h at 4 °C. Sections were washed, incubated in secondary antibody solution (donkey anti-rabbit AlexaFluor 555 [Life Technologies, Cat# A-31572, RRID: AB_162543] in TBST with 3% NDS) for 1 h at room temperature, and washed again. The sections were subsequently incubated in Carbo-Free Blocking Solution (Vector Labs, Cat# SP-5040) for 30 min. Sections were incubated with fluorescein conjugated *Wisteria floribunda* agglutinin (lectin) (WFA) (4 μg/mL, Vector Labs, Cat# FL-1351-2) in TBS for 30 min. Sections were washed (3 × 5 min in 1× TBST) with the first wash containing Hoechst 33342 (2 μg/mL, Life Technologies, Cat# H3570) before mounting on slides. Followed by a 24 h drying period, stained sections were imaged using an LSM900 confocal microscope (Carl Zeiss) with a 20X objective using Zen software (version #3.3). Images of 2–3 fields of view (FOVs) within MEC per mouse were acquired across three channels in 10 × 1.5 μm steps through a 13.5 μm z-plane.

An experimenter blinded to the age and mouse identity of each image then performed image analysis using ImageJ (NIH), using an open-source, semi-automated approach for quantifying the density of PNNs, PV+ INs, and their co-localization (PIPSQUEAK)[151]. In brief, images were cropped to eliminate any areas without stained tissue from the FOV while noting the resulting tissue volume; split by channel; and summed across Z projections. A Z slice-summed composite of PV and WFA channels was also generated. We used the semi-automated double-label analysis within PIPSQUEAK to sequentially perform rolling ball radius background subtraction and estimate ROIs corresponding to PV + IN cell bodies and WFA-labeled PNNs. Background subtraction occasionally resulted in false negatives, so all ROIs were manually inspected, adjusted, and added as necessary to surround all PV INs and PNNs. Colocalization of PV IN and PNN ROIs was automatically assessed and verified using the summed composite image. The number of PV + IN, PNN, and colocalized ROIs and their area and mean pixel intensity relative to background were recorded. While no age differences in PV IN+ and PNN ROI area or intensity were detected, PNNs not colocalized with PV+ INs were doubled in density with age (Fig. 8).

## Reporting summary

Further information on research design is available in the Nature Portfolio Reporting Summary linked to this article.

## Data availability

Pre-processed behavioral data, neural data, and immunohistochemistry images associated with this manuscript are available via Dryad (https://doi.org/10.5061/dryad.8cz8w9h0d). Processed neural data included spike times and myriad waveform features for all recorded cells from all sessions and mice, facilitating replication of manuscript findings and many additional analyses. Raw neural data are under restricted access given their large size, the significant time burden of data processing, and the limited number of additional analyses raw vs processed neural data enable. These data may be requested by contacting the corresponding author. Transcriptomic bulk and single nucleus RNA sequencing data from this study are separately available via the NCBI GEO database (Dataset 1 [Fig. 7 and Supplementary Fig. 7]: Accession Number: GSE263347; Dataset 2 [Supplementary Fig. 8]: Accession Number: GSE281777). Supplementary data tables containing the results of transcriptomic data analyses of each dataset are included with this publication. Most transcriptomic data analyses (Fig. 7 and Supplementary Figs. 7 and 8) used commonly available R and Python packages that are specified in the manuscript Methods. Source data are also provided with this publication. Source data are provided with this paper.

## Code availability

Python code to subsequently post-process and then analyze these behavioral and neural data in the order of manuscript figures (Figs. 1–6 and Supplementary Figs. 1–6) and visualize correlational transcriptomic data and immunohistochemistry data (Figs. 7 and 8 and Supplementary Fig. 8) are publicly available[152]: https://doi.org/10.5281/zenodo.15851471.153

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

## Acknowledgements

We thank A. Diaz for assistance with animal care. F.K. Masuda for providing training on surgeries, VR behavioral training, and neural recordings. C. Li for assistance with VR behavioral training; and I. Low and E. A. A. Jones for sharing code; and Giocomo Lab members for discussions and feedback, especially L. J. Warton. L.M.G. is an HHMI Investigator and this work was supported by the Stanford University Medical Scientist Training Program (T32-GM007365 and T32-GM145402) and the National Institutes of Aging under a Ruth L. Kirschstein National Research Service Award (F30-AG079494) (to C.S.H); the Simons Foundation (SCPAB 811229) (to S.A.V. and L.M.G.); and the NIH Brain Initiatives (U19NS118284), NIMH (MH126904 and MH130452), the Simons Foundation (SCGB 542987SPI), NIDA (DA042012), the Vallee Foundation, and the James S. McDonnell Foundation (to L.M.G.).

## Author contributions

C.S.H.: conceptualization, behavioral and neural recording data methodology, collection, and formal analysis, writing—original draft, writing—review and editing, and visualization. K.J.B.P.: transcriptomic data methodology, collection, and formal analysis, writing—original draft, and visualization. J.M.S.: transcriptomic and immunohistochemical data methodology, collection, and formal analysis, writing—original draft, and visualization. S.A.V.: conceptualization, writing—review and editing, supervision, and funding. L.M.G.: conceptualization, writing—review and editing, supervision, and funding.

## Competing interests

The authors declare no competing interests.
