## [Transparent Peer Review file · Nature Communications]

Spatial Coding Dysfunction & Network Instability in the Aging Medial Entorhinal Cortex

Corresponding Author: Dr Lisa Giocomo

Version 0:

Reviewer comments:

Reviewer #1

(Remarks to the Author)

The study by Herber and colleagues is an in-depth analysis on age-related changes in the MEC. The authors used a behavioral paradigm that they had established in a previous study. It is an elegant approach that allows them the assessment of defined cell types probed by in vivo electrophysiology in mice navigating a virtual environment and that enables the correlation of age-related functional alterations with task performance. The functional characterization includes characterization of spatially tuned cells at the cellular level addressing the question whether/how stability and context-specificity of grid cells and non-grid spatial cells may account for impairments in task performance in aged mice. This in itself is already a huge work load, and the results are very interesting and important for many neuroscientists whose investigations aim at a better understanding as to how spatial coding relates to episodic memory. However, the study comprises also results addressing gene expression changes across ages. This part of the study is more descriptive but is of considerable relevance to researchers aiming at the identification of cellular mechanisms that may account for pathophysiology in neurodegenerative diseases such as AD but also in a number of neurological diseases that affect the hippocampal formation and that exhibit mild cognitive impairments as first symptoms of the disease.

My evaluation is short because I strongly support the publication of this study in the current form with only few minor modifications. I admit that the study is difficult to read even for somebody who is familiar with past and present research in the MEC. But it is also evident that this study underwent already several changes to address questions raised by reviewers. Below please find a couple of questions/comments.

1. In the scheme of Fig. 1b, the location of the reward does not correspond to the what is shown in the Extended Data Figure 1a. However, I strongly recommend to replace Fig. b and f with Extended Data Figure 1a,b as the latter is a much better and more comprehensible illustration of the experimental design.
2. I suggest to replace "Fraction Reward Requested" with "Percentage possible rewards". The authors need not follow my suggestion (the reader will understand what is intended), but I think that "Fraction reward requested" is semantically not correct.
3. The order of the Figure legends differs in the manuscript and Extended data: they are before the main Figures, but after the Figures in the Extended Data. This also contributes to some extent to making the paper difficult to read.
4. Extended Data Fig. 1e is more of a visual contrast sensitivity assessment rather than a visual acuity assessment. The latter would entail visual measuring the finest detail or smallest angle the animal can resolve, and would indeed be the better control. However, at this stage such a demand is not appropriate, and I suggest rewording.
5. Some panels are not mentioned/discussed in the main text (for example Fig. 2c and Extended Data Fig. 8a).
6. Wouldn't the slight improvement in performance in the aged group in Fig. 1e be also expected in Figure 2f for grid cells and Extended Data Fig. 3f-g for NGS cells? If there is a correlation between spatial coding and performance, would this suggest that other cells account for the slight improvement in performance in Fig. 1e for aged mice?
7. Please add the scales and y-ticks in Extended Data Fig.4a-d to allow comparisons across age groups. Since this provides the basis of the remap/alignment analysis, are there differences between different age groups for their optimized average silhouette score?

8. For Fig. 8k, could the authors add the median for these violin plots?

(Remarks on code availability)

Reviewer #2

(Remarks to the Author)

This work studied spatial coding in the aging medial entorhinal cortex (MEC). The authors found disfunction of spatial coding and instable network with age. In vivo electrophysiology identified spatial memory deficits arising with impaired stabilization of context-specific spatial firing. Aged grid networks appeared shifting firing patterns often but with poor alignment to context dynamics. Aged spatial firing was also unstable in an unchanging environment. They further employed transcriptomic tools and identified 61 spatial coding quality-related genes differentially expressed with age in MEC. These genes were interneuron-enriched and related to synaptic plasticity, notably including a perineuronal net component, according to bioinformatic analyses. Collectively, the authors accumulated evidence from the perspectives of neuronal firing and transcriptome, demonstrating the spatial coding ability destabilize with age. The experiments were well designed and the results are sound. However, I have a few concerns that should be addressed to strengthen this manuscript.

Major

1. Throughout this manuscript, I did not find originally recorded data formatting place field. I would suggest the authors to include some examples.
2. The molecular/cellular mechanisms underlying changes of MEC neuronal firing with age regarding destabilized spatial coding were primarily from transcriptomic data. Ideally, the author should provide evidence at the protein level, coherent with the gene/s (for example, Hapln4) exhibiting the top changed expression profile to consolidate their conclusion.
3. Figure 4: Maybe I missed something. The calculation process for band power is not clear to me. In panel b, either the value or the unit for the Power Spectral Density (V^2/Hz) is far beyond the reasonable range. What is the unit for power in panel c?
4. The authors used both male and female mice in this study. While making each statement based on data collected from animals with both sexes, the sex factor should be considered and clearly stated.

Minor

1. P2: "hexagonally tile physical space": grammar problem
2. In Figure 3b, there are several black circles. What do they indicate?

(Remarks on code availability)

Reviewer #3

(Remarks to the Author)

This study reports an impressive dataset describing neuronal firing and gene expression in the medial entorhinal cortex of young, middle-aged and old mice. Important results include age-related deficits in context-dependent behavior, changes with age in the stability and context-dependence of grid cell firing, and age-dependent changes in gene expression that correlate with the electrophysiological deficits. The experiments appear carefully designed and executed. The analyses are thorough and in general convincing. The results will be influential, as they give the first compelling insights into how entorhinal cortex circuit physiology changes with age, and offer hints about underlying molecular changes.

I have some suggestions and minor concerns.

1. A potential concern is that some of the comparisons of spiking data between young, middle-aged and aged groups appear to rely on whether relationships are, or are not, significant within a group. E.g. on p7 there is a reported decrease in stability over sessions for young and middle-aged but not aged mice. The lack of significance in the aged group is insufficient to conclude that it differs from the other groups. More appropriate would be to evaluate models that include age as a factor (as is done for the behavioural analysis). From my reading of the methods this appears to have been done but not reported. I suspect the conclusions will hold, but without reporting these analyses one can't rule out that age has made the data more noisy and so 'significant' relationships are less easily detectable.
2. When comparisons of spiking population data are made between groups, the reporting should clarify what the unit of analysis is. My guess is that these are single, or median, observations per animal. If multiple observations/sessions per animal have been used then a nested analysis would be more appropriate.
3. Can the methods include an explanation for how the pseudorandomisation for alternating trials was implemented?
4. In Figure 1 the implementation of the alternation phase of the split maze task is potentially confusing. In the schematic in Figure 1c, it appears that trials alternate, rather than being randomised, while the y-axis trial numbering in panel 1d implies ordering, whereas the alternating blocks have been re-ordered. This is clear after cross-checking with the methods section, but for the benefit of a lazy reader perhaps modify this part of the axis and clarify in the legend?
5. The methods should clarify the tests used for estimation of statistical significance from linear mixed effect models. A supplemental table reporting the degrees of freedom and test statistics for each of the analyses could be helpful. Procedures

used to assess model validity of model assumptions should also be described.

(Remarks on code availability)

I haven't had time to download the data and run the code but have read through the repository. The code appears well documented and its clear how the figures were generated.

There is a comment in the readme about variation because of random number generation. If this is a concern the authors might add random number seeds.

Version 1:

Reviewer comments:

Reviewer #1

(Remarks to the Author)

This is an important paper in the field and deserves to reach a large readership as soon as possible. I have no further requests.

(Remarks on code availability)

We actually use some of the code and find it very useful. In doing so, we made a new discovery and are very grateful to the authors.

Reviewer #2

(Remarks to the Author)

The clarity and readability of the manuscript have significantly improved, and the previous comments and concerns have been adequately addressed. I only have a few remaining comments, listed below :

1. Figure 1, panel j: In the MA group, two lines terminate at Session 3 and Session 4, respectively, while in the Aged group, two lines terminate at Session 3. This information should be clearly stated in the figure legend, and the potential impact of these terminations on the study's conclusions should be addressed in the discussion. Otherwise, I would recommend that the authors consider removing these samples.
2. Figure 8: The resolution of the representative images in panel a is quite low. Additionally, since the statistics presented in panels b and c, are based on density per μm^3 (cubic volume), it would be more appropriate to include 3D representative examples to better illustrate and support the data shown in panels b and c.
3. Ideally, examples of raw data from the in vivo silicon probe recordings, along with the corresponding identified cell activity, should be provided.

(Remarks on code availability)

Reviewer #3

(Remarks to the Author)

My concerns have been addressed. The study describes an important dataset that gives new insight into age-related changes in spatial cognition.

(Remarks on code availability)

We thank the reviewers for their thoughtful comments on the manuscript, and we appreciate the enthusiasm on the part of the reviewers as to the impact and novelty of this work. We have addressed the reviewer comments through the addition of a new Supplementary Table and multiple figure and text changes. We feel these changes strengthen the paper's conclusions; enhance its rigor; and clarify the molecular and circuit mechanistic insights it provides about MEC aging. Below, we briefly summarize the major changes we made, followed by a detailed response to each Reviewer question and suggestion.

Summary of changes in response to reviewers:

1. To clarify the structure of the Split Maze (SM) task, we have replaced the VR track schematics in Figure 1 with the more intuitive track images from Extended Data Figure 1, added context to the descriptions of SM raster plots that are sorted by context, and better explained pseudorandomization in the revised Methods section.
2. To explore possible mechanisms of the small improvement in aged performance over SM sessions in our dataset, we analyzed speed cell tuning sensitivity and stability over sessions and its relationship to SM task performance (Reviewer Figure 1). This revealed other MEC activity or non-MEC activity is more likely to explain this phenomenon. Reviewer Figures 1e-g are included in the manuscript as revised Extended Data Figure 6h-j.
3. To further exclude the possibility that age biases the k-means algorithm that clusters trials as belonging to discrete grid network spatial maps, we confirmed the average silhouette score at optimized k did not differ significantly across age groups (Reviewer Figure 2).
4. To more comprehensively share linear mixed effects modeling (LMM) results with readers without increasing manuscript length, we have added a new Supplementary Table (new Supplementary Table 1) to the revised manuscript and updated the relevant Methods with details regarding the validation of LMM model assumption and fixed effect significance testing.
5. To improve readability and clarity, we also made the following changes:
 - a. Added median lines to all violin plots compare gene expression in MEC excitatory cells and interneurons.
 - b. Corrected the term visual acuity to contrast sensitivity.
 - c. Corrected the units of local field potential power in Figure 4
 - d. Units of analysis for all LMMs are included in Supplementary Table 1. All other units of analysis are specified in appropriate figure legends or Results text.

Please note that we have provided two figures to aid in the response to reviewer comments, but Reviewer Figure 1a-d, h and Reviewer Figure 2 are not currently incorporated into the revised manuscripts. If the reviewers or editor feel these figures would clarify the manuscript, we would be happy to incorporate them.

Response to Reviewer Comments:

Reviewer 1: The study by Herber and colleagues is an in-depth analysis on age-related changes in the MEC. The authors used a behavioral paradigm that they had established in a previous study. It is an elegant approach that allows them the assessment of defined cell types probed by in vivo electrophysiology in mice navigating a virtual environment and that enables the correlation of age-related functional alterations with task performance. The functional characterization includes characterization of spatially tuned cells at the cellular level addressing the question whether/how stability and context-specificity of grid cells and non-grid spatial cells may account for impairments in task performance in aged mice. This in itself is already a huge work load, and the results are very interesting and important for many neuroscientists whose investigations aim at a better understanding as to how spatial coding relates to episodic memory. However, the study comprises also results addressing gene expression changes across ages. This part of the study is more descriptive but is of considerable relevance to researchers aiming at the identification of cellular mechanisms that may account for pathophysiology in neurodegenerative diseases such as AD but also in a number of neurological diseases that affect the hippocampal formation and that exhibit mild cognitive impairments as first symptoms of the disease.

My evaluation is short because I strongly support the publication of this study in the current form with only few minor modifications. I admit that the study is difficult to read even for somebody who is familiar with past and present research in the MEC. But it is also evident that this study underwent already several changes to address questions raised by reviewers. Below please find a couple of questions/comments.

We thank the reviewer for their precise summary of our study's results and for their positive comments regarding its impact and utility for a broad range of neuroscientists. Below, we address several of the reviewer's helpful suggestions to improve the clarity of the manuscript for readers.

Suggestions & Minor Concerns:

1. In the scheme of Fig. 1b, the location of the reward does not correspond to the what is shown in the Extended Data Figure 1a. However, I strongly recommend to replace Fig. b and f with Extended Data Figure 1a,b as the latter is a much better and more comprehensible illustration of the experimental design.

To clarify the experimental design for readers in the revised manuscript, we have switched the placement of VR track screenshots from the original Extended Data Fig. 1a and b (now Fig. 1b and d) with the VR track schematics in original Figure 1b and f. In addition, we have ensured that the example reward zones for Context A and B trials now match across revised Fig. 1b, d and Extended Data Fig. 1a, b.

2. I suggest to replace “Fraction Reward Requested” with “Percentage possible rewards”. The authors need not follow my suggestion (the reader will understand what is intended), but I think that “Fraction reward requested” is semantically not correct.

We thank the reviewer for this thoughtful suggestion. However, we have opted to keep the term “Fraction Rewards Requested” in the revised manuscript given that it most accurately captures the units of mouse behavioral performance. In particular, the term accounts for the fact that, generally, mice must lick within reward zones to request a reward, which is then delivered by the Unity software. There were also circumstances in which rewards were automatically delivered (i.e. the first 10 Block A trials and first 10

Block B trials in each session) if a mouse failed to request the reward. Split Maze performance analysis intentionally excluded these automatically delivered rewards to avoid inflating Block phase performance. Moreover, requested rewards do not equate to consumed rewards, as consummatory licks often but not always followed reward request licks. However, we appreciate the reviewer's comment and have further clarified this metric in the revised Figure 1 Results (pg. 4) and legend (pg. 24) and Methods (pg. 38-39).

3. The order of the Figure legends differs in the manuscript and Extended data: they are before the main Figures, but after the Figures in the Extended Data. This also contributes to some extent to making the paper difficult to read.

For improved consistency and readability, all figures now precede their corresponding legends in the main manuscript and Extended Data.

4. Extended Data Fig. 1e is more of a visual contrast sensitivity assessment rather than a visual acuity assessment. The latter would entail visual measuring the finest detail or smallest angle the animal can resolve, and would indeed be the better control. However, at this stage such a demand is not appropriate, and I suggest rewording.

We thank the reviewer for noting that our vision control task is more appropriately described as assessing contrast sensitivity rather than acuity. Towards greater precision, all mentions of the task have been reworded to reflect this in the revised manuscript.

5. Some panels are not mentioned/discussed in the main text (for example Fig. 2c and Extended Data Fig. 8a).

We have verified that all figure panels are referenced in the text and added missing references where necessary. In particular, Fig. 2c is referenced with Figs. 2a-d in the revised Results (pg. 6) and Methods (pg. 52). Extended Data Fig. 8a is referenced in the revised Results (pg. 19) and Methods (pg. 62).

6. Wouldn't the slight improvement in performance in the aged group in Fig. 1e be also expected in Figure 2f for grid cells and Extended Data Fig. 3f-g for NGS cells? If there is a correlation between spatial coding and performance, would this suggest that other cells account for the slight improvement in performance in Fig. 1e for aged mice?

The reviewer raised an important question about what might contribute to the small improvement in Split Maze (SM) task performance over sessions among aged mice (revised Fig. 1g, j), since stable and context-specific spatial firing by MEC grid cells related to performance but did not improve significantly over days among aged mice (Fig 2.). Notably, we found that the change in alternation performance over sessions across age groups was related to the extent of improvement in grid network alignment to context (Fig. 3h) ($r = 0.44$, $p = 0.029$). However, this relationship was not observed among aged mice alone. Additionally, we noted that there are sessions in which aged grid network spatial map alignment to context was high but performance was poor, indicating that context-aligned MEC grid network spatial maps are not sufficient for good aged performance (Fig. 3e, right [bottom right of scatter]). Therefore, we agree that aged SM performance improvement is likely to also depend on other features of MEC activity and/or activity in other brain regions, including hippocampus and cortical areas that facilitate set-shifting.

To investigate this, we assessed whether speed coding by positively (+) or negatively (-) tuned speed cells might also relate to SM task performance (Reviewer Figure 1). + and - speed cell FR-speed slope neither changed over sessions in any age group nor related to SM task performance (Reviewer Figure 1a-d). By

contrast, speed tuning stability improved for young and MA but not aged mice over sessions, similar to grid spatial firing stability (Reviewer Figure 1e-f). This was true across SM task phases for + speed cells and during alternation only for - speed cells. Speed tuning stability was also greater in the block vs. alternation phase, also consistent with grid spatial firing effects. Moreover, + speed cell tuning stability correlated with SM performance across age groups (Reviewer Figure 1g-h) (also revised Extended Data Fig. 6h-j).

Reviewer Figure 1. Speed coding stability by positively speed tuned cells improves over sessions and relates to Split Maze performance (continued from previous page)

- (a) Effect of age and session interaction on positively-modulated (+) speed cell block (left) and alternation (right) firing rate (FR)-speed slope, fitted by separate LMMs ($n = 3933$ young, 4062 MA, and 3423 aged cells). Large dots and vertical bars indicate age group mean and SEM, respectively. Small, pale dots represent LMM-fitted session averages, jittered by age group. Block speed slope (Intercept [Mean \pm Standard Error] = 0.040 ± 0.004 , $p < 0.0001$) was predicted by age (Aged vs. Young, $\beta = 0.021$, $p = 0.0010$) but not by sex or cohort. Alternation FR-speed slope (Intercept = 0.028 ± 0.004 , $p < 0.0001$) was predicted by age (Aged vs. Young, $\beta = 0.027$, $p < 0.0001$) but not sex or cohort. Block and alternation speed slope also declined modestly over sessions among aged mice (Block: Session \times Aged, $\beta = -0.003$, $p = 0.0010$; Alt.: Session \times Aged, $\beta = -0.004$, $p < 0.0001$).
- (b) As in (a) but for negatively-modulated (-) speed cells block (left) and alternation (right) FR-speed slope ($n = 1472$ young, 2071 MA, and 1415 aged cells). Block FR-speed slope (Intercept = -0.026 ± 0.005 , $p < 0.0001$) was predicted by age (Aged vs. Young, $\beta = -0.028$, $p < 0.0001$) but not by sex, cohort, or session. The model of alternation speed slope did not converge.
- (c) Task performance vs. mean FR-speed slope across co-recorded + speed cells, split by task phase for each age group ($n = 108$ young, 116 MA, 108 aged task phases). Dots outlined in black vs. not indicate block vs. alternation phase data. FR-speed slope was unrelated to task performance across phases (Young: $p = 0.26$, MA: $p = 0.47$, Aged: $p = 0.31$) or considering alternation alone (Young: $p = 0.97$, MA: $p = 0.40$, Aged: $p = 0.57$).
- (d) As in (c) for task performance vs. mean FR-speed slope of - speed cells ($n = 106$ young, 116 MA, 106 aged task phases). Black lines indicate linear regressions fit by age group across task phases. Only linear regression fits for significant relationships are plotted. Except among MA mice, FR-speed slope was unrelated to task performance across phases (Young: $p = 0.45$, MA: $r = 0.20$, $p = 0.0298$, Aged: $p = 0.06$) or considering alternation alone (Young: $p = 0.66$, MA: $p = 0.05$, Aged: $p = 0.11$).
- (e) As in (a) for + speed cell block (left) and alternation (right) trial stability speed score. Session predicted improvement in block speed coding stability among young but not aged mice (Session \times Young, $\beta = 0.016$, $p < 0.0001$; Session \times Aged, $\beta = 0.003$, $p = 0.19$) (Intercept = 0.169 ± 0.011 , $p < 0.0001$). Similarly, session predicted improvement in young but not aged alternation speed coding stability (Session \times Young, $\beta = 0.019$, $p < 0.0001$; Session \times Aged, $\beta = -0.004$, $p = 0.263$) (Intercept = 0.111 ± 0.012 , $p < 0.0001$). This suggests that + speed cell speed coding stability fails to improve over sessions among aged mice.
- (f) As in (e) for - speed cell block (left) and alternation (right) trial stability speed score. Session predicted a modest improvement in block speed coding stability in young and aged mice (Session \times Young, $\beta = -0.010$, $p < 0.0001$; Session \times Aged, $\beta = -0.009$, $p = 0.011$) (Intercept = -0.180 ± 0.017 , $p < 0.0001$). Conversely, session predicted a modest improvement in young but not aged alternation speed coding stability (Session \times Young, $\beta = -0.015$, $p < 0.0001$; Session \times Aged, $\beta = -0.004$, $p = 0.263$) (Intercept = -0.166 ± 0.018 , $p < 0.0001$). This suggests that - speed cell speed coding stability fails to improve over sessions among aged mice during alternation specifically.
- (g) As in (c), for task performance vs. mean trial stability speed score of + speed cells. Colored lines indicate regression fits for alternation alone. Trial stability speed score related to task performance across phases (Young: $r = 0.35$, $p = 0.0002$, MA: $r = 0.56$, $p < 0.0001$, Aged: $r = 0.40$, $p < 0.0001$) and when considering alternation alone (Young: $r = 0.27$, $p = 0.0497$, MA: $r = 0.55$, $p < 0.0001$, Aged: $r = 0.24$, $p = 0.097$).
- (h) As in (d) for task performance vs. mean trial stability speed score of - speed cells. Mean trial stability speed score was unrelated to task performance across phases (Young: $p = 0.77$, MA: $p = 0.07$, Aged: $p = 0.11$) or in alternation alone (Young: $p = 0.94$, MA: $p = 0.40$, Aged: $p = 0.79$).

Finally, we observed that the change in alternation speed coding stability over sessions related to the change in alternation performance across age groups ($r = 0.69$, $p = 0.0015$) (not shown). Among aged mice, however, improving alternation performance and speed coding stability were unrelated ($r = -0.25$, $p = 0.63$), suggesting that speed cell firing also does not explain the small aged performance over sessions.

Since neither spatial nor speed coding stability improve over sessions in the aged MEC, we agree that the modest increase over sessions in aged alternation performance is likely explained by improvement in other firing properties in MEC or other brain regions that we cannot measure in our dataset. In line with the reviewer's suggestion, we have now added panels Reviewer Figure 1e-g as Extended Data Fig. 6h-j to highlight these findings in the revised manuscript.

7. Please add the scales and y-ticks in Extended Data Fig.4a-d to allow comparisons across age groups. Since this provides the basis of the remap/alignment analysis, are there differences between different age groups for their optimized average silhouette score?

We thank the reviewer for bringing our attention to this formatting error. Extended Data Fig. 4a-d have been updated with y-ticks to better facilitate comparison of R^2 and average silhouette score values across example sessions and age groups.

Additionally, we have verified that mean session silhouette score at optimal k did not differ across age groups (Reviewer Figure 2), confirming that this does not bias remapping and alignment analyses.

Reviewer Figure 2. Mean silhouette score at optimal k value to fit grid network spatial maps is equivalent across age groups.

- (a) Box and whisker plot of mean optimized silhouette score by Split Maze age group ($n = 46$ young, 47 MA, 41 aged sessions with ≥ 10 grid cells and optimal $k = 2 - 4$) with outliers indicated by black circles. Dots indicate session values, colored by mouse identity (see Extended Data Fig. 1c for color legend). The box and whiskers indicate the interquartile range (IQR) and $1.5 \times$ IQR, respectively for each age group. The mean silhouette score of grid network spatial maps fit at optimized k-value for each session did not differ across age groups (mean session silhouette score, young vs. MA vs. aged, 0.4210 ± 0.0083 vs. 0.4299 ± 0.0078 vs. 0.4250 ± 0.0083 , Kruskal Wallis test, $H = 0.50$, $p = 0.78$).

We have included this verification here for ease of reference, but it is currently not included in the revised manuscript. If the reviewers or editor feel it would be helpful to include, we would be happy to do so.

8. For Fig. 8k, could the authors add the median for these violin plots?

Median lines are now included in all violin plots that compare excitatory cell and interneuron gene expression (see revised Figs. 7k, m and Extended Data Figs. 7g, i and 8e, g, i, k, m, and o).

Reviewer 2: This work studied spatial coding in the aging medial entorhinal cortex (MEC). The authors found disfunction of spatial coding and instable network with age. In vivo electrophysiology identified spatial memory deficits arising with impaired stabilization of context-specific spatial firing. Aged grid networks appeared shifting firing patterns often but with poor alignment to context dynamics. Aged spatial firing was also unstable in an unchanging environment. They further employed transcriptomic tools and identified 61 spatial coding quality-related genes differentially expressed with age in MEC. These genes were interneuron-enriched and related to synaptic plasticity, notably including a perineuronal net component, according to bioinformatic analyses. Collectively, the authors accumulated evidence from the perspectives of neuronal firing and transcriptome, demonstrating the spatial coding ability destabilize with age. The experiments were well designed and the results are sound. However, I have a few concerns that should be addressed to strengthen this manuscript.

We thank the reviewer for their comprehensive summary of the manuscript and their appreciation of the strength of our experimental design.

Major Concerns:

1. Throughout this manuscript, I did not find originally recorded data formatting place field. I would suggest the authors to include some examples.

Hippocampal (HPC) recordings from place cells were not performed as part of this study, and such recordings would take several years. Thus, given the breadth of the current manuscript, we feel adding this would be beyond the scope of the current study. In the current work, we elected to prioritize recording from medial entorhinal cortex (MEC) because there were no previous studies of the healthy aged MEC. By contrast, there is a decades-long body of work that has elucidated how hippocampal place coding changes with healthy aging that we discuss in the Introduction (pg. 3). As such, we believe that HPC place field recordings would not improve the manuscript's novelty or impact. However, we designed our VR tasks to examine MEC grid coding in stable, familiar vs. changing, novel environmental contexts to mirror the breadth of this past hippocampal work and to facilitate comparisons with it in the manuscript's Discussion (pg. 19-20). This approach led to the observation that aging induces strikingly consistent changes in spatial coding function across HPC place cell and MEC grid cell populations. In particular, aging impairs flexible, experience-dependent, and stable spatial coding by place and grid cells; increases the frequency of cue-independent remapping by place and grid cell populations; and reduces the power of theta rhythm that coordinates MEC and HPC spatial firing.

As a result, we concur with the reviewer that co-recording of the reciprocally connected cell types in EC and HPC involved in spatial coding would greatly enrich the field and outline this as a priority for future work in the Discussion section of the manuscript (pg. 20-21).

2. The molecular/cellular mechanisms underlying changes of MEC neuronal firing with age regarding destabilized spatial coding were primarily from transcriptomic data. Ideally, the author should provide evidence at the protein level, coherent with the gene/s (for example, Hapln4) exhibiting the top changed expression profile to consolidate their conclusion.

The reviewer raises an important point that not all transcriptomic changes differences are reflected at the protein level, which makes it critical to validate hypotheses about molecular and cellular changes mediated by proteins generated by transcriptomic data. One notable contributor to this is that RNA sequencing and subsequent analyses can have high false positive rates. As such, we have taken multiple steps to mitigate

the risk of false positive correlations between spatial coding and molecular changes in the aging MEC in our dataset.

In particular, we extended the application the Benjamini-Hochberg false discovery rate (FDR) procedure from the analysis of differential gene expression to our correlation analysis (Fig. 7 and Extended Data Figs. 7 and 8) to better control for multiple hypothesis testing among genes related to spatial coding. We opted for a stringent FDR of 0.10 to present the most concise and specific list of putative transcriptomic correlates of aged spatial coding dysfunction (Fig. 7 and Extended Data Fig. 7). At most, 10% of transcriptomic correlates of spatial coding (32/316) are expected to be false positives.

Moreover, we included immunohistochemical analyses demonstrating an age-mediated increase in the density of perineuronal nets (PNNs) at the protein level in the MEC using the fixed hemispheres of Random Foraging (RF) task mice (Fig. 8a-d). As the reviewer suggested, we aligned this change with the observed up-regulation of *Hapln4* (a component of PNNs) (Fig. 7b) and its correlation with spatial coding coherence (Fig. 7g). Validating our transcriptomic analysis, increased *Hapln4* expression in MEC from one hemisphere was related to increased PNN density in the opposite hemisphere across age groups in these mice (Fig. 8d). Importantly, previous work by Sucha et al. in 2020 directly showed that HAPLN4 PNN density increases with age as we would hypothesize based on this correlative relationship.

We acknowledge that detecting HAPLN4 protein expression in young vs. aged MEC using a specific antibody would have been an alternative and rigorous transcriptomic validation approach. We tried this approach initially but were not able to reliably detect HAPLN4 PNNs over background, despite extensive troubleshooting with two commercial anti-HAPLN4 antibodies and multiple variants of antigen retrieval approaches. Given the limited MEC tissue we had from RF mice that underwent electrophysiological recording and MEC neuron RNA sequencing, we opted to instead follow highly reproducible WFA staining protocols common to the PNN field. This allowed us to explore the area and staining intensity of MEC PNNs and their co-localization with parvalbumin-expressing interneurons (PV+ INs) that commonly produce them. While aging did not affect MEC PNN size or the density of N-acetylgalactosamine residues on chondroitin sulfate proteoglycans to which WFA bind, we found that aging alters MEC PNN distribution across excitatory and inhibitory cell types. In particular, the age-related MEC PNN increase was driven by PNNs not co-localized with the PV+ INs (Fig. 8c) and that the density of these additional, differentially localized PNNs correlated with impaired spatial coding quality (Fig. 8d).

3. Figure 4: Maybe I missed something. The calculation process for band power is not clear to me. In panel b, either the value or the unit for the Power Spectral Density (V^2/Hz) is far beyond the reasonable range. What is the unit for power in panel c?

We apologize for this error and thank the reviewer for noticing the typo in the units of power spectral density in Fig. 4b, which have been corrected to $\mu V^2/H$, from V^2/Hz , to reflect that LFP signals from Neuropixels are in the amplitude range of μV , not V . The unit for power in Fig. 4c has now been specified as $\mu V^2/Hz$ as well.

As described in the manuscript Methods (see pg. 59), power spectral density was calculated using Welch's method. In brief, power in each analyzed frequency band was calculated using the `spectral_connectivity` package with a window of 0.5 seconds (Denovellis et al., 2022; <https://spectral-connectivity.readthedocs.io/en/latest/>). In detail, in each frequency range of interest, we used the `Multitaper` class from this package to compute the multitaper Fourier transform of the LFP signal recorded from the probe channel with the greatest theta power in MEC. Then, we used `Connectivity` class to compute power from the resulting Fourier coefficients.

4. The authors used both male and female mice in this study. While making each statement based on data collected from animals with both sexes, the sex factor should be considered and clearly stated.

We thank the reviewer and agree it is critical to consider sex as a fixed effect in our study design and as a potential modifier of the effects of age. As mentioned in the manuscript Methods, we do include animal sex and/or the interaction of animal sex and age group as fixed effects in the linear mixed effects models applied to interpret our behavioral and electrophysiologic data (see pg. 48-49). Using this approach, we observed no significant effects of male vs. female sex itself on spatial memory performance or spatial coding (see new Supplementary Table 1 for full model results). One notable exception highlighted in the manuscript Results (pg. 6) was that male sex predicted better Split Maze (SM) alternation phase performance among aged mice alone (Extended Data Fig. 1h), concurring with previous work (see Discussion, pg. 19). The negative results of all other sex and sex-age interaction hypothesis testing are also included in the output of Python notebooks containing the original code used to generate manuscript figures and made available to readers upon publication. Notably, when the interaction of sex and age group was not a significant predictor of a particular dependent variable and did not improve model fit, it was removed from the final model. We have clarified this in the revised Methods (pg. 48-49).

Next, towards identifying transcriptomic mediators of the strong sexual dimorphism in aged SM alternation performance, we identified 15 autosomal genes that were differentially expressed between aged female and male SM mice in MEC neurons (Extended Data Fig. 8p), notably including immediate early genes *Arc* and *Egr1* and Alzheimer's disease risk gene *ApoE*. Moreover, within aged SM mice, *Egr1* and *Arc* expression related strongly to improvement in SM alternation performance and grid network context alignment (Extended Data Fig. 8q), consistent with their interdependent roles in synaptic plasticity and learning (Li et al. 2005; Shepherd & Bear, 2011; Czerniawski et al., 2011; Wall et al. 2018). In addition, consistent with previous work showing that abnormally persistent *Arc* expression is associated with impaired cognitive flexibility (Wall et al., 2018), aged female mice exhibited significantly worse SM alternation performance and higher *Arc* expression after six recording sessions than their male counterparts. This suggests that the temporal dynamics of MEC *Arc* expression may be dysregulated in aging in sex-specific manner related to impaired spatial memory. Extended Data Fig. 8p-q is now more clearly referenced in the revised Discussion in relation to sex effects on transcriptomic MEC aging (pg. 19).

Suggestions & Minor Concerns:

1. P2: "hexagonally tile physical space": grammar problem

We thank the reviewer for bringing our attention to the improper parallel structure in this sentence. The revised Introduction (pg. 2) now states: "The MEC contains grid cells that fire periodically during environmental traversals and have firing fields that hexagonally tile physical space in rodents, non-human primates, and humans."

2. In Figure 3b, there are several black circles. What do they indicate?

In all box and whisker plots in the manuscript, these black circles indicate outliers (data points $\geq 1.5 \times$ interquartile range of the group). We note this in the legend for Fig. 2d, which is the first such plot.

Reviewer #3: This study reports an impressive dataset describing neuronal firing and gene expression in the medial entorhinal cortex of young, middle-aged and old mice. Important results include age-related deficits in context-dependent behavior, changes with age in the stability and context-dependence of grid cell firing, and age-dependent changes in gene expression that correlate with the electrophysiological deficits. The experiments appear carefully designed and executed. The analyses are thorough and in general convincing. The results will be influential, as they give the first compelling insights into how entorhinal cortex circuit physiology changes with age, and offer hints about underlying molecular changes.

We thank the reviewer for emphasizing the quality of our electrophysiological and transcriptomic datasets, the rigor of our analysis approaches, and the potential impact of the manuscript on the field.

Suggestions & Minor Concerns:

1. A potential concern is that some of the comparisons of spiking data between young, middle-aged and aged groups appear to rely on whether relationships are, or are not, significant within a group. E.g. on p7 there is a reported decrease in stability over sessions for young and middle-aged but not aged mice. The lack of significance in the aged group is insufficient to conclude that it differs from the other groups. More appropriate would be to evaluate models that include age as a factor (as is done for the behavioural analysis). From my reading of the methods this appears to have been done but not reported. I suspect the conclusions will hold, but without reporting these analyses one can't rule out that age has made the data more noisy and so 'significant' relationships are less easily detectable.

As the reviewer noted, age group was included as a categorical fixed effect in all linear mixed effects models, along with the interaction of session number (continuous) and age group (categorical) (Methods, pg. 48-49). This was the case for all models of behavior (incl. revised Fig. 1g [models of block and alternation performance) and spiking data (incl. Fig. 2f-g [models of grid cell spatial firing stability and context-specificity]). In these models, significant age group effects would indicate the y-intercept of relationships between session and performance (or spiking metrics) differ with age, whereas significant session-age group interactions equates to the slope of the relationship between session and performance (or spiking metrics) differ across age groups. As reflected in the plotted data (e.g. Fig. 1j and Fig. 2f-g), age group effects are indeed modest or non-significant while session-age group interactions are significant. Whenever age fixed effects were not significant, we omitted them from the manuscript Results and/or figure legends to improve text concision and optimize clarity for readers. Per the reviewer's suggestion below, we have added a new Supplementary Table 1 that details the complete results of each model, including the beta- and p-values of non-significant fixed effects (see response to Point 5).

Additionally, we would like to clarify that models of spiking data include Session x Age Group as a fixed effect while models of behavioral data include both Session and Session x Age Group as fixed effects. We agree with the reviewer that this limited us to reporting only whether each age groups' spiking significantly changed over sessions or not in the original manuscript, whereas we were able to state that aged mice improved less behaviorally over sessions than young ones. Notably, it is also true that aged grid (and non-grid spatial [NGS]) cell spatial firing stability and context-specificity improve significantly less than that of young grid and NGS cells. This is demonstrated by adding a Session fixed effect to the original spiking data models (see Reviewer Table 1-2 comparison of simple vs. full models). Moreover, the log-likelihood of simple and full models of grid and NGS cell spiking properties are equivalent, indicating equal fit to the data. We opted to present simple models of spiking data (only Session x Age Group effects) in the manuscript for greater concision and clarity. This permits readers to intuitively relate Session x Age Group beta-values

to how the change in cell spiking features across sessions differs among age groups in Fig. 2f-g and Extended Data Fig. 3f-g. If the full model results were reported, readers would be required to (a) understand a Session beta-value reflects the base group (young age group) slope over sessions and (b) add Session and Session x Aged beta-value to calculate the change over sessions among aged cells.

To clarify this, “full” model results are included in the new Supplementary Table 1 (see Point 5) tabs corresponding to Fig 2f-g and Extended Data Fig. 3f-g. A note has also been added to the revised Methods to indicate that indeed, statistically, aged spatial firing improves less than young over sessions (pg. 48).

Finally, Reviewer Table 1-2 includes the standard error for each beta value. Please note that these are similar across age groups. This provides reassurance that it is unlikely that additional noisiness in aged data impairs the detection of significant relationships between spiking properties and session.

Model Characteristics			Simple Model Results		Full Model Results	
Cell Type	Task Phase	Dep. Variable	LL	Session x Age Group (Young [Y], Middle-Aged [MA], Aged [A]) Effects	LL	Session (S) [Young] and S x Age Group Effects
Grid	Block	Stability	1847.9	Y: $\beta = 0.011 \pm 0.002$, $p < 0.0001$ MA: $\beta = 0.036 \pm 0.003$, $p < 0.0001$ A: $\beta = 0.000 \pm 0.003$, $p = 0.881$	1847.9	S: $\beta = 0.011 \pm 0.002$, $p < 0.0001$ S x MA: $\beta = 0.025 \pm 0.003$, $p < 0.0001$ S x A: $\beta = -0.011 \pm 0.004$, $p = 0.002$
Grid	Alternation	Stability	4457.5	Y: $\beta = 0.009 \pm 0.001$, $p < 0.0001$ MA: $\beta = 0.027 \pm 0.002$, $p < 0.0001$ A: $\beta = -0.010 \pm 0.002$, $p < 0.0001$	4457.5	S: $\beta = 0.009 \pm 0.001$, $p < 0.0001$ S x MA: $\beta = 0.018 \pm 0.002$, $p < 0.0001$ S x A: $\beta = -0.019 \pm 0.002$, $p < 0.0001$
Grid	Block	Similarity Ratio	-984.6	Y: $\beta = 0.022 \pm 0.003$, $p < 0.0001$ MA: $\beta = 0.064 \pm 0.004$, $p < 0.0001$ A: $\beta = -0.005 \pm 0.005$, $p = 0.277$	-984.6	S: $\beta = 0.022 \pm 0.003$, $p < 0.0001$ S x MA: $\beta = 0.042 \pm 0.005$, $p < 0.0001$ S x A: $\beta = -0.027 \pm 0.006$, $p < 0.0001$
Grid	Alternation	Similarity Ratio	7102.6	Y: $\beta = 0.012 \pm 0.001$, $p < 0.0001$ MA: $\beta = 0.011 \pm 0.001$, $p < 0.0001$ A: $\beta = 0.001 \pm 0.001$, $p = 0.398$	7102.6	S: $\beta = 0.012 \pm 0.001$, $p < 0.0001$ S x MA: $\beta = -0.002 \pm 0.001$, $p = 0.243$ S x A: $\beta = -0.011 \pm 0.002$, $p < 0.0001$

Reviewer Table 1. Simple and full models of session x age effects on grid cell spatial firing stability and context-specificity (similarity ratio) are mathematically equivalent (related to manuscript Fig. 2f-g).

Each row conveys the results of two separate linear mixed effects models of the same dependent (dep.) grid cell variable (spatial firing stability or similarity ratio, as a proxy of context-specificity) during a particular Split Maze task phase (block or alternation). Simple model refers to a model with only Session x Age Group modeled as a fixed effect, compared to full models, which also include Session as a fixed effect in addition to the Session x Age Group interaction. For each model, we report the log-likelihood (LL), beta-value \pm standard error, and p-values. When simple models supported the hypothesis that a grid cell spiking quality

improved significantly over sessions for young but not aged cells, the results are highlighted green. When full models supported the hypothesis that aged grid cell improvement over sessions was less than that of young grid cells ($S \times A: \beta < 0, p < 0.05$), the results are highlighted blue.

Model Characteristics			Simple Model Results		Full Model Results	
Cell Type	Task Phase	Dep. Variable	LL	Session x Age Group (Young [Y], Middle-Aged [MA], Aged [A]) Effects	LL	Session (S) [Young] and S x Age Group Effects
NGS	Block	Stability	2019.1	Y: $\beta = 0.018 \pm 0.002$, $p < 0.0001$ MA: $\beta = 0.030 \pm 0.002$, $p < 0.0001$ A: $\beta = 0.006 \pm 0.003$, $p = 0.042$	2019.1	S: $\beta = 0.018 \pm 0.002$, $p < 0.0001$ S x MA: $\beta = 0.012 \pm 0.003$, $p < 0.0001$ S x A: $\beta = -0.012 \pm 0.004$, $p = 0.001$
NGS	Alternation	Stability	3889.9	Y: $\beta = 0.011 \pm 0.002$, $p < 0.0001$ MA: $\beta = 0.023 \pm 0.002$, $p < 0.0001$ A: $\beta = -0.001 \pm 0.002$, $p = 0.540$	3889.9	S: $\beta = 0.011 \pm 0.002$, $p < 0.0001$ S x MA: $\beta = 0.011 \pm 0.002$, $p < 0.0001$ S x A: $\beta = -0.013 \pm 0.003$, $p < 0.0001$
NGS	Block	Similarity Ratio	755.7	Y: $\beta = 0.024 \pm 0.003$, $p < 0.0001$ MA: $\beta = 0.031 \pm 0.003$, $p < 0.0001$ A: $\beta = 0.006 \pm 0.004$, $p = 0.089$	755.7	S: $\beta = 0.024 \pm 0.003$, $p < 0.0001$ S x MA: $\beta = 0.007 \pm 0.004$, $p = 0.086$ S x A: $\beta = -0.018 \pm 0.004$, $p < 0.0001$
NGS	Alternation	Similarity Ratio	8093.1	Y: $\beta = 0.007 \pm 0.001$, $p < 0.0001$ MA: $\beta = 0.007 \pm 0.001$, $p < 0.0001$ A: $\beta = 0.004 \pm 0.001$, $p = 0.002$	8093.1	S: $\beta = 0.007 \pm 0.001$, $p < 0.0001$ S x MA: $\beta = 0.000 \pm 0.001$, $p = 0.971$ S x A: $\beta = -0.004 \pm 0.001$, $p = 0.016$

Reviewer Table 2. Simple and full models of session x age effects on NGS cell spatial firing stability and context-specificity (similarity ratio) are mathematically equivalent (related to Extended Data Fig. 3f-g).

Each row conveys the results of two separate linear mixed effects models of the same dependent (dep.) non-grid spatial (NGS) cell variable (spatial firing stability or similarity ratio, as a proxy of context-specificity) during a particular Split Maze task phase (block or alternation). Simple model refers to a model with only Session x Age Group modeled as a fixed effect, compared to full models, which also include Session as a fixed effect in addition to the Session x Age Group interaction. For each model, we report the log-likelihood (LL) and beta-value \pm standard error and p-values. When simple models supported the hypothesis that NGS cell spiking improved significantly over sessions for young but not aged NGS cells, the results were highlighted green. When full models supported the hypothesis that aged NGS cell improvement over sessions is also likely less than that of young NGS cells ($S \times A: \beta < 0, p < 0.05$), the results are highlighted blue.

2. When comparisons of spiking population data are made between groups, the reporting should clarify what the unit of analysis is. My guess is that these are single, or median, observations per animal. If multiple observations/sessions per animal have been used then a nested analysis would be more appropriate.

Generally, the units of analysis for spiking linear mixed effects models (LMMs) are cells (Figs. 2f-g, 6j, and Extended Data Figs. 2h, j, 3f-g, 6h-i). Indeed, these linear mixed effects models have nested designs to account for the variability at the cell and animal levels relevant to biological variables like age and sex (see Methods, pg. 48). To best illustrate quantified effects, the corresponding figures report LMM-fitted session and animal means, as stated in the figure legends. We have ensured that the Methods clearly present cells as the units of analysis for neural activity LMMs, and we have specified the units of analysis for each LMM in the manuscript in Supplementary Table 1.

Notably, several spiking-related metrics could only be calculated at the session or task-phase level, such as the density of classified cell types, network remapping frequency, network map-context alignment fraction, LFP power, and decoder score. When this was the case, session was specified to be the unit of analysis in the corresponding figure legend (see Figs. 2d, 3b, 4b-c, 5d, 5i and Extended Data Figs. 2a, 4h, 5c-e, j, 6d [first] and 6e). In each case, we verified that results of non-parametric statistical comparisons across two or three age groups (using Wilcoxon Rank Sums or Kruskal-Wallis tests followed by post-hoc Conover correction for multiple comparisons, respectively) held when linear mixed effects models were constructed with session as the unit of analysis, grouped by animal. To improve readability, we opted to present the former analysis in these cases.

Next, some spiking metrics calculated at the cell level were compared across age groups using the non-parametric tests above, after averaging the metric across co-recorded cells in sessions (Figure 5e-g, 6d-e and Extended Data Fig. 6d [second to last], f, g). As such, in these figures, session was the unit of analysis, which we have specified in the corresponding figure legends. As co-recorded cells have highly correlated firing in MEC, this approach is statistically more robust than considering each cell an independent measure (Stensola et al., 2012; Beed et al., 2013; Campbell et al. 2018; Butler et al., 2019; Hardcastle et al., 2017). However, it lacks the other advantages of LMMs (e.g. handling missing data like a mouse without data from all six sessions and/or flexibly modeling complex interaction effects). In each case, we also verified that results of non-parametric statistical comparisons across two or three age groups held when nested linear mixed effects models were constructed with cells as the unit of analysis. We also have clarified this point in the revised Methods (pg. 47). When no session or session x age group interaction effects were observed by LMM, we opted for the approach of averaging metrics across cells in a session for greater clarity and simplicity for readers. The one exception to this rule was Fig. 6e given that trial stability speed score was session-modulated (see Reviewer Figure 1e-f). Fig. 6e results were analyzed by statistical comparisons across session mean metrics for consistency with Fig. 6d.

Finally, there are a few instances of statistical comparisons of spiking metrics calculated at the cell level, across cell types within age groups or between real and shuffled cell data pooled across age groups (e.g. Fig. 2c, 5b-c, 6b-c, and Extended Data Figs. 2b-d, f and 3c, 6b-c). This was appropriate as the purpose of these analyses was not to assess the effects of biological variables on spiking, but rather to validate the classification of functional cell types, both across and within age groups.

3. Can the methods include an explanation for how the pseudorandomisation for alternating trials was implemented?

A note has been added to Methods (pg. 38) to better specify how alternation trial context was pseudo-randomized in the Split Maze VR task.

4. In Figure 1 the implementation of the alternation phase of the split maze task is potentially confusing. In the schematic in Figure 1c, it appears that trials alternate, rather than being

randomised, while the y-axis trial numbering in panel 1d implies ordering, whereas the alternating blocks have been re-ordered. This is clear after cross-checking with the methods section, but for the benefit of a lazy reader perhaps modify this part of the axis and clarify in the legend?

To better clarify for readers that context is randomized on each trial during the alternation phase of the Split Maze, we have added an explanatory note to the legend of Fig 1c. in the revised manuscript. In addition, we have now added clarifying statements to the legends for Split Maze lick and spike raster plots (revised Figs. 1f [formerly Fig. 1d] and 2a) to emphasize that alternation trials are sorted by context, such that trial number does not reflect trial order.

5. The methods should clarify the tests used for estimation of statistical significance from linear mixed effect models. A supplemental table reporting the degrees of freedom and test statistics for each of the analyses could be helpful. Procedures used to assess validity of model assumptions should also be described.

The Methods section on linear mixed effects modeling (LMM, pg. 48-49) has been updated to include additional details on the estimation of model statistical significance using Wald testing and the validation of model assumptions, specifically linearity, homoscedasticity, and normality of residuals. Wald testing results and code to optionally generate residual plots that confirm model assumptions are included in the Python notebooks associated with each LMM in the code repository that will be made available to readers upon publication. Additionally, we have added a new Supplementary Table (revised Supplementary Table 1) to the revised manuscript that includes full LMM results, including Wald testing results, degrees of freedom per fixed effect, and test statistics related to Figs. 1g-h, 2f-g, and 6j and Extended Data Figs. 1g-k, 2h, 2j, 3f-g, and 6h-i.

In response to Point 1 (see above) and to support the conclusion that spiking metrics improve significantly less for aged vs. young mice over sessions, the tabs corresponding to Figs. 2f-g and Extended Data Figs. 3f-g and 6h-i include both simple and full (Session effect added) model results.

In response to Point 2 (see above) and to clarify the unit of analysis for each LMM-based analysis, each model is labeled with its unit of analysis.

We thank the reviewers for their supportive comment regarding the revised manuscript. Below, we have addressed the remaining comments of Reviewer 2, further improving the clarity of the manuscript.

Summary of changes in response to reviewers:

1. The legend for Figure 1j now notes the exclusion of certain behavior sessions, explained in detail in the manuscript Methods (pg. 39). In tables included here, we demonstrate that the exclusion of any completed behavioral sessions does not impact the study's conclusions. We have also added a point to the manuscript's Methods stating this.
2. A less compressed version of Figure 8 with higher image resolution images is included in this submission. We have clarified in the legend of Figure 8a that our 2D representative images are the product of summing across Z steps through a 3D volume and are therefore consistent with our image processing and analysis approach.
3. We have included a link below for reviewers to preview the Dryad data repository containing the processed behavioral, neural, and immunohistochemical data that will be made publicly available upon manuscript publication. We have also updated our manuscript's Data availability statement to provide the options for users to request the raw neural dataset.

Response to Reviewer Comments:

Reviewer #1:

This is an important paper in the field and deserves to reach a large readership as soon as possible. I have no further requests. We actually use some of the code and find it very useful. In doing so, we made a new discovery and are very grateful to the authors.

We thank the reviewer for their positive comments on our study's impact and the quality and utility of the related code. We are happy to hear that our code has helped enable a new discovery.

Reviewer #2:

The clarity and readability of the manuscript have significantly improved, and the previous comments and concerns have been adequately addressed. I only have a few remaining comments, listed below:

We thank the reviewer for their positive comment regarding the improved clarity of the manuscript and their helpful additional comments, which we have addressed below.

1. Figure 1, panel j: In the MA group, two lines terminate at Session 3 and Session 4, respectively, while in the Aged group, two lines terminate at Session 3. This information should be clearly stated in the figure legend, and the potential impact of these terminations on the study's conclusions should be addressed in the discussion. Otherwise, I would recommend that the authors consider removing these samples.

We have updated the Figure 1j legend to note the exclusion of certain behavioral sessions and cite our manuscripts' Methods section, which describes the reason for early termination of recordings in certain mice and details our exclusion criteria for behavioral session data (pg. 39). In brief, reflecting humane husbandry practices, when mice exhibited craniotomy infection, aberrant extended freezing behavior, or other clear signs of distress, we terminated certain recording sessions early. To avoid conflating aging effects on cognition with impairments from systemic illness or distress in these middle-aged or aged mice, we excluded behavioral data from the two last sessions (the terminated session and the one prior with qualitatively normal animal wellbeing) from analysis. Notably, the behavioral data from any completed sessions that were excluded from analysis are also included in the Dryad data repository linked below.

Finally, we have also verified that reported age group effects on Split Maze performance across block and alternation (alt.) phases are not altered by including the four completed but excluded sessions (see tables below). Incomplete sessions (i.e. no or fewer alternation phase trials) are still omitted here to avoid biasing our performance metric, calculated as the fraction of trials on which rewards were requested. We used linear mixed effects models in part because they are more robust to missing data compared to t-testing and ANOVAs (Yu et al., 2022). A note emphasizing this fact and that we verified that the study's behavioral conclusions are not impacted by these exclusions is included in the revised Methods (pg. 49).

Split Maze Block Phase	Session	Session x Aged	Aged vs. Young
Sessions Excluded	$\beta = 0.045, p = 0.004$	$\beta = -0.008, p = 0.709$	$\beta = -0.182, p = 0.079$
No Session Excluded	$\beta = 0.045, p = 0.003$	$\beta = -0.020, p = 0.369$	$\beta = -0.209, p = 0.037$

Split Maze Alt. Phase	Session	Session x Aged	Aged vs. Young
Sessions Excluded	$\beta = 0.106, p < 0.0001$	$\beta = -0.085, p < 0.0001$	$\beta = -0.292, p = 0.059$
No Session Excluded	$\beta = 0.180, p < 0.0001$	$\beta = -0.086, p = 0.0010$	$\beta = -0.285, p = 0.073$

2. Figure 8: The resolution of the representative images in panel a is quite low. Additionally, since the statistics presented in panels b and c, are based on density per μm^3 (cubic volume), it would be more appropriate to include 3D representative examples to better illustrate and support the data shown in panels b and c.

In this submission, we have included the least compressed, highest resolution images possible in Figure 8a while meeting constraints on file sizes for submissions in review. We will work with the editor to publish the original, uncompressed images. We also appreciate the reviewer's suggestion regarding 3D representative examples. However, we have chosen to keep representative images in 2D as parvalbumin (PV) interneuron and perineuronal net density analysis was performed after summing across the 10 x 1.5 micron steps in each image (see Methods, manuscript pg. 66). Therefore, to most closely reflect the slice-summed images analyzed via PIPSQUEAK, Figure 8a images are slice-summed composite of PV and WFA channel signals. To clarify that the representative images in Figure 8a are the result of Z slice-summing, we have updated the corresponding figure legend. All processed 2D immunohistochemistry images after slice-summing and the results of imaging processing and analysis are included in our Dryad data repository linked below.

3. Ideally, examples of raw data from the in vivo silicon probe recordings, along with the corresponding identified cell activity, should be provided.

We agree with the reviewer about the importance of making neural recording data accessible to the public. As stated in the manuscript's Data availability section, we will make the pre-processed behavioral data, neural data, and immunohistochemistry images associated with this manuscript available to the public upon manuscript publication. This Dryad data repository is complete, annotated, and set up to be used in conjunction with our GitHub code code. It may be previewed by reviewers here: http://datadryad.org/share/pWTDvwUiB3yIXkRK40ZgzWgOyY3_eqOcosPx91A_0wA.

We have opted to share pre-processed rather than raw neural data for several reasons. First, raw neural data are approximately 100 Gb / hour in size while pre-processed behavioral and neural data are only 100 MB / hour size (total dataset size: ~21.6 Tb vs. 21.6 Gb). Sharing the raw data would be significantly more costly, as most approved repository tools do not support cost-free sharing of datasets over 50 Gb. Second, users would need to re-process and then synchronize raw neural data with raw behavioral data before being able to replicate our analyses or implement new ones. This would be highly time-consuming for users, as the ecephys pipeline, unit curation, and synchronization steps take on average 4-6 hours for each session (given 254 sessions, ~1270 hours or 53 days). Third, the Kilosort algorithm that assigns spikes to clusters corresponding to individual cells, as part of the ecephys pipeline, is not deterministic. This yields variable numbers of high-quality units each time it is run with arbitrary cell identifiers. Additionally, subsequent unit curation has significant inter-user variability and was therefore performed by a single rater. Taken together, this means that re-processed datasets would neither be immediately compatible with our available code nor exactly replicate the manuscript's figures. While we expect that analyzing re-processed data would closely reproduce the paper's major findings, it is notable that the variability introduced by re-processing would be compounded by stochastic analysis processes like cell type classifications based on shuffle procedures and K-means clustering of trials into spatial maps (described further in our GitHub README). Given these three factors, we believe that pre-processed data constitute the most efficient, accessible, and informative way of sharing this dataset. To accommodate individual users who may request raw neural data, we have updated our Data availability statement to indicate that they may contact the corresponding author to devise an individualized approach to transferring that large volume of data.

One important tradeoff of providing pre-processed versus raw neural data is that users can access only the template waveform for each unit and spike times versus the waveforms of each individual spike fired by a given unit. To address this and to enable recapitulation of analyses of waveform features and quality metrics for all recorded units included in the manuscript, we provided the results of comprehensive waveform feature analyses output by ecephys in our Dryad repository (see waveform_metrics folder).

Reviewer #3:

My concerns have been addressed. The study describes an important dataset that gives new insight into age-related changes in spatial cognition.

We thank the reviewer for their positive comments regarding the manuscript's impact.